# HUMAN BEHAVIOR ATLAS: BENCHMARKING UNIFIED PSYCHOLOGICAL AND SOCIAL BEHAVIOR UNDERSTANDING

**Keane Ong[1,2]\***, **Wei Dai[1]\***, **Carol Li[1]**, **Dewei Feng[1]**, **Hengzhi Li[1,3]**, **Jingyao Wu[1]**,
**Jiaee Cheong[4]**, **Rui Mao[5]**, **Gianmarco Mengaldo[2]**, **Erik Cambria[5]**, **Paul Pu Liang[1]**
[1]MIT    [2]National University of Singapore    [3]Imperial College London
[4]Harvard University    [5]Nanyang Technological University

## ABSTRACT

Using intelligent systems to perceive psychological and social behaviors, that is, the underlying affective, cognitive, and pathological states that are manifested through observable behaviors and social interactions, remains a challenge due to their complex, multifaceted, and personalized nature. Existing work tackling these dimensions through specialized datasets and single-task systems often miss opportunities for scalability, cross-task transfer, and broader generalization. To address this gap, we curate HUMAN BEHAVIOR ATLAS, a unified benchmark of diverse behavioral tasks designed to support the development of foundation models for understanding psychological and social behaviors. HUMAN BEHAVIOR ATLAS comprises over 100,000 samples spanning text, audio, and visual modalities, covering tasks on *affective states*, *cognitive states*, *pathologies*, and *social processes*. Our unification efforts can reduce redundancy and cost, enable training to scale efficiently across tasks, and enhance generalization of behavioral features across domains. On HUMAN BEHAVIOR ATLAS, we train three models: OMNISAPIENS-7B SFT, OMNISAPIENS-7B BAM, and OMNISAPIENS-7B RL. We show that training on HUMAN BEHAVIOR ATLAS enables models to consistently outperform existing multimodal LLMs across diverse behavioral tasks. Pretraining on HUMAN BEHAVIOR ATLAS also improves transfer to novel behavioral datasets; with the targeted use of behavioral descriptors yielding meaningful performance gains. The benchmark, models, and codes can be found at: `https://github.com/MIT-MI/human_behavior_atlas`.

## 1 INTRODUCTION

Developing AI systems to perceive and understand psychological and social behaviors, that is, the underlying affective, cognitive, and pathological states that are manifested through observable behavior and social interactions, has long been a central goal of AI (Picard, 2000; Breazeal, 2000). However, these behaviors are complex, multifaceted, and personalized, which makes it challenging to capture their diverse dimensions (Pantic and Rothkrantz, 2003; Inkpen et al., 2019; Mathur et al., 2024; Liang et al., 2024b). Existing work typically tackles these complexities through specialized datasets and single-task systems, e.g., sentiment analysis (Cambria, 2016), depression detection (Yang et al., 2025), human action recognition (Shaikh et al., 2024). However, despite the breadth of efforts, current models remain primarily specialized for the task they were trained on, rather than capturing a holistic understanding of psychological and social behaviors. This constrains reuse or extension beyond their originally intended domains, leading to inefficiency and costly duplication of effort. Each task typically demands bespoke architectures, fresh data collection, and separate training pipelines (Gao et al., 2024). Additionally, focusing on niche task-specific solutions misses opportunities for scalability, cross-task transfer, and broader generalization (Chen et al., 2024a; Dai et al., 2025).

Yet, training a foundation model that can process the full spectrum of psychological and social behaviors remains a major challenge (Amin et al., 2023). Despite the volume of available datasets, the longstanding emphasis on single-task specialization has led to highly disparate formats and evaluation protocols. On the one hand, datasets vary in their input representations. Some rely on pre-extracted features (Zadeh et al., 2016; Ringeval et al., 2019; Liang et al., 2024a), while others operate on raw video and audio clips (Cao et al., 2014; Livingstone and Russo, 2018; Zadeh et al., 2018). On the

---

\*Equal contribution

other hand, these approaches diverge in their output formats and evaluation practices, with subjective labeling schemes (Susanto et al., 2020) and inconsistent benchmarking conventions (Kollias, 2022), hindering consistent evaluation across tasks. As a result, the community has seen limited success in training foundation models for psychological and social behavior analysis.

To address these gaps, we build HUMAN BEHAVIOR ATLAS, a benchmark to support the development of foundation models for general psychological and social behavior understanding. HUMAN BEHAV-IOR ATLAS captures a diverse range of datasets and tasks under a broad taxonomy of psychological and social behaviors (affective states, cognitive states, pathology, social processes), while standardizing them into a consistent format for training a foundation model. To achieve this, all datasets are converted into a common schema by reformulating each sample into a prompt–target format, where prompts consistently reference or contain the available modalities (e.g., transcripts, audio, video) and targets are explicitly standardized into either free-text responses or discrete label sets. In addition, evaluation metrics are standardized across datasets to ensure comparability while maintaining task-specific nuance. We further augment the benchmark with behavioral descriptors extracted from MediaPipe (Lugaresi et al., 2019) and OpenSMILE (Eyben et al., 2010), providing additional signals beyond raw video and audio inputs. The result is a diverse large-scale multimodal benchmark of psychological and social behavior data – 101,964 unified and standardized samples spanning 35046 videos, 10287 audio clips, 25385 transcripts, augmented with behavioral descriptors.

Leveraging HUMAN BEHAVIOR ATLAS, we train OMNISAPIENS-7B SFT, which applies supervised fine-tuning; OMNISAPIENS-7B BAM, which integrates behavioral descriptors into the SFT model through a novel residual-style Behavioral Adapter Module (BAM); and OMNISAPIENS-7B RL, which applies GRPO-style reinforcement learning (Shao et al., 2024). Our results show that (1) training on the diverse behavioral datasets in HUMAN BEHAVIOR ATLAS enables the three OMNISAPIENS-7B variants to consistently outperform existing multimodal LLMs across various behavioral tasks, with each variant demonstrating distinct advantages and trade-offs; (2) pretraining on HUMAN BEHAVIOR ATLAS can meaningfully improve transfer learning to novel behavioral datasets and tasks; (3) while a single foundation model can be broadly effective, behavioral descriptors can be compatibly integrated into existing LLM architectures through BAM, enhancing performance on targeted behavioral tasks.

**Research Contributions.** 1. We develop and release HUMAN BEHAVIOR ATLAS, one of the first large-scale multimodal benchmarks to support the development of foundation models for general understanding of psychological and social behaviors. 2. To support future work, we conduct experiments and analyses with three variants of OMNISAPIENS-7B: SFT, BAM, and RL. 3. In developing this benchmark, we establish foundational practices for constructing human behavior atlases, from general domains like psychological and social to specialized areas such as autism. These include creating a taxonomy of behavioral phenomena, standardizing datasets, defining unified evaluation metrics, and assessing models on the benchmark to guide future research. Together, these practices provide a foundation for building large-scale, multimodal resources that support foundation behavioral models.

## 2 RELATED WORK

**Psychological and social behavioral analysis** is a field that develops systems to model human behaviors across various dimensions (Picard, 2000; Liu et al., 2024). In *affective states*, methods include facial action unit detection (Ekman and Rosenberg, 1997), valence–arousal modeling (Valenza et al., 2011), and explainable emotion detection (Cambria et al., 2024). Work on *cognitive states* has examined mental attention (Kaushik et al., 2022), cognitive workload (Gerjets et al., 2014), and communicative intent (Li et al., 2023a). In *pathology*, efforts include depression (Valstar et al., 2017), anxiety (Zhang et al., 2025), suicidal ideation detection (Ji et al., 2022), and PTSD detection (Sawadogo et al., 2024). Finally, research on *social processes* has addressed humor (Hasan et al., 2019), non-verbal communication (Li et al., 2025a), and sarcasm detection (Castro et al., 2019). Yet, despite the progress, there have been limited efforts to develop a unified system capable of understanding the aforementioned dimensions of *affective states, cognitive states, pathology*, and *social processes* in a unified manner.

**Multimodal multitask pretraining** involves training on multiple modalities and tasks to build general-purpose models (Liang et al., 2024c). PaLI (Chen et al., 2022) shows that scaling across languages and multimodal tasks yields strong generalization, BLIP (Li et al., 2022; 2023b) illustrates

Table 1: Comparison with other large behavioral datasets. Raw = Raw videos, audios, text; BD = Behavioral Descriptors; Aud = Audio; Vis = Vision; Txt = Text. All other acronyms for dimensions and tasks follow Sec. 3.1. HUMAN BEHAVIOR ATLAS provides authentic human recordings that can often reflect the richness and variability of real-world human behavior more faithfully (Reddy et al., 2023).

| Benchmark | Samples | Purpose | Format | | Modalities | | | Dimensions | | | | Behavioral Tasks | | | | | | | | | |
|---|---|---|---|---|---|---|---|---|---|---|---|---|---|---|---|---|---|---|---|---|---|
| | | | Raw | BD | Aud | Vis | Txt | Aff | Cog | Path | Soc | SEN | EMO | SOC | INT | NVC | HUM | SAR | ANX | DEP | PTSD |
| EAV (Lee et al., 2024) | 8.4k Real | Conversational Emotion Recognition | ✓ | ✗ | ✓ | ✓ | ✗ | ✗ | ✗ | ✗ | ✗ | ✗ | ✓ | ✗ | ✗ | ✗ | ✗ | ✗ | ✗ | ✗ | ✗ |
| eMotions (Wu et al., 2025) | 27k Real | Short-video emotion analysis | ✓ | ✗ | ✓ | ✓ | ✗ | ✗ | ✗ | ✗ | ✗ | ✗ | ✓ | ✗ | ✗ | ✗ | ✗ | ✗ | ✗ | ✗ | ✗ |
| MMHU (Li et al., 2025b) | 57k Real | Human driving understanding | ✓ | ✗ | ✗ | ✓ | ✓ | ✗ | ✗ | ✗ | ✗ | ✗ | ✗ | ✗ | ✓ | ✓ | ✗ | ✗ | ✗ | ✗ | ✗ |
| EMO-SUPERB (Shon et al., 2024) | 85k Synthetic | Speech emotion recognition | ✓ | ✗ | ✓ | ✗ | ✓ | ✓ | ✗ | ✗ | ✗ | ✗ | ✓ | ✗ | ✗ | ✗ | ✗ | ✗ | ✗ | ✗ | ✗ |
| HumanOmni (Zhao et al., 2025) | 2400k Synthetic | Human scene understanding | ✓ | ✗ | ✓ | ✓ | ✓ | ✓ | ✗ | ✗ | ✓ | ✓ | ✓ | ✗ | ✗ | ✓ | ✗ | ✗ | ✗ | ✗ | ✗ |
| **Human Behavior Atlas (Ours)** | 101k Real | Psychological and social behavior understanding | ✓ | ✓ | ✓ | ✓ | ✓ | ✓ | ✓ | ✓ | ✓ | ✓ | ✓ | ✓ | ✓ | ✓ | ✓ | ✓ | ✓ | ✓ | ✓ |

how lightweight multimodal modules in pretraining can enhance understanding and generalization, and Kosmos (Huang et al., 2023; Peng et al., 2023) highlights the value of multimodal instruction following across diverse tasks for reasoning. Together, they demonstrate that large-scale pretraining is a promising approach for robust and generalizable systems. However, the capabilities and limitations of pretraining in the domain of psychological and social behaviors remain largely underexplored.

# 3 HUMAN BEHAVIOR ATLAS

To curate HUMAN BEHAVIOR ATLAS, we begin by defining a broad behavioral taxonomy, before collecting 13 publicly available multimodal datasets aligned with the taxonomy's dimensions. To facilitate the training of a foundation model, we reformulate all samples in standardized prompt-target format, while introducing a unified evaluation framework to ensure consistent performance comparability across datasets. Finally, we extract behavioral descriptors to enrich the benchmark. We summarize HUMAN BEHAVIOR ATLAS, its curation process and statistics with Fig. 1.

## 3.1 BEHAVIORAL TAXONOMY AND DATASET COLLECTION

**Psychological and Social Behavior Taxonomy.** To capture behaviors that express internal psychological states and encompass external social processes, we define four overarching behavioral dimensions. **(i) Affective states** *(Aff)* encompass emotions and moods – i.e., short-term feelings (anger, joy) to longer-lasting sentiments. **(ii) Cognitive states** *(Cog)* are internal mental processes – i.e., attention, reasoning, surprise, decision-making; which are often inferred through external cues. **(iii) Pathology** *(Path)* covers psychological and psychiatric conditions – i.e., depression, anxiety; where internal states are assessed via verbal or nonverbal indicators. Finally, **(iv) social processes** *(Soc)* describe social interaction and communicative behaviors – i.e., humour, intent, and cooperation.

Table 2: Datasets in HUMAN BEHAVIOR ATLAS. CLS = Classification (evaluated by direct label matching). TXTR = Text-Response (evaluated by an LLM judge).

| Dataset | Dimension | Task(s) | Task Type | Modalities | Samples | Eval. Metric |
|---|---|---|---|---|---|---|
| CMU-MOSEI | Aff; Cog | EMO, SEN | CLS | T / A / V | 31,454 | Binary weighted F1 (SEN), Mean weighted acc. (EMO) |
| MELD | Aff; Soc; Cog | EMO, SEN | CLS | T / A / V | 27,412 | Binary weighted F1 (SEN), Mean weighted acc. (EMO) |
| TESS | Aff; Cog | EMO | CLS | T / A / – | 2,800 | Mean weighted accuracy |
| CREMA–D | Aff | EMO | CLS | T / A / – | 7,442 | Mean weighted accuracy |
| CH–SIMSv2 | Aff | SEN | CLS | T / A / V | 4,403 | Binary weighted F1 |
| Social-IQ 2.0 | Soc; Cog | SOC | TXTR | T / A / V | 6,437 | Accuracy (LLM–Judge) |
| IntentQA | Soc; Cog | INT | TXTR | T / A / V | 16,297 | Accuracy (LLM–Judge) |
| MimeQA | Soc | NVC | TXTR | T / A / V | 806 | Accuracy (LLM–Judge) |
| UR–FUNNYv2 | Soc | HUM | CLS | T / A / V | 2,125 | Weighted F1 |
| MUStARD | Soc | SAR | CLS | T / A / V | 690 | Weighted F1 |
| DAIC–WOZ | Path | DEP | CLS | T / A / – | 189 | Weighted F1 |
| MMPsy | Path | DEP, ANX | CLS | T / – / – | 1,275 | Weighted F1 |
| PTSD–in–the–Wild | Path | PTSD | CLS | T / A / V | 634 | Weighted F1 |

are internal mental processes – i.e., attention, reasoning, surprise, decision-making; which are often inferred through external cues. **(iii) Pathology** *(Path)* covers psychological and psychiatric conditions – i.e., depression, anxiety; where internal states are assessed via verbal or nonverbal indicators. Finally, **(iv) social processes** *(Soc)* describe social interaction and communicative behaviors – i.e., humour, intent, and cooperation.

**Collecting Tasks and Datasets.** The defined behavioral dimensions map onto established tasks in the literature, providing the basis for collecting a diverse set of datasets. In practice, one task may align with multiple dimensions – e.g., emotion recognition can involve understanding affect (sadness) and cognition (surprise). Our collated tasks include *sentiment polarity (SEN)*: classifying attitudes as

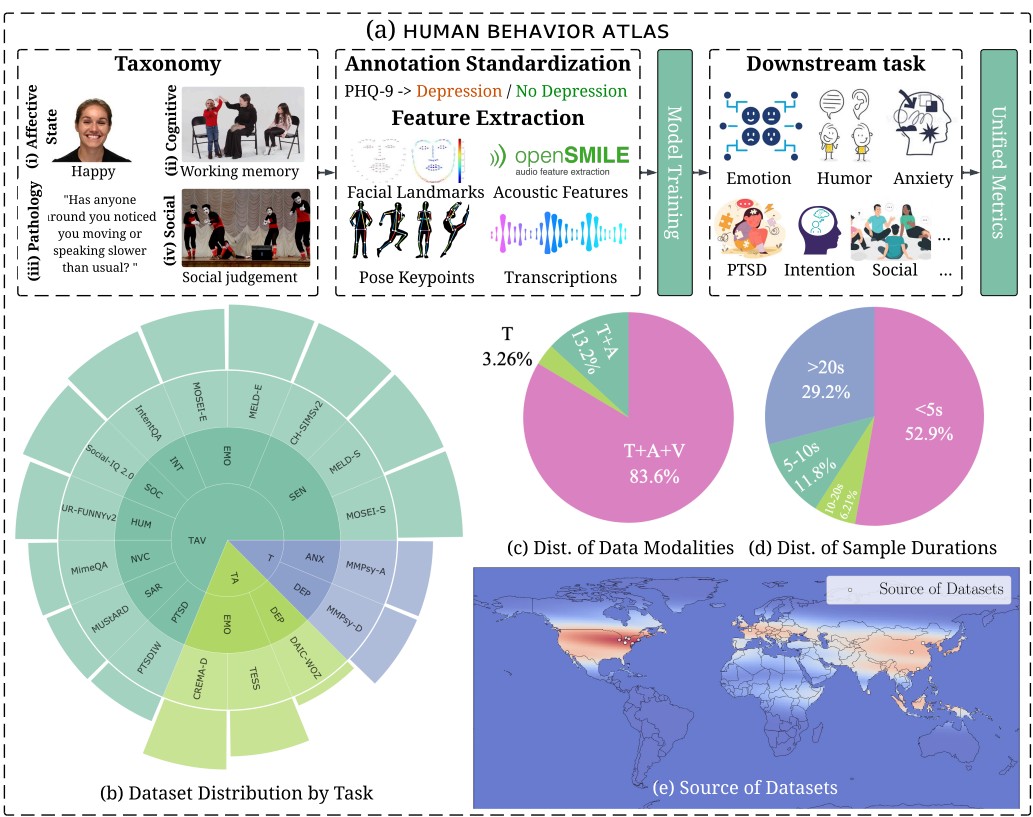

Figure 1: **Overview of HUMAN BEHAVIOR ATLAS.** (a) Selection criteria and preprocessing pipeline of datasets. (b) Dataset distribution across 10 behavior related tasks. Inner circle indicates the modality combination of the input data, where T=Text, A=Audio and V=Video. Middle ring describes the tasks of the dataset, as defined in Sec. 3.1. The outer ring and bars lists the datasets and its sample sizes respectively. (c) Distribution of data modalities. Our dataset has a focus on video understanding as it comprises both vision and audio modalities, with 83.6% of samples containing video data. (d) Distribution of sample durations. Both short and long videos/audio tasks are covered, with 29.2% of video/audio clips lasting more than 20 seconds. (e) Source of datasets. Datasets are sourced from diverse geographic regions across North America, Europe and Asia.

positive, negative, or neutral; *emotion recognition (EMO)*: identifying emotions (anger, joy, sadness); *social reasoning (SOC)*: understanding socially grounded judgments like empathy or appropriateness; *intent recognition (INT)*: identifying the underlying purpose behind a behavior; and *non-verbal communication (NVC)*: interpreting gestures and facial expressions. They also cover *humor detection (HUM)*, *sarcasm detection (SAR)*, *anxiety detection (ANX)*, *depression detection (DEP)*, and *PTSD detection (PTSD)*. Table 2 summarizes all datasets, tasks, and their behavioral dimensions. App. A describes the datasets and our rationale for their behavioral dimension categorizations.

## 3.2 DATASET SCHEMA UNIFICATION

**Standardizing Samples for Foundation Model Training**. The collected behavioral datasets in Table 2 comprise samples that involve varying formats, often requiring task-specific input processing and output heads. This makes it difficult to build a foundation model which requires a standardized training interface. To resolve this heterogeneity, every benchmark sample is reformulated into a standardized prompt–target format (Table 3), designed to align with recent multimodal LLMs, that currently underpin foundation model development, e.g., Qwen2.5-Omni (Xu et al., 2025a). We handcraft and

Table 3: Example of converting a MELD sample into a unified prompt–target format. More examples are in App. A, Table 22.

| Dataset / Task | Original Inputs / Outputs | Reformulated Prompt | Target |
|---|---|---|---|
| MELD (SEN) | Video + Audio + Transcript / Sentiment classification (positive, negative, neutral) | `<video>` `<audio>` Oh. That's so Monica can keep track. That way if one on them is missing, she can be like, 'Where's number 27?!' The above is a video and audio recording from a conversation, along with the transcript. What is the sentiment of the speaker in this recording? Choose from the following sentiment labels: Positive, Negative, Neutral. | **Discrete label set:** {Positive, Negative, Neutral} 

 **Free-text response:** Answer: Positive / Negative / Neutral |

create new prompts that explicitly reference the available modalities and includes textual transcripts where available, while the target is standardized as free-text answers or a fixed discrete label set. To elaborate on standardizing the target format, continuous outputs (e.g., PHQ-9 scores in MMPsy) are discretized into categories based on their original paper guidelines, whereas categorical outputs (e.g., sentiment in MELD) retain their original form. Through this step, all outputs are cast into discrete categories, which can then be represented as free-text answers or elements of a discrete label set. Conversely, tasks requiring open-ended inference via text generation (e.g., MimeQA) have their targets preserved as free-text responses. Depending on the model design, HUMAN BEHAVIOR ATLAS provides the choice of representing the target as free-text or as a discrete label set. For designs that rely solely on a decoder head, the target is provided in free-text form. Alternatively, for designs that incorporate classifier heads, the target is instead provided as a discrete label set. This enhances versatility and enables a broader range of models to be developed, acknowledging that classifier heads may offer more stability for classification tasks (Edwards and Camacho-Collados, 2024).

### 3.3 STANDARDIZED EVALUATION FRAMEWORK

**Unifying Evaluation Metrics Across Datasets.** Integrating diverse datasets is further complicated by how dataset evaluation metrics differ, even when tasks overlap (i.e., emotion recognition in CMU-MOSEI (Zadeh et al., 2018) uses weighted accuracy, while MELD (Poria et al., 2019) uses weighted F1). To enable explicit performance comparisons, we introduce a standardized evaluation framework (Table 2). We assign each task a single standardized metric and apply it uniformly across all datasets associated with that task. For sentiment polarity (*SEN*), we use binary weighted F1, computed over positive and negative labels, since sentiment datasets often differ in class schemes (e.g., three- or five-point scales, inclusion or exclusion of neutral (Poria et al., 2019; Yu et al., 2022)), and prior work highlights positive and negative sentiment as the most informative and foundational (Socher et al., 2013). For emotion recognition (*EMO*), where datasets and emotion categorization models often contain multiple often-imbalanced emotion classes (e.g., Ekman's model has 4 out of 6 negative emotion categories), we adopt the mean of per-class weighted accuracies (Zadeh et al., 2018) to ensure balanced evaluation. Another important refinement for emotion recognition involves consolidating labels that represent the same underlying emotion (e.g., joining 'joy' and 'happiness' labels) while differentiating those that are intrinsically distinct (e.g., disjoining 'surprise' into 'positive surprise' and 'negative surprise'). For other classification tasks (*HUM*, *SAR*, *ANX*, *DEP*, *PTSD*), we use weighted F1, which is well suited for binary detection tasks and mitigates the effects of label imbalance in their corresponding datasets. Finally, for free-text response tasks (*SOC*, *INT*, *NVC*), we use an LLM-judge to evaluate whether the generated text output aligns with the ground truth, reporting accuracy as the proportion of responses judged TRUE. Full details on evalaution metric formulas, LLM judging prompts can be found in App. A

### 3.4 BEHAVIORAL DESCRIPTORS EXTRACTION

**Behavioral Descriptors.** To augment HUMAN BEHAVIOR ATLAS, we extract behavioral descriptors that capture fine-grained affective and social signals. These descriptors may not be explicitly present in raw video and audio, yet they are often critical for understanding psychological and social behaviors, encoding rich cues about prosody, facial expressions, and body language (Zhang et al., 2022; Narayanan and Georgiou, 2013). For the visual modality, we use MediaPipe (Lugaresi et al., 2019) to extract facial landmarks and body pose keypoints, capturing muscle movements, expressions, posture, and gestures relevant to emotion and non-verbal communication. For the audio modality, we extract acoustic features with OpenSMILE (Eyben et al., 2010), using the ComParE 2016 configuration (Schuller et al., 2016) to capture prosodic cues (pitch, energy, spectral properties, and voice quality). Finally, where ground truth transcriptions are missing, we extract text transcriptions from audio content using Whisper v3 Large model (Radford et al., 2023).

### 3.5 OMNISAPIENS-7B SFT

We train a multimodal foundation model for psychological and social behaviors, OMNISAPIENS-7B SFT on HUMAN BEHAVIOR ATLAS. Full model details can be found in App. B.

**Backbone & Output Heads.** We initialize a pretrained multimodal LLM (Qwen2.5-Omni-7B), which includes an audio and image encoder, alongside an LLM. We project raw audio, visual, and textual inputs into a shared embedding space, enabling the LLM backbone to process multimodal benchmark samples. We take the pooled hidden representations at $h_{\text{penult}}$, the penultimate layer

immediately before the final hidden states, as input to task-specific classifier heads, which produce categorical predictions (i.e., positive/ negative sentiment) for classification tasks. For tasks requiring open-ended inference via free-text responses, a separate decoder head generates outputs directly from the final hidden states after $h_{\text{penult}}$. Consistent with the standardized prompt-target format (Sec. 3.2), the classifier heads operate on targets represented as discrete label sets, while the decoder head leverages free-text responses. We use cross-entropy loss and teacher forcing for the classifier and decoder respectively.

## 3.6 OmniSapiens-7B BAM

We introduce a simple variant, OmniSapiens-7B BAM, by equipping OmniSapiens-7B SFT with a Behavioral Adapter Module (BAM) that incorporates behavioral descriptors, allowing us to examine the performance impact of integrating the behavioral descriptors (i.e., pose, prosody, and facial landmarks) into the LLM backbone. Behavioral analysis literature has sometimes framed end-to-end models trained on raw signals and feature-based approaches trained on the descriptors as *mutually exclusive* choices (Li and Mahmoud, 2025; Luo et al., 2025). We investigate if the approaches can interoperate non-invasively, allowing descriptors to augment end-to-end models without compromising backbone representations, which could otherwise cause destabilized features or catastrophic forgetting (Ramasesh et al., 2021).

**Post-Pretraining Residual Adapter.** After full training, OmniSapiens-7B SFT is frozen and an adapter integrating behavioral descriptors is attached; with only the adapter and the classifier/decoder heads receiving gradient updates. Formally, we describe the time-dependent descriptors:

$$\mathbf{f}_{\text{raw}} \in \mathbb{R}^{T \times D_{\text{raw}}},$$

where $T$ is the temporal dimension and $D_{\text{raw}}$ the raw feature dimensionality (i.e., concatenated OpenSmile and/or MediaPipe streams). A temporal pooling operator computes both the mean and standard deviation across timestamps in a single audio or video clip:

$$\mu_f = \frac{1}{T}\sum_{t=1}^{T} \mathbf{f}_{\text{raw}}[t], \qquad \sigma_f = \sqrt{\frac{1}{T}\sum_{t=1}^{T}(\mathbf{f}_{\text{raw}}[t] - \mu_f)^2}, \quad \mathbf{f} = [\,\mu_f,\,\sigma_f\,] \in \mathbb{R}^{D_{\text{feat}}}, \quad D_{\text{feat}} = 2D_{\text{raw}},$$

The pooled behavioral descriptors $\mathbf{f}$ are normalized and regularized with dropout before being passed through a small feed-forward network:

$$x_f = \text{Dropout}\big(\text{Norm}(\mathbf{f})\big), \qquad z_f = \phi\big(W_2\,\phi(W_1 x_f + b_1) + b_2\big).$$

Output $z_f$ produces a residual update to the penultimate hidden state $h_{\text{penult}}$ (defined in Sec. 3.5), scaled by a learnable scalar $\alpha$; the adapted representation $h_{adapt}$ is fed into respective heads:

$$\Delta h_f = \alpha \cdot z_f, \qquad h_{\text{adapt}} = h_{\text{penult}} + \Delta h_f.$$

Because the update follows a residual design – preserving $h_{\text{penult}}$ and adding a learned adjustment, it supplements rather than overwrites the original hidden state. Removing BAM reduces $h_{\text{adapt}}$ back to $h_{\text{penult}}$, ensuring the backbone remains unaffected. We utilize a lightweight feed-forward network to enable computationally efficient adaptation of the multimodal LLM using behavioral features (see computational study in App. D.6). The temporal descriptors are pooled via mean and standard deviation to capture the distributional characteristics of the behavioral signals within each video or audio clip, and to maintain compatibility with the static text, audio, and video modality streams that are taken in by the multimodal LLM. We integrate the adapter at the penultimate layer rather than at the early fusion stage, as the early textualization and alignment of behavioral features would require substantially more training cost (Xu et al., 2025b), which is contrary to the intended plug-and-play nature of the adapter design.

## 3.7 OmniSapiens-7B RL

OmniSapiens-7B RL reuses the same multimodal backbone as OmniSapiens-7B SFT, but handles all tasks within a single decoder head. OmniSapiens-7B RL does not make use of any classifier heads; all outputs are produced directly from a single generative decoder. This preserves the full flexibility of the LLM decoding architecture while enabling a clean comparison against the hybrid classifier-head-plus-decoder setup used in OmniSapiens-7B SFT. We leverage the benchmark samples that have been standardized into a unified prompt–target format (Sec. 3.2), with all targets represented in free-text form. We then optimize OmniSapiens-7B RL with Group Relative Policy Optimization (GRPO) (Shao et al., 2024). Details of the training process and reward structures are in App. B.2.

Table 4: Results grouped by behavioral tasks (headers) and relevant datasets (sub-headers). Best results are bolded, second best are underlined. Following the unified metrics (Sec. 3.3), we use binary weighted F1 for SEN; mean per-class weighted accuracy for EMO; weighted F1 for HUM, SAR, ANX, DEP, PTSD; and LLM-Judge accuracy for SOC, INT, NVC. *MMPSY uses text-only input and excludes BAM; as the backbone is preserved, results are equivalent to OMNISAPIENS-7B SFT.

| Model | EMO | | | | HUM | INT | PTSD | ANX | DEP | | | SEN | | | SAR | SOC | NVC |
|---|---|---|---|---|---|---|---|---|---|---|---|---|---|---|---|---|---|
| | CREMA-D | MELD (E) | MOSEI (E) | TESS | UR-FUNNY | IntentQA | PTSD-WILD | MMPSY (A) | MMPSY (D) | DAIC-WOZ | MELD (S) | CH-SIMSv2 | MOSEI (S) | MUStARD | Social-IQ 2.0 | MimeQA |
| Gemma-3-4B | .495 | .642 | .565 | .499 | .597 | .227 | .499 | .601 | .788 | .137 | **.785** | .813 | .617 | .529 | .191 | .023 |
| HumanOmniV2-7B | **.560** | .633 | .558 | .637 | .638 | .263 | .824 | .527 | .672 | .636 | .768 | .825 | .633 | .395 | .282 | .093 |
| Qwen 2.5-Omni-7B | .521 | .661 | .580 | .568 | .543 | .254 | .760 | .793 | .791 | .636 | .700 | .714 | .602 | .656 | .254 | .069 |
| Qwen-2.5-VL-7B | .501 | .571 | .592 | .499 | .583 | .249 | .755 | .631 | .653 | .623 | .674 | .524 | .317 | .511 | .231 | .098 |
| OMNISAPIENS-7B RL | .501 | .699 | .581 | .510 | .639 | **.486** | .968 | **.919** | .814 | .729 | .571 | .393 | .224 | .647 | **.304** | .133 |
| OMNISAPIENS-7B SFT | .542 | .709 | **.614** | .658 | .532 | .256 | **1.00** | .909 | **.839** | .626 | .746 | .813 | .744 | .624 | .257 | .121 |
| OMNISAPIENS-7B BAM | .548 | **.711** | .607 | **.715** | **.644** | .177 | **1.00** | .909* | **.839*** | .738 | .744 | **.837** | **.775** | **.795** | .201 | **.162** |

# 4 EXPERIMENTS

To offer insights for future research on HUMAN BEHAVIOR ATLAS, we assess the three OMNISAPIENS-7B models trained on the benchmark against existing multimodal LLMs pretrained on general multimodal and human-related data, with the latter comparison meant to contrast HUMAN BEHAVIOR ATLAS against other large scale human-data related benchmarks (i.e. HumanOmni). Full experimental details are in App. C.

## 4.1 MULTI-TASK TRAINING

In this section, we study the general understanding of psychological and social behaviors gained through pretraining on HUMAN BEHAVIOR ATLAS. Namely, we pretrain OMNISAPIENS-7B SFT, OMNISAPIENS-7B BAM and OMNISAPIENS-7B RL in a multi-task setup, where a single instance of the model is trained jointly across all behavioral tasks in the benchmark.

**General multimodal LLMs exhibit limited capabilties across the breadth of behavioral tasks.** Conversely, we observe that OMNISAPIENS-7B SFT, OMNISAPIENS-7B BAM, and OMNISAPIENS-7B RL, all pretrained on behavioral tasks in HUMAN BEHAVIOR ATLAS, consistently exceed the performance of these general multimodal LLMs (Qwen 2.5-Omni-7B (Xu et al., 2025a), Qwen 2.5-VL-7B (Bai et al., 2025), Gemma-3-4B (Team et al., 2025)). From Figure 2, OMNISAPIENS-7B SFT and OMNISAPIENS-7B BAM each outperform the general multimodal LLMs on 8 of 10 behavioral tasks, while OMNISAPIENS-7B RL does so on 7 of 10. This demonstrates that a comprehensive understanding of psychological and social behaviors is better supported by specialized adaptation across behavioral tasks. Specifically, pretraining on HUMAN BEHAVIOR ATLAS can improve the capabilities of general multimodal LLMs on diverse behavioral tasks.

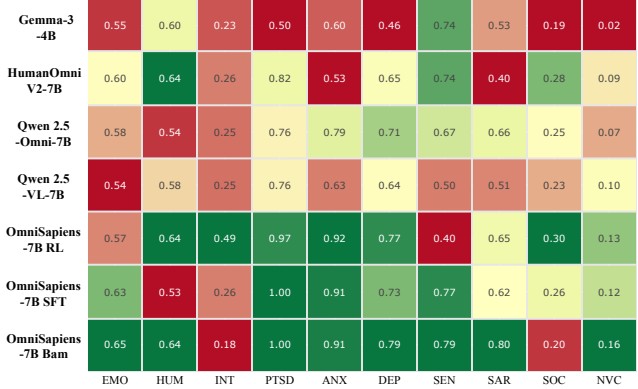

Figure 2: Multitask results across tasks for each model. Each result reports the average score across all datasets for that task. Best to worst = dark green → yellow → dark red. Upon training on HUMAN BEHAVIOR ATLAS, OMNISAPIENS-7B SFT & RL outperform existing pretrained models across most behavioral tasks.

**Existing efforts to develop multimodal LLMs trained on large-scale human-related data remain few, and tend to focus on understanding narrow range of behavioral phenomena.** One such example, HumanOmniV2-7B, primarily targets human-centric scenes involving emotion recognition and facial description (Zhao et al., 2025). While this specialized focus allows it to achieve competitive

scores on tasks such as HUM, SEN and SOC, its performance does not generalize broadly. In contrast, our variants consistently outperform HumanOmniV2-7B across the spectrum of behavioral tasks. OMNISAPIENS-7B SFT leads on 7 of 10 tasks, while both OMNISAPIENS-7B BAM and OMNISAPIENS-7B RL lead on 8 of 10. This highlights that developing a foundation model for comprehensive understanding of psychological and social behaviors requires more than unstructured exposure to general human-related content – it demands systematic and deliberate adaptation across a diverse set of tasks pertaining to these behavioral dimensions.

**While a foundation model improves performance across behavioral tasks, the best-performing variant depends on task type.** For structured classification tasks (EMO, HUM, PTSD, ANX, DEP, SEN, SAR), both OMNISAPIENS-7B BAM and OMNISAPIENS-7B SFT emerge as the strongest performers, each surpassing existing multimodal LLM baselines. The only exceptions are SAR and HUM, where OMNISAPIENS-7B SFT shows relatively weaker performance. In contrast, OMNISAPIENS-7B RL demonstrates the most consistent strong performance on open-ended text generation tasks, achieving the strongest results on INT (0.49) and SOC (0.30), and the second-best on NVC (0.13). Taken together, these results highlight a contrast between supervised fine-tuning approaches, which dominate on structured classification benchmarks, and reinforcement learning, which is better suited to tasks requiring free-form reasoning and generation. Future work can focus on exploring hybrid training strategies, integrating RL's strengths in reasoning with SFT reliability on structured recognition tasks, to achieve balanced performance across the full spectrum of tasks.

## 4.2 TRANSFER LEARNING TO HELDOUT DATASETS

Besides multi-task training, we study the transfer capabilities enabled by pretraining on HUMAN BEHAVIOR ATLAS. We withhold four datasets during multi-task pretraining of OMNISAPIENS-7B SFT. These held-out datasets fall under two settings: (i) novel datasets for tasks already represented in pretraining – i.e. MOSEI as a heldout sentiment dataset, with sentiment still represented in the pretraining tasks by CH-SIMSv2, and (ii) a novel dataset introducing a new behavioral task not seen in pretraining. The pretrained OMNISAPIENS-7B SFT is then fine-tuned separately on each held-out dataset under a fixed, minimal epoch budget. For comparison, Qwen 2.5-Omni-7B SFT – which shares the same architecture as OmniSapiens-7B SFT, differing only in its absence of pretraining on HUMAN BEHAVIOR ATLAS, is also fine-tuned on these held-out datasets under the same training conditions. This enables a clean isolation of the effect of training on our benchmark, ensuring that observed improvements can be attributed to the data and tasks within Human Behavior Atlas rather than to architectural changes.

**Pretraining on HUMAN BEHAVIOR ATLAS enables OMNISAPIENS-7B SFT to adapt more effectively when transferring to new datasets that involve overlapping behavioral tasks (setting i).** From Table 5, the pretrained OMNISAPIENS-7B SFT consistently outperforms Qwen 2.5-Omni-7B SFT on held-out datasets – MOSEI (SEN) +18.3% (+0.112), MELD (EMO) +3.95% (+0.027), and DAIC-WOZ (DEP) +29.4% (+0.17). This highlights that pretraining on diverse behavioral tasks in HUMAN BEHAVIOR ATLAS can yield transfer

Table 5: Transfer to held-out datasets after minimal epoch fine-tuning (1 epoch). Bold denotes best score. DAIC-WOZ[*] involves 2 epochs as it has only 107 training samples. MUStARD[†] presents a novel behavioral task of SAR (sarcasm detection). ‡ Other held-out datasets that have tasks represented during pretraining.

| Dataset | OMNISAPIENS-7B SFT | Qwen 2.5-Omni-7B SFT |
|---|---|---|
| MOSEI[‡] (SEN) | **0.724** | 0.612 |
| MELD[‡] (EMO) | **0.711** | 0.684 |
| DAIC-WOZ[*‡] (DEP) | **0.749** | 0.579 |
| MUStARD[†] (SAR) | **0.658** | 0.473 |

benefits to unseen datasets, provided that these datasets comprise tasks represented in pretraining. Therefore, pretraining on HUMAN BEHAVIOR ATLAS may hold promise for transfer to real-world applications where the underlying behavioral task remains consistent but the datasets or domains vary – i.e. sentiment analysis across product reviews or news settings, although real-world validation could offer further insight on this potential.

**Pretraining on HUMAN BEHAVIOR ATLAS enables OMNISAPIENS-7B SFT to adapt more effectively when transferring to a new dataset that involves a novel behavioral task (setting ii).** From Table 5, OMNISAPIENS-7B SFT outperforms Qwen 2.5-Omni-7B on MUStARD, a dataset for sarcasm detection (SAR), by 39.1% (+0.185), even though SAR was excluded from pretraining tasks. This suggests that pretraining may not only benefit transfer to tasks already represented in pretraining, but also provides a foundation for adapting to new behavioral tasks or phenomena. Thus, pretraining on HUMAN BEHAVIOR ATLAS can hold potential for transfer to real-world settings where

new psychological or social behavioral phenomena emerge, which are often understudied and have minimal available data – i.e. evolving patterns of online interaction (Oksanen et al., 2024). Yet, we qualify that the complexity of these behaviors means transfer performance may likely vary across behavioral phenomena. Therefore, understanding which novel behavioral tasks remain challenging to transfer to is a research direction that future work on HUMAN BEHAVIOR ATLAS can explore.

**Beyond the transfer improvements themselves, the comparison of OmniSapiens-7B SFT against the single-task trained Qwen 2.5-Omni-7B SFT, as shown in Table 5., clarifies how the diversity of tasks in Human Behavior Atlas shapes the learned features while training on the benchmark.** On represented tasks (SEN, EMO, DEP), OmniSapiens-7B SFT outperforms the single-task pretrained Qwen model even after minimal fine-tuning, indicating that exposure to multiple behavioral tasks enables the model to internalize shared behavioral features across heterogeneous tasks that generalize across datasets. Meanwhile, the fact that OmniSapiens-7B SFT also surpasses Qwen 2.5-Omni-7B on MUStARD, despite SAR being absent during pretraining, suggests that multi-task behavioral pretraining provides broader, more general behavioral patterns that may occasionally extend to novel behavioral phenomena. These transfer patterns, when contrasted against the single-task pretrained baseline of Qwen 2.5-Omni-7B, suggests that the diverse and heterogeneous behavioral tasks in Human Behavior Atlas contribute complementary and synergistic behavioral features, rather than conflicting or contradictory supervision.

**Qualitative analysis suggests that pretraining on behavioral tasks in HUMAN BEHAVIOR ATLAS yields stronger pragmatic recognition capabilities.** An interesting pattern observed in the held-out task of SAR is that OMNISAPIENS-7B SFT shows a greater ability to recognize pragmatic cues (i.e. contextual signals beyond literal word meaning) after extensive pretraining on behavioral datasets in HUMAN BEHAVIOR ATLAS. An example (Figure 3) from MUStARD, the heldout sarcasm detection (SAR) dataset, illustrates this. While OMNISAPIENS-7B SFT correctly interprets Chandler's remark about putting up balcony lights as sarcasm rather

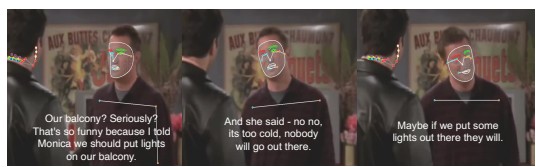

Figure 3: Example from MUStARD where the speaker (Chandler, from *Friends*) sarcastically suggests putting up balcony lights. While Qwen2.5-Omni-7B predicts no sarcasm, OMNISAPIENS-7B SFT correctly identifies the instance as sarcasm.

than a serious suggestion, Qwen2.5-Omni-7B, without the same behavioral pretraining, defaults to a prediction of "no sarcasm" (a literal, face-value interpretation of Chandler's remark), even after a full epoch of fine-tuning. This case is not isolated: across testing, Qwen2.5-Omni-7B tends to default to "no sarcasm" predictions for 93.2% of samples, resulting in a lower weighted recall of 0.534. In contrast, OMNISAPIENS-7B SFT achieves a higher weighted recall of 0.670 by more reliably identifying sarcastic instances, assigning the "sarcasm" label to 30.1% of samples. These results suggest that pretraining on HUMAN BEHAVIOR ATLAS can improve the general capacity to capture the subtle, context-dependent nuances of psychological and social behaviors. Future research could explore whether this capacity is emergent and scales with model size, potentially enabling more reliable detection of behaviors that go beyond surface-level or literal cues.

## 4.3 EFFECT OF BEHAVIORAL DESCRIPTORS

**Behavioral descriptors can provide targeted performance gains for specific behavioral tasks.** We integrate behavioral descriptors through OMNISAPIENS-7B BAM, an extension of OMNISAPIENS-7B SFT, that applies post-training residual updates through a Behavioral Adapter Module (BAM) (Sec. 3.6), while leaving the backbone representations unchanged. From Table 6, OMNISAPIENS-7B BAM shows performance improvements over OMNISAPIENS-7B SFT on a considerable number of tasks – NVC (+33.00%), SAR (+29.00%), HUM (+21.00%), DEP (+8.21%), EMO (+3.17%) and SEN (+2.60%), but the

Table 6: Δ highlights the change in performance from OMNISAPIENS-7B SFT to OMNISAPIENS-7B BAM, shown as percentage (%) and absolute (Abs). BAM provides notable performance gains for a considerable number of behavioral tasks, although its benefits are not consistent across all tasks.

| Task | OMNISAPIENS-7B SFT | OMNISAPIENS-7B BAM | Δ % (Abs) |
|------|------|------|------|
| NVC | 0.12 | 0.16 | +33.00 (+0.04) |
| SAR | 0.62 | 0.80 | +29.00 (+0.18) |
| HUM | 0.53 | 0.64 | +21.00 (+0.11) |
| DEP | 0.73 | 0.79 | +8.21 (+0.06) |
| EMO | 0.63 | 0.65 | +3.17 (+0.02) |
| SEN | 0.77 | 0.79 | +2.60 (+0.02) |
| PTSD | 1.00 | 1.00 | 0.00 (+0.00) |
| ANX | 0.91 | 0.91 | 0.00 (+0.00) |
| SOC | 0.26 | 0.20 | -23.08 (-0.06) |
| INT | 0.26 | 0.18 | -30.77 (-0.08) |

same gains do not hold for PTSD, ANX, SOC and INT. Yet, because BAM is implemented in a modular, residual fashion, it can be flexibly applied where useful and omitted where not, without altering the backbone representation of the model. In this fashion, behavioral descriptors can be selectively applied to improve the performance of OMNISAPIENS-7B SFT, or foundation models more broadly, on targeted tasks. Therefore, while existing literature may often frame feature-based approaches and end-to-end models as mutually exclusive choices, our results highlight that descriptors can still be useful within end-to-end models – perhaps not as a blanket integration into the shared backbone, since not all behavioral tasks benefit equally, but as a means of targeted adaptation for tasks that benefit most.

**Qualitative analysis suggests that behavioral descriptors can help improve foundation models through the detection of subtle behavioral cues.** In our experiments, we observe cases where subtle cues, such as a fleeting smile in CH-SIMSv2 (Figure 4), go undetected by OMNISAPIENS-7B SFT, which relies solely on raw audio-visual signals, leading to an incorrect prediction of negative sentiment. However, when behavioral descriptors are incorporated in OMNISAPIENS-7B BAM, the subtle smile appears to register, with the prediction changing to positive.

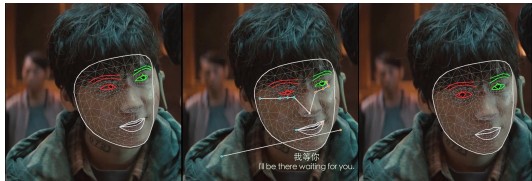

Figure 4: Example from the CH-SIMSv2 dataset where the speaker briefly displays a split-second smile, signaling positive sentiment. While OMNISAPIENS-7B SFT misses the subtle cue and predicts negative sentiment, OMNISAPIENS-7B BAM correctly predicts positive.

This suggests that subtle behavioral cues may be more effectively registered with the additional behavioral descriptors than with raw audio and video alone. A potential hypothesis is that coarser raw video and audio input streams may overshadow fine-grained behavioral expressions, while descriptors such as MediaPipe facial keypoints more explicit representations of these expressions. Yet, while we observe instances consistent with this hypothesis, further research can systematically pinpoint the data distributions and context where behavioral descriptors provide the most benefit for foundation models, thereby guiding their effective integration into such models.

## 5  CONCLUSION

In this work, we introduce HUMAN BEHAVIOR ATLAS and evaluated three model variants, OMNISAPIENS-7B SFT, OMNISAPIENS-7B BAM, OMNISAPIENS-7B RL, alongside other multimodal LLMs. Our experiments demonstrate that training on this benchmark produces significant gains in both multi-task and transfer learning scenarios. Moreover, incorporating behavioral descriptors through BAM enables targeted improvements for specific domains, highlighting the importance of structured behavioral information. At the same time, our findings expose new research questions critical for advancing foundation models capable of comprehensively understanding psychological and social behaviors. To support the research community, we release HUMAN BEHAVIOR ATLAS, and its associated OMNISAPIENS-7B models, aiming to foster investigation into these challenges and contribute to the development of foundation models for psychological and social behavior analysis.

In constructing this benchmark, we propose a set of foundational practices for developing human behavior atlases, designed to span the full spectrum of behavioral domains—from broad areas such as psychological and social behaviors to more specialized domains like autism. These practices establish a methodological framework that includes: (i) deriving a taxonomy that organizes diverse behavioral phenomena into a coherent structure, (ii) standardizing heterogeneous datasets into a consistent format that supports the training of foundation models, (iii) defining robust and comparable evaluation metrics, (iv) refining label spaces by consolidating categories that capture the same underlying concept (e.g., merging semantically overlapping labels) while distinguishing those that are inherently different (e.g., splitting broad or ambiguous categories into finer-grained, meaningful sublabels), and (v) systematically evaluating foundation models on the benchmark to surface limitations and highlight directions for future research.

Together, these practices provide the building blocks for the next generation of large-scale, multimodal resources that enable the development of foundation models with comprehensive capabilities for analyzing human behavior. By offering both methodological guidance and an empirical testbed, this benchmark lays the groundwork for advancing the scientific study and computational modeling of psychological, social, and domain-specific behaviors.

## ACKNOWLEDGMENTS

This research/project is supported by the Asian Institute of Digital Finance, National University of Singapore; Ministry of Education, Singapore under its MOE Academic Research Fund Tier 2 (MOE-T2EP20123-0005: "Neurosymbolic AI for Commonsense-based Question Answering in Multiple Domains"), MOE Tier 1 Award (MOE-T2EP50221-0028: "Discipline-Informed Neural Networks for Interpretable Time-Series Discovery"), and by the RIE2025 Industry Alignment Fund – Industry Collaboration Projects (I2301E0026: "Generative AI"), administered by A*STAR, as well as supported by Alibaba Group and NTU Singapore. The authors would also like to acknowledge Nvidia for their support through the provision of GPU and compute resources.

## ETHICS AND REPRODUCIBILITY STATEMENT

This research focuses on curating a benchmark for training foundation models for psychological and social behavior understanding. Leveraging this benchmark, we also developed a foundation model, and experimented with different methods (SFT, BAM, RL). All datasets, and model architectures utilized have been properly cited, and are publicly available. Additionally, there are no discrimination, bias, or fairness issues that should arise in this paper. Furthermore, the models trained are not expected to generate harmful content. For reproducibility, we provide all experimental details in Section 3 and the corresponding appendices. We will release the benchmark, models and codes to support reproducibility.

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

# A    BENCHMARK DETAILS

In this appendix, we describe the corresponding details of our curated HUMAN BEHAVIOR ATLAS benchmark.

## A.1    DATASET DESCRIPTIONS

Below, we describe the datasets used in our benchmark and provide a concise rationale for their assigned behavioral dimensions.

- **CMU-MOSEI** Zadeh et al. (2018) is one of the largest and most widely used multi-modal datasets for sentiment and emotion analysis, with over 30,000 samples. It provides a diverse set of real-world conversations in which speakers express opinions on various topics. Each video is annotated for sentiment polarity and emotion categories, enabling study on how speech, text, and facial expressions contribute to sentiment and emotional expression. This dataset reflects affective states through emotion and sentiment, and also relates to cognitive states through categories such as surprise, which involve inference of internal mental processes beyond affect.
- **MELD** (Poria et al., 2019) The Multimodal EmotionLines Dataset (MELD) contains a large collection of multi-party conversations drawn from 1,433 dialogues in the TV series *Friends*. Each utterance is annotated with emotion categories (neutral, joy, sadness, anger, surprise, fear, disgust) and sentiment polarity labels (positive, neutral, negative). This dataset reflects affective states through emotions and sentiment, relates to cognitive states through categories such as surprise and the need to reason over conversational context, and captures social processes through multi-party dialogue.
- **UR-FUNNYv2** (Hasan et al., 2019) is a dataset of humor detection from more than 16,000 TED talk videos, with each short video clip annotated for humor indicators from words, gestures, and prosodic cues. This dataset reflects social processes through the recognition of humor in communication.
- **MUStARD** (Castro et al., 2019) is a corpus of 690 videos for research in sarcasm detection from TV shows, such as Friends and the Big Bang Theory. It consists of audio-visual utterances annotated with sarcasm labels, with additional contextual information about the scenarios provided. This dataset reflects social processes through the interpretation of sarcasm in dialogue.
- **DAIC-WOZ** (Valstar et al., 2016) is designed for automatic depression diagnosis. The dataset consists of clinical interviews conducted with the help of a virtual agent (controlled by a human behind the scenes), with nearly 200 samples. Participants were asked mental health-related questions, and their responses were recorded for multi-modal analysis. This dataset reflects pathological dimensions by focusing on clinical depression.
- **CREMA-D** (Cao et al., 2014) is a dataset contains 7,442 audio files recorded by 91 actors (48 males and 43 females). Categorical emotion labels (the six basic emotions; anger, disgust, fear, happy, neutral, sad) for the perceived emotion displayed were collected via crowd-sourcing from 2,443 raters. This dataset reflects affective states by providing categorical and intensity-based emotional expressions.
- **CH-SIMS** v2 (Yu et al., 2022) includes 4,403 samples with sentiment annotations. The dataset was collected from 11 different scenarios designed to simulate real-world human–computer interaction, with annotations based on sentiment scores from strongly negative to strongly positive. This dataset reflects affective states by capturing sentiment polarity and intensity.
- **MMPsy** (Zhang et al., 2025) is a dataset containing 1,275 real student records, including mental health scale scores, therapy interview audio recordings, and transcripts. The data were collected from over 15,000 middle school students in Guangdong Province, China. Annotations were obtained using standard psychological scales such as GAD-7 (Generalized Anxiety Disorder-7) and MHT (Mental Health Test), in collaboration with the local Department of Education. This dataset reflects pathological dimensions through the assessment of depression and anxiety.
- **PTSD in the Wild** (Sawadogo et al., 2024) is a publicly available video database that provides a diverse range of real-world videos of individuals with PTSD symptoms. Videos were annotated as 'With PSTD' or Without PTSD'. This dataset reflects pathological dimensions through the diagnosis of post-traumatic stress disorder.
- **TESS** (Pichora-Fuller and Dupuis, 2020) The Toronto emotional Speech Set consists of 200 target words spoken by two actresses demonstrating seven emotions (anger, disgust, fear, happiness, pleasant surprise, sadness, and neutral), with over 2000 samples. This dataset reflects affective states through explicit emotion portrayals, and also relates to cognitive states in that it includes

categories such as surprise, which involve the inference of internal mental processes distinct from affect.

- **Social-IQ 2** (Wilf et al., 2023) is a dataset containing over 1,000 videos and 6000 samples designed to evaluate AI's social intelligence by testing comprehension of human and behaviors in YouTube videos. This dataset reflects social processes through its focus on understanding interactions, and also relates to cognitive states as it requires inferring internal mental processes such as intent and reasoning from observed behavior.
- **IntentQA** (Li et al., 2023a) is a video question-answering dataset built from the NExT-QA dataset Xiao et al. (2021), comprising over 16000 samples. It contains causal and temporal inference questions that require models to recognize the underlying intent behind human actions in everyday social scenarios. This dataset reflects social processes through its focus on intent in interaction, and also relates to cognitive states insofar as intent recognition involves inferring internal mental processes underlying human actions.
- **MimeQA** Li et al. (2025a) is a video question-answering dataset derived from 8 hours of YouTube mime videos that focuses on nonverbal social reasoning with over 800 samples. It contains human annotated question-answer pairs designed to evaluate AI models' ability to understand and interpret nonverbal gestures and movement without spoken language. This dataset reflects social processes by evaluating understanding of nonverbal communication.

## A.2 DETAILS OF BENCHMARK TRAINING SPLITS

We follow the official training splits of the original datasets whenever they are available. When such splits are unavailable, we create new splits by randomly partitioning the data at the level of separate video and audio clips. This approach ensures that samples from the same recording do not leak across train, validation, and test sets, while maintaining consistency with prior dataset usage. For transparency, the exact training, validation, and test counts for each dataset are reported in Table 7.

Table 7: Train-validation-test breakdown with per-dataset totals and the combined total across all datasets. *Social-IQ 2.0 does not have a public test set, therefore we utilize the validation set in line with existing literature Xie and Park (2023); Chen et al. (2024b) **MimeQA has no validation set.

| Dataset | Train | Val | Test | Total |
|---|---|---|---|---|
| CMU-MOSEI (SEN) | 16326 | 1871 | 4659 | 22856 |
| CMU-MOSEI (EMO) | 6246 | 599 | 1753 | 8598 |
| MELD (SEN) | 9988 | 1108 | 2610 | 13706 |
| MELD (EMO) | 9988 | 1108 | 2610 | 13706 |
| TESS | 1960 | 420 | 420 | 2800 |
| CREMA-D | 5209 | 1116 | 1117 | 7442 |
| CH-SIMSv2 | 2722 | 647 | 1034 | 4403 |
| Social-IQ 2.0* | 5562 | 0 | 875 | 6437 |
| IntentQA | 12119 | 2044 | 2134 | 16297 |
| MimeQA** | 633 | 0 | 173 | 806 |
| UR-FUNNYv2 | 1709 | 192 | 224 | 2125 |
| MUStARD | 483 | 104 | 103 | 690 |
| DAIC-WOZ | 107 | 35 | 47 | 189 |
| MMPsy | 893 | 191 | 191 | 1275 |
| PTSD-in-the-Wild | 506 | 64 | 64 | 634 |
| **Combined Total** | **74451** | **9499** | **18014** | **101964** |

## A.3 GEOGRAPHIC DISTRIBUTIONS OF DATASETS

We provide a breakdown of the geographic distribution of the datasets contained within our benchmark in Table 21

## A.4 REFORMULATED PROMPT−TARGET PROMPTS

In Table 22, we provide further examples of how we reformulated datasets within our benchmark to a standardized prompt-target format that is compatible with foundation model training. To standardize the target format across datasets, we process them as follows:

- **MMPsy**: continuous PHQ-9 scores are discretized into categories following its paper guidelines.
- **IntentQA, MimeQA, Social-IQ 2.0**: targets are inherently open-ended and preserved as free-text responses, since they require inference through text generation.
- **All other datasets** (e.g., MELD, MOSEI, CH-SIMSv2, CREMA-D, TESS, UR-FUNNY, MUStARD, PTSD-in-the-Wild, DAIC-WOZ): contain discrete categorical outputs (e.g., sentiment, emotion, humor, sarcasm) and are retained in their original categorical form.

A.5    DETAILS OF BENCHMARK EVALUATION METRICS

In this section, we provide the full details of the evaluation metrics. We highlight the formulas used to compute the various metrics reported in Table 2.

For tasks HUM (Humour Detection), SAR (Sarcasm Detection), DEP (Depression Detection), ANX (Anxiety Detection), and PTSD (PTSD Detection), we compute the weighted F1 score. The F1 score is defined as:

$$F1 = \frac{2 \cdot \text{Precision} \cdot \text{Recall}}{\text{Precision} + \text{Recall}},$$

where

$$\text{Precision} = \frac{TP}{TP + FP}, \qquad \text{Recall} = \frac{TP}{TP + FN}.$$

The weighted F1 is then computed as:

$$\text{Weighted-F1} = \sum_{c \in C} \frac{n_c}{N} \cdot F1_c,$$

where $n_c$ is the number of true instances in class $c$, $N$ is the total number of instances, and $C$ is the set of classes.

For the SEN (Sentiment Detection) task, we use binary weighted F1, following Yu et al. (2022), which applies the same formula but only over the positive and negative sentiment classes. Since datasets such as MOSEI (Liang et al., 2018) may contain fine-grained sentiment scales (e.g., seven-point labels: strongly negative, negative, weakly negative, neutral, weakly positive, positive, strongly positive), we map all non-neutral predictions and labels into either negative or positive classes.

For the EMO (Emotion Recognition) task, we compute the mean/ average weighted accuracy across all emotion classes (e.g., fear, surprise, joy), using the weighted accuracy formula, following Liang et al. (2018):

$$\text{Weighted-Accuracy} = 0.5 \cdot \frac{TP}{P} + 0.5 \cdot \frac{TN}{N},$$

where $TP$ and $TN$ are the number of true positives and true negatives for the target class, and $P$ and $N$ denote the total number of positive and negative samples, respectively.

For free-text response tasks such as NVC (Non-Verbal Communication), INT (Intent Recognition), and SOC (Social Reasoning), we leverage an LLM judge to grade the generated responses. Specifically, we provide task-specific prompts to the LLM judge and record the proportion of responses marked as TRUE as an estimate of accuracy:

$$\text{Accuracy} = \frac{\text{number of TRUE responses}}{\text{number of total responses}}$$

We utilize the following LLM grading prompts with the GPT-5-nano (OpenAI, 2025) model. Evaluation Protocol for LLM-Judge Scoring. For all QA-style tasks requiring LLM-judge evaluation, we ensure strict isolation between the model under evaluation and the judging model. Each evaluation is performed in a fresh session, with no shared history across queries. The LLM judge receives only the dataset-provided question and the model's generated answer, with no additional system context. All models are presented with identical prompts to maintain consistency and prevent evaluation bias. This protocol guarantees that scoring reflects the intrinsic quality of model responses rather than prompt design or accumulated conversational context.

## LLM Judge Prompts

### System Instructions

**MimeQA**: Carefully consider the following question and answers regarding understanding of a mime performance. You will be shown a "gold-standard" answer from a human annotator, referred to as the "Reference Answer", and a "Candidate Answer". Your task is to determine whether the candidate captures the core meaning of the reference answer using the following criteria:

- The candidate must concisely state one primary answer in its first clause. Option lists without choosing one and deferrals are not allowed.
- The candidate does not contain misleading information and does not hallucinate story plots not present in the reference answer.
- Since the videos are mime performances, invisible actions, objects, or the mime actor portraying objects should be considered correct if and only if they are relevant to the question.
- The candidate answer can be a good answer in place of the reference answer as long as they are in the same ballpark. However, the candidate must not refer to a different subject or object not supported by the question/reference. If the candidate's answer centers on a different primary subject/object than the reference, it is incorrect.

**Social-IQ 2.0**: Carefully consider the following question and answer regarding understanding of a video. You will be shown a "gold-standard" answer from human annotators, referred to as the "Reference Answer", and a "Candidate Answer". Your task is to judge whether the candidate captures the core meaning of the reference answer using the following criteria:

- The candidate must concisely state one primary answer in its first clause. Option lists without choosing one and deferrals are not allowed.
- The candidate's explanation is in the same ballpark as the reference and does not add a claim that conflicts with it. The wording need not be the same; minor additions are allowed if they are consistent with the reference and the question.
- The candidate should not assert a conflicting explanation or introduce factually incompatible details. The candidate must not refer to a different subject or object not supported by the question/reference. If the candidate's answer centers on a different primary subject/object than the reference, it is incorrect.

**IntentQA**: Carefully consider the question and answers about the intent behind actions in a video. You will be shown a "gold-standard" answer from human annotators, referred to as the "Reference Answer", and a "Candidate Answer". Your task is to judge whether the candidate gives a plausible interpretation of the intent that does not contradict the reference, using the following criteria:

- The candidate must concisely state one primary answer in its first clause. Option lists without choosing one and deferrals are not allowed.
- The candidate's explanation is in the same ballpark as the reference and does not add a claim that conflicts with it. The wording need not be the same; minor additions are allowed if they are consistent with the reference and the question.
- The candidate should not assert a conflicting explanation, introduce factually incompatible details, or miss the core intent. The candidate must not refer to a different subject or object not supported by the question/reference. If the candidate's answer centers on a different primary subject/object than the reference, it is incorrect.

### User Instruction

Question: [question]
Candidate Answer: [candidate answer]
Reference Answer: [reference answer]
Please evaluate whether the candidate answer is correct according to the grading instructions.

## B MODEL IMPLEMENTATION DETAILS

In this appendix, we provide further details on the implementation of the models that we trained on our benchmark, OMNISAPIENS-7B SFT and OMNISAPIENS-7B RL, in appendices B.1 and B.2 respectively, while the implementation of OMNISAPIENS-7B BAM is detailed in Section 3.6.

### B.1 ARCHITECTURE AND TRAINING OBJECTIVES OF OMNISAPIENS-7B SFT

OMNISAPIENS-7B processes text, audio, and vision modalities from raw transcripts, audio clips, images, and videos by projecting their respective features into a shared embedding space. In the first stage, audio and vision tokens are projected into the backbone LLM embedding space to align with textual embeddings. This allows the model to perform cross-modal integration, where information from textual semantics, speech, facial expression, and broader scene context are fused to capture complex behavioral dynamics. Formally, let $T$ denote the number of text tokens, and $A$ and $V$ the numbers of audio and visual feature frames, respectively. We obtain modality-specific embeddings as:

$$
\begin{aligned}
z_{\text{text}} &= E_{\text{text}}(w_{1:T}) \in \mathbb{R}^{T \times H}, \\
z_{\text{aud}} &= P_{\text{aud}}(E_{\text{aud}}(f_{1:A}^{\text{aud}})) \in \mathbb{R}^{A \times H}, \\
z_{\text{vis}} &= P_{\text{vis}}(E_{\text{vis}}(f_{1:V}^{\text{vis}})) \in \mathbb{R}^{V \times H},
\end{aligned}
$$

where $w_{1:T}$ are text tokens embedded by $E_{\text{text}}$, and $f_{1:A}^{\text{aud}}$ and $f_{1:V}^{\text{vis}}$ are raw frame-level audio inputs and patch-level visual inputs, which are first encoded by $E_{\text{aud}}$ and $E_{\text{vis}}$ and then projected into the shared hidden space $\mathbb{R}^H$ via $P_{\text{aud}}$ and $P_{\text{vis}}$. This allows audio and vision sequences to be aligned with text in the backbone transformer LLM embedding space.

The fused multimodal sequence:

$$
z = [\, z_{\text{text}};\, z_{\text{aud}};\, z_{\text{vis}} \,] \in \mathbb{R}^{(T+A+V) \times H}
$$

is processed by the transformer LLM backbone $F$ with $L$ layers. We denote the penultimate layer (immediately before the final hidden layer) as:

$$
h_{\text{penult}} = F^{(L-1)}(z) \in \mathbb{R}^{(T+A+V) \times H}.
$$

To obtain a fixed-size representation for classification, we apply masked mean pooling across the sequence dimension:

$$
\tilde{h}_{\text{penult}} = \frac{\sum_{t=1}^{T+A+V} m_t \cdot h_{\text{penult},t}}{\sum_{t=1}^{T+A+V} m_t} \in \mathbb{R}^H,
$$

where $m_t \in \{0, 1\}$ is an attention mask indicating valid tokens.

**Classification head.** Each task in HUMAN BEHAVIOR ATLAS is assigned its own classifier head. For task $t$ with labels $Y_t$, a linear head $C_t : \mathbb{R}^H \to \mathbb{R}^{|Y_t|}$ produces

$$
\ell_t = C_t(\tilde{h}_{\text{penult}}), \qquad \hat{y}_t = \text{softmax}(\ell_t),
$$

where $Y_t$ represents the complete set of unique labels for that task. When multiple datasets map to the same task, they share a single classifier head, with $Y_t$ constructed as the union of the label sets across those datasets. We train with cross-entropy loss, with mini-batch size $B$:

$$
\mathcal{L}_{\text{cls}} = -\frac{1}{B} \sum_{i=1}^{B} \sum_{k=1}^{|Y_t|} \mathbf{1}[y_i = k] \, \log \hat{y}_{i,k}.
$$

**Decoder head.** For open-ended generation, the sequence representation $h_{\text{penult}}$ continues into the language modeling head. A conditional decoder $G$ generates text autoregressively, where $\widetilde{y}_{1:L}$ are target tokens and $V_{\text{out}}$ is the output vocabulary size. At step $\ell$,

$$
o_\ell = G(h_{\text{penult}}, \widetilde{y}_{<\ell}) \in \mathbb{R}^{V_{\text{out}}}, \qquad p(\widetilde{y}_\ell \mid \widetilde{y}_{<\ell}, h_{\text{penult}}) = \text{softmax}(o_\ell).
$$

We train with the following teacher forcing loss:

$$\mathcal{L}_{\text{qa}} = -\frac{1}{B} \sum_{i=1}^{B} \sum_{\ell=1}^{L} \log p\big(\widetilde{y}_{i,\ell} \mid \widetilde{y}_{i,<\ell}, \, h_{\text{penult},i}\big).$$

For multi-task pretraining of OMNISAPIENS-7B SFT, we jointly optimize both objectives, minimizing the overall loss.

$$\mathcal{L} = \mathcal{L}_{\text{cls}} + \mathcal{L}_{\text{qa}}.$$

## B.2 DETAILS OF REINFORCEMENT LEARNING

**Preliminaries.** Group Relative Policy Optimization (GRPO) (Shao et al., 2024) is a reinforcement learning method designed specifically for fine-tuning language models. In standard reinforcement learning for LLMs, the model (policy) generates text responses to prompts, receives rewards based on response quality, and updates its parameters to maximize expected rewards. GRPO distinguishes itself from other methods like Proximal Policy Optimization (PPO) by eliminating the need for a separate value network to estimate advantages, instead computing them directly from groups of sampled responses.

The core insight of GRPO is that for language generation tasks, the relative quality of responses to the same prompt provides sufficient signal for optimization. For a given prompt $q$ at training iteration $t$, GRPO samples multiple responses $\{o_{(q,i,t)}\}_{i=1}^{|G_q|}$ from the current policy, forming a group $G_{(q,t)}$. Each response $o_{(q,i,t)} = (o_{(q,i,t):1}, o_{(q,i,t):2}, \ldots, o_{(q,i,t):n})$ consists of a sequence of $n$ tokens. After generation, each response receives a scalar reward $r_{(q,i,t)}$ from a reward function that evaluates response quality according to task-specific criteria.

Rather than estimating the value function as in actor-critic methods, GRPO computes the advantage—a measure of how much better a response is compared to the expected performance—directly from the reward distribution within each group. The advantage for the $i$-th response is computed as:

$$\hat{A}_{(q,i,t)} = \frac{r_{(q,i,t)} - \hat{\mu}_{G_{(q,t)}}}{\hat{\sigma}_{G_{(q,t)}} + \varepsilon}, \tag{1}$$

where $\hat{\mu}_{G_{(q,t)}}$ and $\hat{\sigma}_{G_{(q,t)}}$ are the empirical mean and standard deviation of rewards in the group, and $\varepsilon$ is a small constant for numerical stability. This normalization ensures that advantages have zero mean and unit variance within each group, which stabilizes training across prompts with different reward scales and enables consistent gradient updates regardless of the absolute reward magnitudes.

The policy optimization follows a trust region approach similar to PPO, updating the model parameters to improve responses while preventing excessive deviation from the current policy. The GRPO objective incorporates the normalized advantages with importance sampling and clipping:

$$\tilde{A}_{(q,i,t):k}(\theta) = \min\big(\varphi_{(q,i,t):k}(\theta) \cdot \hat{A}_{(q,i,t)}, \text{clip}\big(\varphi_{(q,i,t):k}(\theta), 1-\epsilon, 1+\epsilon\big) \cdot \hat{A}_{(q,i,t)}\big), \tag{2}$$

$$\varphi_{(q,i,t):k}(\theta) = \frac{\pi_\theta(o_{(q,i,t):k} \mid q, o_{(q,i,t):<k})}{\pi_{\theta_{\text{old}}}(o_{(q,i,t):k} \mid q, o_{(q,i,t):<k})}, \tag{3}$$

where $\varphi_{(q,i,t):k}(\theta)$ represents the importance sampling ratio between the current policy $\pi_\theta$ and the old policy $\pi_{\theta_{\text{old}}}$ at token position $k$. The clipping operation with parameter $\epsilon$ prevents large policy updates that could destabilize training. The per-token formulation allows the model to learn which specific parts of the response contribute to higher rewards.

The complete GRPO training objective combines the clipped advantage with a KL divergence penalty:

$$J_{\text{GRPO}}(\theta) = \mathbb{E}_{q \sim \mathcal{D}, \{o_{(q,i,t)}\} \sim \pi_{\theta_{\text{old}}}} \left[ \frac{1}{|G_{(q,t)}|} \sum_{i=1}^{|G_{(q,t)}|} \frac{1}{n_{o_{(q,i,t)}}} \sum_{k=1}^{n_{o_{(q,i,t)}}} \tilde{A}_{(q,i,t):k}(\theta) - \beta D_{\text{KL}}\big(\pi_\theta \| \pi_{\text{ref}}\big) \right], \tag{4}$$

where $\mathcal{D}$ is the prompt distribution, $n_{o_{(q,i,t)}}$ is the length of response $i$, and $\beta$ controls the strength of KL regularization against a reference policy $\pi_{\text{ref}}$ (typically the initial supervised fine-tuned model).

This regularization prevents the model from deviating too far from coherent language generation while optimizing for task-specific rewards. The expectation is approximated through sampling during training, with gradient ascent performed on batches of prompt-response pairs to maximize this objective.

**Training Dataset.** We train OMNISAPIENS-7B RL on the unified prompt-target pairs from HUMAN BEHAVIOR ATLAS, where all 101k samples have been standardized into a consistent format (Section 3.2). Each sample's target is represented as free-text to enable end-to-end optimization through the decoder head.

**Prompt Structure.** To enhance the model's reasoning capabilities, we augment each original prompt with explicit reasoning instructions. Specifically, for each behavioral analysis task with original prompt content, we construct the training prompt as:

```
{content}
You FIRST think about the reasoning process as an internal
    monologue and then provide the final answer. The reasoning
    process MUST BE enclosed within <think> </think> tags. The
    final answer MUST BE put in \boxed{}.
```

This structure encourages the model to generate intermediate reasoning steps before producing the final behavioral classification or interpretation, improving both interpretability and accuracy.

**Reward Architecture.** We design a composite reward function that evaluates three aspects of model outputs. Let $o_{(q,i,t)}$ denote the model's generated response for prompt $q$, and $y_q$ the ground truth label. The total reward is:

$$r_{(q,i,t)} = r_{\text{acc}}(o_{(q,i,t)}, y_q) + \lambda_{\text{format}} \cdot r_{\text{format}}(o_{(q,i,t)}) + \lambda_{\text{sim}} \cdot r_{\text{sim}}(o_{(q,i,t)}, y_q), \tag{5}$$

where:

- $r_{\text{acc}}(o_{(q,i,t)}, y_q) = \mathbb{1}[\text{extract}(o_{(q,i,t)}) = y_q]$ is the accuracy reward, which equals 1 if the extracted answer from the boxed content matches the ground truth exactly (case-insensitive), and 0 otherwise.

- $r_{\text{format}}(o_{(q,i,t)}) = \mathbb{1}[\text{has\_think}(o_{(q,i,t)})] + \mathbb{1}[\text{has\_boxed}(o_{(q,i,t)})]$ evaluates format compliance, awarding 0.5 points each for proper use of `<think>` tags and boxed notation.

- $r_{\text{sim}}(o_{(q,i,t)}, y_q) = \cos(\mathbf{e}(o_{(q,i,t)}), \mathbf{e}(y_q))$ computes the cosine similarity between sentence embeddings of the predicted and ground truth labels, providing partial credit for semantically similar but not exact matches.

In our experiment, we set $\lambda_{\text{format}} = 0.2$ and $\lambda_{\text{sim}} = 0.5$.

## C  EXPERIMENTAL DETAILS & HYPERPARAMETERS

In this appendix, we provide the full details of our training and inference in our experimental setups.

### C.1  EXPERIMENTAL PROCEDURES

In the multi-task experiments (Section 4.1), general multimodal LLMs pretrained on broad multimodal data (Qwen 2.5-Omni-7B, Qwen 2.5-VL-7B, Gemma-3-4B) and on human-related datasets (HumanOmni-7B) are taken directly from Hugging Face and evaluated in inference mode, without additional fine-tuning. We use the prompt–target samples formulated in Section 3.2, which contain free-text responses for all tasks and datasets, ensuring direct compatibility with the models' trained configurations.

In the transfer learning experiments (Section 4.2), we fine-tune both the pretrained OMNISAPIENS-7B SFT and Qwen 2.5-Omni-7B separately on held-out sets. For a fair comparison, we attach a classifier head to the penultimate layer of Qwen 2.5-Omni-7B's backbone, following the same design described in Appendix B.1. This setup helps control for design differences, making the primary factor of comparison the presence or absence of pretraining on the diverse behavioral tasks in HUMAN BEHAVIOR ATLAS.

For training the Behavioral Adapter Module (BAM), we keep the backbone of the pretrained OMNISAPIENS-7B SFT fixed and integrate BAM as described in Section 3.6. Gradient updates are applied only to the BAM and the classifier or decoder heads. BAM is lightweight and additive, introducing minimal computational overhead and leaving the backbone representations intact, which makes it practical to train separate BAMs for each dataset. This approach accounts for variations in behavioral descriptors and label spaces across datasets, with dataset-specific BAMs capturing these nuances while mitigating interference from the differing behavioral descriptor signals across datasets. For datasets that include both audio and visual modalities, we train two separate BAMs: one for the MediaPipe visual descriptors and one for the OpenSmile audio descriptors, each providing an additive residual update to $h_{penult}$. For datasets with only a single modality, we train a BAM solely for the available set of behavioral descriptors.

## C.2 HYPERPARAMETERS UTILIZED

For pretraining OMNISAPIENS-7B SFT on HUMAN BEHAVIOR ATLAS, whether in the multitask setup (Section 4.1) or the transfer learning setup (Section 4.2, with 4 datasets held out from the pretraining corpus), we train for 5 epochs and select the checkpoint with the lowest cross-entropy loss on the validation set. We adopt an effective batch size of 512 (micro-batch size of 2 across 2 GPUs, with 128 gradient accumulation steps). Low-rank adaptation (LoRA) is applied in full float32 precision, with a learning rate of $1 \times 10^{-4}$ selected after a parameter search between $1 \times 10^{-5}$ and $1 \times 10^{-3}$. The LoRA configuration uses rank $r = 32$, scaling factor $\alpha = 64$, dropout rate 0.05, and targets the modules [q_proj, k_proj, v_proj, o_proj, gate_proj, down_proj, up_proj]. We employ a cosine learning rate scheduler with 50 warmup steps to stabilize training during the initial phase.

For fine-tuning OMNISAPIENS-7B SFT and Qwen 2.5-Omni-7B on the separate held-out sets in the transfer learning experiments (Section 4.2), we adopt consistent configurations to ensure a fair comparison. Specifically, we reuse the same LoRA parameters and learning rate search as described in the pretraining configuration above, while adjusting the effective batch sizes to account for the varying sizes of the held-out datasets (e.g., DAIC-WOZ contains only ~100 samples). The effective batch sizes are as follows: 4 (micro-batch size of 2 across 2 GPUs, gradient accumulation of 1) for DAIC-WOZ; 16 (micro-batch size of 2 across 2 GPUs, gradient accumulation of 4) for MUStARD, 32 (micro-batch size of 2 across 2 GPUs, gradient accumulation of 8) for MELD (EMO) and MOSEI (SEN).

For training the Behavioral Adapter Module (BAM) (Section 3.6), we first perform a learning rate search. For the unfrozen classifier and decoder heads, we explore learning rate values between $1 \times 10^{-4}$ and $5 \times 10^{-4}$, selecting $1 \times 10^{-4}$ as the optimal rate. For the BAM adapters, we search between $1 \times 10^{-4}$ and $8 \times 10^{-4}$, selecting $5 \times 10^{-4}$. Each BAM is trained for four epochs, and the epoch achieving the lowest cross-entropy loss on the validation set is retained for testing. Consistent with the transfer learning experiments, we adjust effective batch sizes to account for dataset-specific sample sizes. The effective batch sizes are: 4 (micro-batch size of 2 across 2 GPUs, gradient accumulation of 1) for DAIC-WOZ; 16 (micro-batch size of 2 across 2 GPUs, gradient accumulation of 4) for MUStARD and MIMEQA; and 32 (micro-batch size of 2 across 2 GPUs, gradient accumulation of 8) for the remaining datasets. Additionally, since BAM includes an intermediate hidden layer (Section 3.6), we set its hidden dimension to 256, while applying a dropout rate of 0.10.

SFT and BAM training was done on 8x Nvidia H200 141GB GPUs, the Adam optimizer was utilized for all experiments. Additionally, while this was disable during transfer experiments and the training of the Behavioral Adapter Module. All experiments were conducted on the Pytorch and the Accelerate framework.

For reinforcement learning, we train OMNISAPIENS-7B RL using GRPO with the following key hyperparameters. The model is initialized from Qwen2.5-Omni-7B and trained for 10 epochs with a batch size of 256. We use a learning rate of $5e-7$ with the AdamW optimizer. For each prompt during rollout, we sample $n = 5$ responses to form a group for advantage estimation. The maximum prompt and response lengths are both set to 4096 tokens. We process multimodal inputs (audio, video, images) with modality-aware batching to ensure efficient GPU utilization. The PPO mini-batch size is set to 128 with a micro-batch size of 2 per GPU. We disable KL regularization in the reward ($\beta = 0$ in the objective) following Yu et al. (2025). We perform validation every 5 steps to monitor convergence.

# D    ADDITIONAL EXPERIMENTS

## D.1    ZERO-SHOT TRANSFER LEARNING

Following from the experimental setup in Section 4.2, we conduct an additional transfer learning experiment to test the zero-shot transfer of OMNISAPIENS-7BRL. Specifically, we trained OmniSapiens-7B RL on Human Behavior Atlas while holding out MOSEI (SEN), MELD (EMO), DAIC-WOZ (DEP), and MUStARD (SAR) entirely from the training corpus. We then evaluated the model in a strict zero-shot setting on these held-out datasets. For comparison, we used Qwen 2.5-Omni-7B, which shares the same underlying architecture as OmniSapiens-7B RL but is not pretrained on Human Behavior Atlas. This enables a clean attribution of the transfer gains solely from pretraining on HUMAN BEHAVIOR ATLAS, as opposed to any architectural modifications.

Table 8: Zero-shot evaluation on held-out datasets. Bold denotes best score. Heldout datasets are defined the same way as in Table 5

| Dataset | OMNISAPIENS-7B RL | Qwen 2.5-Omni-7B |
|---|---|---|
| MOSEI$^\ddagger$ (SEN) | **0.247** | 0.201 |
| MELD$^\ddagger$ (EMO) | **0.549** | 0.403 |
| DAIC-WOZ$^\ddagger$ (DEP) | **0.499** | 0.108 |
| MUStARD$^\dagger$ (SAR) | **0.596** | 0.445 |

From this table, we observe that pretraining on the Human Behavior Atlas benchmark yields substantial zero-shot gains on held-out datasets. OmniSapiens-7B RL outperforms Qwen 2.5-Omni-7B across all four datasets, with improvements of +22.99% (+0.046) on MOSEI (SEN), +36.2% (+0.146) on MELD (EMO), +362.04% (+0.391) on DAIC-WOZ (DEP), and +33.9% (+0.151) on MUStARD. These findings suggest that Human Behavior Atlas pretraining provides strong transferable representations across diverse behavioral tasks, under zero-shot settings.

## D.2    TRANSFER TO ADDITIONAL UNSEEN DATASET

Table 9: Transfer to IEMOCAP after minimal fine-tuning (1 epoch).

| Dataset | OMNISAPIENS-7B SFT | Qwen 2.5-Omni-7B |
|---|---|---|
| IEMOCAP | **0.6213** | 0.5625 |

We further assess cross-dataset generalization by evaluating transfer to IEMOCAP, a large emotion-recognition benchmark that is fully excluded from pretraining and differs considerably from Human Behavior Atlas datasets in domain, speaker context, and annotation style. As shown in Table 9, Under a one-epoch fine-tuning budget, OmniSapiens-7B SFT (pretrained on Human Behavior Atlas) achieves a +10.5% improvement (+0.059) over Qwen 2.5-Omni-7B (without Human Behavior Atlas pretraining). This result suggests that the behavioral representations learned during multi-task pretraining may retain some adaptability to new emotion-recognition settings with distributional and contextual properties not present in Human Behavior Atlas. This offers an encouraging indication for transfer to new affective or emotion-recognition benchmarks, where contextual and annotation variability often limits the generalization performance of models trained on existing emotion datasets (Montag et al.).

## D.3    BAM ABLATION OF RAW AUDIO AND VIDEO FEATURES

To analyze the isolated contribution of behavioral descriptors, we conducted an ablation in which we removed the raw audio and video features during the BAM model's training and inference. The resulting model, which we term OmniSapiens-7B BAM (ABL), is evaluated across all tasks. From Table 10, we observe that removing the raw video and audio modalities leads to consistently weaker performance compared to the full BAM model, with OmniSapiens-7B BAM (ABL) underperforming BAM on most datasets. This suggests that behavioral descriptors provide complementary information to the raw audio and visual encoders, rather than serving as an effective standalone substitute. These findings offer practical guidance for deploying BAM within unified omni-modal architectures: behavioral descriptors are most beneficial when integrated jointly with raw multimodal signals

Table 10: Δ highlights the change in performance from OMNISAPIENS-7B SFT to OMNISAPIENS-7B BAM and OMNISAPIENS-7B BAM (ABL), shown as percentage (%) and absolute (Abs). Behavioral adapters (BAM, ABL) provide notable performance gains for several behavioral tasks, although benefits are not consistent across all tasks.

| Task | OMNISAPIENS-7B SFT | OMNISAPIENS-7B BAM | OMNISAPIENS-7B BAM (ABL) | $\Delta_{BAM}$ % (Abs) | $\Delta_{ABL}$ % (Abs) |
|---|---|---|---|---|---|
| NVC | 0.12 | 0.16 | 0.06 | +33.00 (+0.04) | -50.41 (-0.06) |
| SAR | 0.62 | 0.80 | 0.75 | +29.00 (+0.18) | +20.97 (+0.13) |
| HUM | 0.53 | 0.64 | 0.64 | +21.00 (+0.11) | +20.30 (+0.11) |
| DEP | 0.73 | 0.79 | 0.75 | +8.21 (+0.06) | +2.74 (+0.02) |
| EMO | 0.63 | 0.65 | 0.59 | +3.17 (+0.02) | -6.35 (-0.04) |
| SEN | 0.77 | 0.79 | 0.76 | +2.60 (+0.02) | -1.30 (-0.01) |
| PTSD | 1.00 | 1.00 | 1.00 | 0.00 (+0.00) | 0.00 (+0.00) |
| ANX | 0.91 | 0.91 | 0.909 | 0.00 (+0.00) | 0.00 (+0.00) |
| SOC | 0.26 | 0.20 | 0.22 | -23.08 (-0.06) | -14.40 (-0.04) |
| INT | 0.26 | 0.18 | 0.18 | -30.77 (-0.08) | -29.69 (-0.08) |

## D.4 BAM ABLATION OF HIDDEN DIMENSIONS

We evaluate a more computationally expensive configuration of BAM using 512 hidden dimensions, from Table 11. From our results, we observe that increasing the hidden dimension does not necessarily yield better performance: 6 out of 10 tasks show noticeably worse performance, while only 2 out of 10 tasks improve. This highlights that larger adapters do not consistently translate to performance gains, with a lighter design of 256 hidden dimensions striking a more effective balance between efficiency and accuracy." We hope this clarifies the reviewer's question on the effect of varying the BAM design

Table 11: Δ highlights the change in performance from OMNISAPIENS-7B SFT to OMNISAPIENS-7B BAM (256) and OMNISAPIENS-7B BAM (512), shown as percentage (%) and absolute (Abs). Behavioral adapters generally provide performance gains for several behavioral tasks, though not uniformly across all tasks.

| Task | OMNISAPIENS-7B SFT | OMNISAPIENS-7B BAM (256) | OMNISAPIENS-7B BAM (512) | $\Delta_{BAM (256)}$ % (Abs) | $\Delta_{BAM (512)}$ % (Abs) |
|---|---|---|---|---|---|
| NVC | 0.12 | 0.16 | 0.11 | +33.00 (+0.04) | -9.09 (-0.01) |
| SAR | 0.62 | 0.80 | 0.76 | +29.00 (+0.18) | +22.58 (+0.14) |
| HUM | 0.53 | 0.64 | 0.67 | +21.00 (+0.11) | +25.94 (+0.14) |
| DEP | 0.73 | 0.79 | 0.77 | +8.21 (+0.06) | +5.48 (+0.04) |
| EMO | 0.63 | 0.65 | 0.64 | +3.17 (+0.02) | +1.59 (+0.01) |
| SEN | 0.77 | 0.79 | 0.77 | +2.60 (+0.02) | 0.00 (+0.00) |
| PTSD | 1.00 | 1.00 | 1.00 | 0.00 (+0.00) | 0.00 (+0.00) |
| ANX | 0.91 | 0.91 | 0.91 | 0.00 (+0.00) | 0.00 (+0.00) |
| SOC | 0.26 | 0.20 | 0.15 | -23.08 (-0.06) | -41.63 (-0.11) |
| INT | 0.26 | 0.18 | 0.22 | -30.77 (-0.08) | -14.06 (-0.04) |

## D.5 BAM ABLATION OF AUDIO AND VISUAL BEHAVIORAL DESCRIPTORS

Given that OMNISAPIENS-7B BAM integrates both the audio and visual behavioral descriptors, we conduct an additional experiment to ablate each visual and audio descriptor stream during OMNISAPIENS-7B BAM's training and inference. This allows us to analyze the contribution of each visual and audio stream.

Table 12: Δ highlights the change in performance from OMNISAPIENS-7B SFT to each BAM variant (with audio and visual behavioral features ablated), shown as percentage (%) and absolute (Abs). Dash (-) highlights unavailable features due to the absence of a modality etc.

| Task | OMNISAPIENS-7B SFT | OMNISAPIENS-7B BAM (AUD+VIS) | OMNISAPIENS-7B BAM (AUD) | OMNISAPIENS-7B BAM (VIS) | $\Delta_{BAM (Aud+Vis)}$ % (Abs) | $\Delta_{BAM (Aud)}$ % (Abs) | $\Delta_{BAM (Vis)}$ % (Abs) |
|---|---|---|---|---|---|---|---|
| EMO | 0.63 | 0.65 | 0.64 | 0.64 | +3.17 (+0.02) | +1.59 (+0.01) | +1.59 (+0.01) |
| HUM | 0.53 | 0.64 | 0.62 | 0.65 | +21.00 (+0.11) | +16.54 (+0.09) | +22.18 (+0.12) |
| INT | 0.26 | 0.18 | 0.26 | 0.18 | -30.77 (-0.08) | – | -29.69 (-0.08) |
| PTSD | 1.00 | 1.00 | 1.00 | 1.00 | 0.00 (+0.00) | 0.00 (+0.00) | 0.00 (+0.00) |
| ANX | 0.91 | 0.91 | 0.91 | 0.91 | 0.00 (+0.00) | +0.11 (+0.00) | +0.11 (+0.00) |
| DEP | 0.73 | 0.79 | 0.78 | 0.73 | +8.21 (+0.06) | +6.85 (+0.05) | – |
| SEN | 0.77 | 0.79 | 0.78 | 0.78 | +2.60 (+0.02) | +1.30 (+0.01) | +1.30 (+0.01) |
| SAR | 0.62 | 0.80 | 0.81 | 0.81 | +29.00 (+0.18) | +30.65 (+0.19) | +30.65 (+0.19) |
| SOC | 0.26 | 0.20 | 0.26 | 0.18 | -23.08 (-0.06) | – | -29.96 (-0.08) |
| NVC | 0.12 | 0.16 | 0.12 | 0.18 | +33.00 (+0.04) | – | +48.76 (+0.06) |

From Table 12, the results indicate that while integrating behavioral features is generally beneficial, the synergistic effects of combining acoustic and visual descriptors are task-dependent. For example, tasks such as EMO and SEN exhibit clear gains when both streams are used together: BAM (Aud+Vis)

achieves scores of 0.65 and 0.79, respectively, compared with 0.64 and 0.78 when using either BAM (Vis) or BAM (Aud) alone. However, for other tasks such as HUM and SAR, a single descriptor stream can outperform the combined configuration. BAM (Vis) achieves 0.65 on HUM and 0.81 on SAR, while BAM (Aud) reaches 0.81 on SAR. In contrast, BAM (Aud+Vis) performs 0.64 on HUM and 0.80 on SAR. These mixed outcomes highlight that the utility of each behavioral stream varies across tasks. This reinforces the motivation for BAM's lightweight and computationally efficient design (two layers with 256 hidden dimensions). Namely, it enables practitioners to selectively discard a behavioral descriptor stream (Vis/ Audio) and retrain BAM when a particular descriptor stream does not provide meaningful benefit.

## D.6 STUDY ON THE COMPUTATIONAL EFFICIENCY OF BAM

BAM is designed to be a lightweight module (i.e. with a small Feed-Forward Network) that can be integrated into multimodal LLMs with minimal additional computational cost. To quantify BAM's runtime and memory cost, we evaluate its impact on latency and GPU memory consumption. Specifically We mea- sure the mean and standard deviation of latency, as well as peak VRAM usage, by running forward passes over 500 samples from the Human Behavior Atlas. All experiments use batch size 1, FP16 mixed precision, and are run on 4× NVIDIA H200 GPUs, with the first batch used for warm-up.

Table 13: Inference latency and peak VRAM usage with and without the BAM adapter, evaluated over 500 samples from the Human Behavior Atlas. All experiments use FP16, batch size 1, and 4× NVIDIA H200 GPUs.

| Configuration | Mean Latency (s) | Std. Latency (s) | Peak VRAM (MB) |
|---|---|---|---|
| WITHOUT BAM | 0.284 | 0.029 | 24,265 |
| WITH BAM | 0.300 | 0.047 | 24,291 |

As shown in Table 13, BAM introduces only a marginal increase in both latency and VRAM usage (e.g., +0.016s mean latency and +26 MB peak VRAM). These results highlight that BAM provides a plug-and- play adaptation mechanism that adds negligible computational overhead

## D.7 ANALYSIS OF DATASET IMBALANCE

We conducted an analysis on the potential impact of dataset imbalance on performance in HUMAN BEHAVIOR ATLAS, as shown by Table 14 and Table 15. This includes computing the spearman correlation of performance against sample counts, and analyzing the performance rank of datasets with fewer samples.

Table 14: Analysis of dataset sample frequency versus model results.

| Dataset | Sample Count | Sample Rank | SFT Score | SFT Rank | RL Score | RL Rank | BAM Score | BAM Rank |
|---|---|---|---|---|---|---|---|---|
| MOSEI (SEN) | 31453 | 1 | 0.744 | 6 | 0.224 | 15 | 0.775 | 6 |
| IntentQA (INT) | 16297 | 2 | 0.256 | 15 | 0.486 | 12 | 0.177 | 15 |
| MELD (EMO) | 13706 | 3 | 0.709 | 7 | 0.699 | 5 | 0.711 | 10 |
| MELD (SEN) | 13706 | 3 | 0.746 | 5 | 0.571 | 9 | 0.744 | 7 |
| MOSEI (EMO) | 8598 | 5 | 0.614 | 11 | 0.581 | 8 | 0.607 | 12 |
| CREMA-D (EMO) | 7442 | 6 | 0.542 | 12 | 0.501 | 11 | 0.548 | 13 |
| Social-IQ (SOC) | 6437 | 7 | 0.257 | 14 | 0.304 | 14 | 0.201 | 14 |
| CH-SIMSv2 (SEN) | 4403 | 8 | 0.813 | 4 | 0.393 | 13 | 0.837 | 4 |
| TESS (EMO) | 2800 | 9 | 0.658 | 8 | 0.510 | 10 | 0.715 | 9 |
| UR-FUNNY (HUM) | 2125 | 10 | 0.532 | 13 | 0.639 | 7 | 0.644 | 11 |
| MMPsy (ANX) | 1275 | 11 | 0.909 | 2 | 0.919 | 2 | 0.909 | 2 |
| MMPsy (DEP) | 1275 | 11 | 0.839 | 3 | 0.814 | 3 | 0.839 | 3 |
| MimeQA (NVC) | 806 | 13 | 0.121 | 16 | 0.133 | 16 | 0.162 | 16 |
| MUStARD (SAR) | 690 | 14 | 0.624 | 10 | 0.647 | 6 | 0.795 | 5 |
| PTSD-in-the-Wild (PTSD) | 634 | 15 | 1.000 | 1 | 0.968 | 1 | 1.000 | 1 |
| DAIC-WOZ (DEP) | 189 | 16 | 0.626 | 9 | 0.729 | 4 | 0.738 | 8 |

Our findings suggest that dataset imbalance has negligible impact on per-dataset performance. As shown in Table 14 and Table 15, we compare each dataset's sample count and sample-count rank against its downstream performance under SFT, RL, and BAM. Interestingly, several of the smallest datasets (i.e., those ranked in the bottom five by sample count) achieve performance in the top half of all datasets across training methods, with the only exception being MimeQA. To quantify this more formally, we compute Spearman rank correlations between dataset sample count and model

Table 15: Spearman correlations between dataset size and performance across SFT, RL, and BAM models.

| Metric | Value |
|---|---|
| Spearman (SFT) | -0.134554126 |
| Spearman (RL) | -0.47971471 |
| Spearman (BAM) | -0.342235494 |

performance. Across all three SFT, RL, BAM, training methods, the correlations are negative, with the magnitudes being weak or non-existent, indicating no consistent positive monotonic relationship between dataset size and task performance. In other words, having more samples for a given dataset does not systematically correspond to better performance on that dataset. These results suggest that the performance disparities across datasets are more likely driven by differences in task difficulty and complexity rather than by dataset imbalance within the benchmark

## D.8 TRANSFER COMPARISON

We contrast the SFT results (i.e. OMNISAPIENS-7B SFT and Qwen 2.5-Omni-7B SFT) in our transfer experiments in Section 4.2 to the models that have not been fine-tuned on any of the heldout datasets, with the Table 16.

Table 16: Transfer to held-out datasets after minimal epoch fine-tuning (1 epoch). Bold denotes best score. DAIC-WOZ[*] uses 2 epochs due to only 107 training samples. MUStARD[†] presents a novel behavioral task (sarcasm). ‡ Other held-out datasets with tasks represented during pretraining.

| Dataset | OMNISAPIENS-7B SFT | Qwen 2.5-Omni-7B SFT | Gemma-3-4B | HumanOmniV2-7B | Qwen 2.5-Omni-7B | Qwen-2.5-VL-7B |
|---|---|---|---|---|---|---|
| MOSEI[‡] (SEN) | **0.724** | 0.612 | 0.617 | 0.633 | 0.602 | 0.317 |
| MELD[‡] (EMO) | **0.711** | 0.684 | 0.642 | 0.633 | 0.661 | 0.571 |
| DAIC-WOZ[*‡] (DEP) | **0.749** | 0.579 | 0.137 | 0.636 | 0.636 | 0.623 |
| MUStARD[†] (SAR) | **0.658** | 0.473 | 0.529 | 0.395 | 0.656 | 0.511 |

## D.9 ANALYSIS ON RETENTION OF GENERAL TEXT-GENERATION CAPABILITIES

We evaluate the free-form text responses generated by OmniSapiens-7B RL and by Qwen 2.5-Omni-7B, which shares the same architecture but lacks the additional Human Behavior Atlas training, across all question-answering tasks (IntentQA, MimeQA, Social-IQ 2.0) to analyze whether fine-tuning on Human Behavior Atlas influences general text generation quality. This involves computing perplexity over the generated outputs, and then evaluating fluency, coherence, and relevance using LLM-based scoring, following from established evaluation protocols for general text generation (Zheng et al., 2023; Radford et al., 2019; Chen et al., 2023). Across the four metrics as shown in Table 17, we observe no meaningful differences between Qwen 2.5-Omni-7B and OmniSapiens-7B RL. These findings indicate that continued training on Human Behavior Atlas does not degrade the underlying text generation ability of the model.

Table 17: Human evaluation and perplexity results.

| Model | Fluency | Coherence | Relevance | Perplexity |
|---|---|---|---|---|
| Qwen 2.5-Omni-7B | 3.682 | 3.562 | 3.365 | 542.676 |
| OmniSapiens-7B RL | 3.698 | 3.566 | 3.375 | 542.676 |

We highlight the LLM-judge prompt that we utilize for analyzing fluency, coherence, relevance in the following.

**LLM Judge Prompts**

**System Instructions**

You are an impartial evaluator of AI-generated answers to open-ended questions. You will be given: - A user question - A model_answer to evaluate
Your job is to rate the model_answer on three dimensions from 1 to 5:
1) Fluency: - 1: Very poor - many grammatical errors, hard to understand. - 2: Poor - noticeable errors or very awkward phrasing; sometimes unclear. - 3: Fair - understandable overall; a few minor errors or clunky phrases. - 4: Good - clear and natural with only rare minor issues. - 5: Excellent - polished, natural, very easy to read.
2) Coherence: - 1: Very poor - disjointed, self-contradictory, or very confusing. - 2: Poor - some logical gaps or confusing wording; hard to follow. - 3: Fair - mostly coherent but with minor confusion or awkward flow. - 4: Good - ideas are easy to follow; no significant contradictions. - 5: Excellent - very clear and internally consistent.
3) Relevance to the question: - 1: Very poor - mostly irrelevant; barely addresses the question. - 2: Poor - partially on topic but misses the core of the question. - 3: Fair - addresses the general topic but may miss important aspects or be too generic. - 4: Good - directly answers the question and focuses on what was asked. - 5: Excellent - directly and specifically answers the question with minimal irrelevance.
Important: - Ignore `<think>` tags and their content. - Ignore follow-up questions. - Do NOT penalize for brevity; short answers are acceptable if they satisfy the criteria above. - Do NOT penalize for minor stylistic preferences. - Do NOT judge factual correctness (accuracy is measured separately), except insofar as obviously nonsensical statements hurt coherence.

**User Instruction**

Question: {question}
Model Answer: {model_answer}

## D.10 LLM-JUDGE RELIABILITY

To assess the reliability of our LLM-judge for free-text response evaluation, we randomly sample 500 predictions from Qwen 2.5-VL-7B across the *SOC*, *INT*, and *NVC* tasks. An independent human evaluator manually judges each prediction against the ground truth. We then compute Cohen's kappa between these human judgments and the original LLM-judge (GPT-5-nano) to quantify human–judge agreement. To further justify our choice of a closed-source judge, we additionally evaluate Qwen3-30B as an open-source judge under the same protocol.

Table 18: Reliability and robustness of LLM-based judges for free-text evaluation. We report Cohen's $\kappa$ between human annotations, GPT-5-nano, and Qwen3-30B under different judging setups. All results are computed over 500 randomly sampled predictions from Qwen 2.5-VL-7B across SOC, INT, and NVC tasks.

| Judge | Compared Against | Prompt Setting | Cohen's $\kappa$ |
|---|---|---|---|
| GPT-5-nano | Human | Original judge prompt | 0.78 |
| GPT-5-nano | Qwen3-30B | Original judge prompt | 0.69 |
| Qwen3-30B | Human | Original judge prompt | 0.64 |
| GPT-5-nano | Human | Paraphrased / reordered prompt | 0.75 |

From Table 18, we find that GPT-5-nano exhibits strong agreement with human judgments, achieving a Cohen's kappa of $\kappa = 0.78$. It also shows reasonable inter-judge consistency with Qwen3-30B ($\kappa = 0.69$). Qwen3-30B proves to be a weaker judge overall, with a lower human–judge agreement of $\kappa = 0.64$.

To evaluate the robustness of LLM-judge performance to prompt variations, we construct an alternative judging prompt by automatically paraphrasing the existing prompt using GPT-5-nano and randomly reordering rubric items. Under this altered prompt, GPT-5-nano maintains high hu-

man–judge agreement ($\kappa = 0.75$), as shown from Table 18. These findings collectively support the reliability and robustness of using LLM-based judges for scoring free-text generation tasks. We highlight the paraphrased judge prompt in the following.

---

**Paraphrased LLM Judge Prompts to Evaluate Prompt Sensitivity**

**System Instructions**

**MimeQA**: Carefully consider the following question and answers regarding understanding of a mime performance. You will be shown a "gold-standard" answer from a human annotator, referred to as the "Reference Answer", and a "Candidate Answer". Your task is to determine whether the candidate captures the core meaning of the reference answer using the following criteria:

- The candidate must avoid adding incorrect or invented details and should not introduce narrative elements absent from the reference.
- The candidate must provide one clear, primary answer and must not include option lists, hedging, or deferrals.
- Because the videos depict mime performances, unseen actions, objects, or mimed elements are acceptable if and only if they are meaningfully relevant to the question.
- The candidate answer should align closely enough with the reference answer that it could reasonably replace it, without shifting to a different subject or introducing unsupported entities.

**Social-IQ 2.0**: Carefully consider the following question and answer regarding understanding of a video. You will be shown a "gold-standard" answer from human annotators, referred to as the "Reference Answer", and a "Candidate Answer". Your task is to judge whether the candidate captures the core meaning of the reference answer using the following criteria:

- The candidate must offer one definite answer, without listing alternatives or postponing judgment.
- The candidate's explanation must remain within the conceptual scope of the reference answer and must not introduce claims that contradict it; minor consistent elaborations are allowed.
- The candidate should not introduce distracting, inaccurate, or incompatible details and must focus on the same primary subject or object as the reference answer.

**IntentQA**: Carefully consider the question and answers regarding the intent behind actions in a video. You will be shown a "gold-standard" answer from human annotators, referred to as the "Reference Answer", and a "Candidate Answer". Your task is to judge whether the candidate presents a plausible interpretation of intent that does not contradict the reference answer using the following criteria:

- The candidate must present a single, specific interpretation of intent and must not provide alternative possibilities or noncommittal responses.
- The explanation should remain compatible with the reference answer and aligned with the question; minor additions are acceptable if they do not introduce conflict.
- The candidate must capture the core intent accurately without inserting misleading, irrelevant, or unsupported details and must not shift focus to a different subject.

**User Instruction**

Question: [question]
Candidate Answer: [candidate answer]
Reference Answer: [reference answer]
Please evaluate whether the candidate answer is correct according to the grading instructions.

---

### D.11 FULL RESULTS FOR SENTIMENT ANALYSIS

We provide the full results for sentiment analysis for CMU-MOSEI (Zadeh et al., 2018), which involves a 7-point sentiment labelling scale, in the following Table 19.

Table 19: 7 class Results on CMU-MOSEI emotion recognition.

| Model | CMU-MOSEI | | CH-SIMS v2 | |
|---|---|---|---|---|
| | 7-class F1 | 7-class Acc | 7-class F1 | 7-class Acc |
| Gemma-3-4B | .216 | .221 | .439 | .435 |
| HumanOmniV2-7B | .210 | .196 | .432 | .417 |
| Qwen 2.5-Omni-7B | .199 | .215 | .397 | .391 |
| Qwen-2.5-VL-7B | .229 | .202 | .411 | .361 |
| OMNISAPIENS-7B RL | .235 | .270 | .451 | .434 |
| OMNISAPIENS-7B SFT | .308 | .325 | .470 | .484 |
| OMNISAPIENS-7B BAM | .310 | .328 | .484 | .483 |

## D.12 RL ABLATION

We provide an additional ablation of our RL experiments, by altering the different rollouts and learning rates, as shown in Table 20. Beside showing the sensitivity of results to the different learning rates and rollouts, this highlights the rationale behind our choice of hyperparameters (rollouts of 5, learning rate of 5e-7), and learning algorithm (GRPO).

Table 20: Ablation studies on number of rollouts, learning rate, and training method. Best results are bolded. We report binary weighted F1 for SEN; mean per-class weighted accuracy for EMO; weighted F1 for HUM, SAR, ANX, DEP, PTSD; and LLM-Judge accuracy for SOC, INT, NVC.

| Model | EMO | | | | HUM | INT | PTSD | ANX | DEP | SEN | | | SAR | SOC | NVC |
|---|---|---|---|---|---|---|---|---|---|---|---|---|---|---|---|
| | CREMA-D | MELD (E) | MOSEI (E) | TESS | UR-FUNNY | IntentQA | PTSD_WILD | MMPSY (A) | MMPSY (D) | MELD (S) | CH-SIMSv2 | MOSEI (S) | MUStARD | Social-IQ 2.0 | MimeQA |
| **Number of Rollouts (lr = 5e-7)** | | | | | | | | | | | | | | | |
| Rollouts = 2 | .500 | .563 | .576 | .499 | **.643** | .422 | **.984** | .669 | .700 | **.588** | .383 | **.253** | **.670** | .168 | .162 |
| Rollouts = 5 | **.501** | **.699** | **.581** | **.510** | .639 | **.486** | .968 | **.919** | .814 | .571 | .393 | .224 | .647 | **.304** | .133 |
| Rollouts = 20 | .495 | .572 | **.581** | .507 | .600 | .448 | **.984** | .882 | **.841** | .577 | **.403** | .252 | **.670** | .179 | **.178** |
| **Learning Rate (rollouts = 5)** | | | | | | | | | | | | | | | |
| lr = 1e-6 | .496 | .531 | **.588** | .500 | **.640** | .408 | **.968** | .851 | **.843** | .574 | .371 | .224 | .640 | .161 | .150 |
| lr = 5e-7 | **.501** | **.699** | .581 | **.510** | .639 | **.486** | **.968** | **.919** | .814 | .571 | **.393** | **.224** | **.647** | **.304** | .133 |
| lr = 1e-7 | **.501** | .521 | .577 | .508 | .636 | .399 | .937 | .755 | .702 | **.577** | .370 | .221 | .580 | .161 | **.174** |
| **Training Method (rollouts = 5, lr = 5e-7)** | | | | | | | | | | | | | | | |
| RLOO | **.501** | .585 | .573 | .499 | .613 | .436 | **.984** | .877 | **.852** | **.617** | **.404** | **.276** | **.675** | .173 | **.170** |
| GRPO | **.501** | **.699** | **.581** | **.510** | **.639** | **.486** | .968 | **.919** | .814 | .571 | .393 | .224 | .647 | **.304** | .133 |

Table 21: Multimodal datasets for behavioral analysis with their data sources and geographic locations. Bold locations indicate confirmed data collection sites; other locations are based on project authors' institutions.

| Dataset | Source | Location |
|---|---|---|
| CMU-MOSEI | YouTube videos (1000+ speakers, various topics) | Pittsburgh, PA, USA (CMU) |
| MELD | Friends TV series episodes | Singapore (SUTD) |
| TESS | Controlled lab recordings with professional actresses | **Toronto, Canada** |
| CREMA-D | Controlled lab recordings with 91 professional actors | Philadelphia, PA, USA (UPenn) |
| CH-SIMSv2 | Chinese movies, TV series, multimedia content | Beijing, China (Tsinghua University) |
| Social-IQ 2.0 | YouTube videos, movie clips, vehicle recordings | Pittsburgh, PA, USA (CMU) |
| IntentQA | Natural daily social activity videos (NExT-QA) | Xi'an, China (Xi'an Jiaotong Univ.) |
| MimeQA | YouTube mime performance videos | Cambridge, MA, USA (MIT) |
| UR-FUNNYv2 | TED talk videos (1741 speakers) | Rochester, NY, USA (Univ. of Rochester) |
| MUStARD | TV shows (Friends, Big Bang Theory, etc.) | Ann Arbor, MI, USA (Univ. of Michigan) |
| DAIC-WOZ | Clinical interviews with virtual interviewer "Ellie" | **Los Angeles, CA, USA** (data collection at USC) |
| MMPsy | Web-based anxiety/depression interviews (Mandarin) | Guangdong Province, China |
| PTSD-in-the-Wild | Natural video recordings with varied conditions | Paris, France (Université de Paris) |

Table 22: Examples of converting heterogeneous datasets into a unified prompt–target format by referencing or including the modalities explicitly in the prompts, while the prediction targets are standardized either in a discrete label set for classification head setups, or a free-text response for decoder head setups. Reformulated prompts with targets are released as part of our benchmark.

| Dataset | Original Inputs / Outputs | Reformulated Prompt | Target |
|---|---|---|---|
| MMPsy (Depression) | Text (Transcript) / PHQ-9 (0–27 numerical score) | 嗯，受到了情绪影响，嗯，之前感觉状态都挺好，但是后面有感觉有点难，生活难。写作业，打游戏，睡觉、吃饭。父母，学校还有一点呢。学业太多，还有呢，老是有点不太那个哎呀同学，哎。呃，较多吧，我家就挺欣赏挺牛逼的。呃呃，学业真多，作业写不完，被老师罚。嗯，带来了无法控制的影响，呃，还好。不想讲不想讲，不是。`The above is an audio transcript of a participant speaking in a clinical setting. Based on the speech, does the speaker show signs of depression? Respond with the following: Depression or No Depression.` | **Discrete label set:** `{Depression, No depression}` **Free-text response:** `Answer: Depression / No depression` |
| URFunny (Humor) | Video + Audio + Transcript / Humor detection (true, false) | `<video> <audio> you know the story The above is a video and audio recording from a conversation, along with a transcript. Does this video segment contain a humorous punchline? Answer with one word: True or False.` | **Discrete label set:** `{True, False}` **Free-text response:** `Answer: True / False` |
| MOSEI (Emotion) | Video + Audio + Transcript / Emotion recognition (joy, sadness, anger, neutral) | `<video> <audio> I had a really long day at work and I'm just exhausted, but seeing you made me smile The above is a video and audio recording from a conversation, along with the transcript. What emotion is the speaker expressing? Choose from the following: Happy, Sadness, Anger, Surprise, Disgust, Fear.` | **Discrete label set:** `{Happy, Sadness, Anger, Surprise, Disgust, Fear}` **Free-text response:** `Answer: Happy / Sadness / Anger / Surprise / Disgust / Fear` |
| CREMA-D (Emotion) | Audio + Transcript / Emotion recognition (anger, disgust, fear, happy, neutral, sad) | `<audio> It's 11 o'clock. The above is a speech recording along with the transcript from a clinical context. What emotion is the speaker expressing? Answer with one word from the following: Anger, Disgust, Fear, Happy, Neutral, Sadness.` | **Discrete label set:** `{Anger, Disgust, Fear, Happy, Neutral, Sadness}` **Free-text response:** `Answer: Anger / Disgust / Fear / Happy / Neutral / Sadness` |

