# OpenReview forum: "Human Behavior Atlas: Benchmarking Unified Psychological And Social Behavior Understanding"
_ICLR.cc/2026/Conference — ICLR 2026 Poster_

### Official Review · Reviewer_tGVo · 2025-10-25

**Soundness:** 3
**Presentation:** 3
**Contribution:** 3
**Rating:** 6
**Confidence:** 4

**Summary:**

By integrating diverse modalities and heterogeneous datasets under a single LLM-based paradigm, it demonstrates the potential of large models to generalize across emotional, cognitive, and behavioral domains. While some architectural or methodological innovations are limited, the work’s scale, data diversity, and behavior-centric vision represent a meaningful contribution that could serve as a foundation for future research in multimodal affective computing.

**Strengths:**

1. The authors successfully harmonize heterogeneous datasets and tasks (e.g., emotion recognition, personality, mental health indicators) under a single large language model (LLM)-based paradigm, which demonstrates strong potential for generalization across domains.
2. Leverages an unprecedented scale of multimodal behavioral data, enabling the model to learn rich representations that reflect both emotional and cognitive dimensions of human behavior.
3. Experimental results demonstrate solid transfer capabilities across a wide range of downstream behavioral and affective computing tasks

**Weaknesses:**

1. The authors designed two types of output: a dedicated classifier for categorical predictions, and a decoder that generates open-ended responses from the final hidden states. However, this classifier+decoder structure seems inconsistent with the logic of using large language models. Ideally, we expect an LLM to use a single decoder to produce all outputs, while the classifier approach merely leverages the model’s encoding capability, which deviates from the intended design philosophy.
2. The main contribution lies in integrating various tasks and datasets into a unified LLM-based training paradigm. However, the authors did not introduce any specific architectural designs to enhance model capability, for example, handling behavioral temporal dynamics, or effectively leveraging facial landmarks, acoustic cues, or pose keypoints. Instead, these modalities are simply fed into the model as additional data for brute-force learning, which shows limited innovation.
3. Currently, LLMs perform poorly on regression tasks, yet affective computing tasks such as valence-arousal estimation or depression assessment require fine-grained regression rather than simple binary classification. The authors’ choice to convert PHQ-9 into a binary classification problem thus oversimplifies the task and fails to reflect real-world application scenarios.
4. A common issue in multi-task learning is negative transfer or gradient conflict, but the authors did not address whether their method incorporates any targeted solutions to mitigate these problems.
5. The superior performance in transfer learning could be attributed to the use of large-scale and diverse multimodal data (visual/acoustic cues) rather than the proposed model architecture or data structure itself, the paper does not clarify this distinction.

**Questions:**

1. Although the authors devote substantial discussion to explaining how behavioral descriptors benefit model training (a point already well established in many multi-task affective models) it is unclear why the model itself is not designed to predict these descriptors directly. Doing so would align more closely with the paper’s title, “Human Behavior Atlas”, which implies an explicit mapping or prediction of behavioral factors.
2. In the implementation section, the authors carefully adjust minibatch sizes across different tasks. However, it would be more informative if they could quantitatively present the impact of task diversity on training. Moreover, since the datasets vary greatly in size, it remains unclear how the authors address data imbalance during training or does imbalance affect the model performance.
3. The comparison models were not trained on such a large and diverse dataset, which raises concerns about fairness. A more rigorous and convincing evaluation would involve cross-dataset validation, for example, testing affective models across IEMOCAP, Aff-Wild2, or AffectNet, to demonstrate the generalization ability of the proposed approach.

---

> ### Author Response · Authors · 2025-11-17
> **Author's Response - Thank you for your feedback (part 1)**
>
> We thank the reviewer for the constructive feedback. To the best of our ability, we believe we have addressed
> the critiques provided. This includes clarfiying the rationale behind our paper, as well as including additional
> experiments and analyses. Please let us know should there be any outstanding concerns.
>
> > The authors designed two types of output: a dedicated classifier for categorical predictions, and a decoder that generates open-ended responses from the final hidden states. However, this classifier+decoder structure seems inconsistent with the logic of using large language models. Ideally, we expect an LLM to use a single decoder to produce all outputs, while the classifier approach merely leverages the model’s encoding capability, which deviates from the intended design philosophy.
>
> We thank the reviewer for raising this concern. Indeed, we agree with the utility of accomplishing the
> different tasks within a single decoder head, and how that falls in line with the motivation behind utilizing
> LLMs. To this end, we would like to clarify that we had in fact included a variant of the model trained on
> the benchmark (OmniSapiens 7B-RL), which does not include classifier heads and instead leverages a sole
> decoder output structure. We noted this in L300-301, stating that OmniSapiens-7B RL “handles all tasks
> within a single decoder head”. We apologize if this was unclear. To avoid ambiguity, we will make this
> more explicit, and add the following statement to the paragraph in L300-304, “_OmniSapiens-7B RL does
> not make use of any classifier heads; all outputs are produced directly from a single generative decoder. This
> preserves the full flexibility of the LLM decoding architecture while enabling a clean comparison against
> the hybrid classifier-head-plus-decoder setup used in OmniSapiens-7B SFT_” .
> Additionally, we would like to affirm that the classifier and decoder structure (OmniSapiens-7B SFT)
> was only one of three variants (OmniSapiens-7B RL, OmniSapiens-7B BAM) that we experimented with.
> Its purpose is solely to provide comparative insights across different architectural choices for the community. This aligns with the core contribution of the paper, the curated benchmark, as well as the empirical experiments and analysis.
>
> > The main contribution lies in integrating various tasks and datasets into a unified LLM-based training paradigm. However, the authors did not introduce any specific architectural designs to enhance model capability, for example, handling behavioral temporal dynamics, or effectively leveraging facial land-marks, acoustic cues, or pose keypoints. Instead, these modalities are simply fed into the model as additional data for brute-force learning, which shows limited innovation.
>
> We thank the reviewer for this thoughtful comment. Regarding this point, we would like to clarify that our paper is positioned primarily as a benchmark and datasets contribution. We noted this in our introductory paragraph, which states that “These model variants are meant as vanilla instances rather than new methods, intended primarily to investigate Human Behavior Atlas” (L85-L87). Accordingly, the core contributions of the work lie in (i) the standardization of datasets and evaluation metrics, (ii) the construction of hand-crafted instruction-prompt sets, (iii) the development of standardized behavioral-descriptor features, and (iv) the accompanying empirical studies and analyses that demonstrate the benchmark’s utility. On this note, we qualify that the model architectures are kept intentionally vanilla, without any significantly new model architectures implemented; this allows us to most cleanly attribute our empirical findings to the design of the benchmark itself, as opposed to any significant modifications in the model architecture, which may inadvertently distract/ confound our scientific findings regarding the benchmark’s training analysis.
>
> That being said, while proposing new model architectures is not the central focus of this paper, we introduce the Behavioral Adapter Module (BAM), which, to our knowledge, is among the first adapter-style components designed specifically to infuse behavioral descriptors into an LLM backbone post-training, while freezing the LLM’s backbone to prevent disruption of learned backbone representations. This specific module, as pointed out in Section 3.6 (L267-L297), does explore how behavioral descriptors, which include facial landmarks, acoustic cues, pose keypoints, can be integrated in a manner that complements the raw multimodal inputs. We apologize if this was not sufficiently clear. To avoid ambiguity, we will add the following line to L267-L297, “_We introduce a simple variant, OmniSapiens-7B BAM, by equipping OmniSapiens-7B SFT with a Behavioral Adapter Module (BAM) that incorporates behavioral descriptors, allowing us to examine the performance impact of integrating the behavioral descriptors (i.e., pose, prosody, and facial landmarks) into the LLM backbone_”.

---

> > ### Author Response · Authors · 2025-11-17
> > **Author's Response - Thank you for your feedback (part 2)**
> >
> > > The main contribution lies in integrating various tasks and datasets into a unified LLM-based training paradigm. However, the authors did not introduce any specific architectural designs to enhance model capability, for example, handling behavioral temporal dynamics, or effectively leveraging facial landmarks, acoustic cues, or pose keypoints. Instead, these modalities are simply fed into the model as additional data for brute-force learning, which shows limited innovation.
> >
> > **(continued from author's response from part 1)**
> >
> > Finally, we emphasize that the motivation for introducing this behavioral adapter module is to study, rather than to propose, new architectures. Specifically, this is to investigate how different behavioral descriptors (facial landmarks, prosody, etc.) contribute to interpreting psychological and social behaviors, within the context of quickly adapting an already pretrained LLM. We additionally highlighted a series of experiments in Section 4.3 (L435-L475) that investigates the effects of integrating these specialized behavioral signals within LLMs, with the goal of making these insights accessible to the community.
> >
> > > Although the authors devote substantial discussion to explaining how behavioral descriptors benefit model training (a point already well established in many multi-task affective models) it is unclear why the model itself is not designed to predict these descriptors directly. Doing so would align more closely with the paper’s title, “Human Behavior Atlas”, which implies an explicit mapping or prediction of behavioral factors.
> >
> > We thank the reviewer for the constructive feedback. We agree that prior literature has demonstrated the benefits of behavioral descriptors for affective modeling. However, we would like to clarify that the focus of our investigation is not on the descriptors themselves, but on how they can be integrated into an LLM backbone in a non-invasive manner, that is, without altering or retraining the core LLM representations. While existing work has utilized such descriptors, to the best of our knowledge, no prior work has examined how to fuse these features into a large multimodal LLM while preserving its end-to-end backbone and generative structure. Motivated by this gap, our experiments examine the effectiveness of an adapter-style fusion mechanism. This keeps the LLM backbone frozen to systematically study the impact of integrating these features while maintaining the integrity of the LLM architecture. We hope this clarifies both the motivation and the novelty of this component of our analysis.
> >
> > Regarding the suggestion of directly predicting the behavioral descriptors. We appreciate the chance to clarify that the model and benchmark’s primary tasks fall within the scope of the interpreting human psychological and social states (i.e., affect, cognition), within a multi-task and generalization setting. Estimating behavioral descriptors such as pose or gaze is not itself a target task. Rather, these descriptors serve as auxiliary preprocessing signals that can enhance the interpretation of higher-level psychological and social states. In fact, this emphasis of our paper is closely aligned with the research interests of the broader affective and cognitive computating community [2, 3]. Introducing descriptor-prediction tasks would shift the focus away from the central aim of the benchmark, which is to analyze and map psychological and social behaviors.
> >
> > In this context, the title Human Behavior Atlas is intended to evoke a “mapping” or “discovery” of internal states achieved by a unified model, not the prediction of low-level behavioral cues themselves. We hope this response clarifies the intent and scope of our work.
> >
> > With these points in mind, it is our belief that papers are best understood when evaluated in relation to how well-scoped they are with respect to the community’s ongoing research efforts, the goals they aim to address, and the credibility of their scientific claims. Introducing additional objectives, even when potentially valuable, may shift attention toward areas outside the work’s intended aims, which can inadvertently make it more difficult to interpret the work’s primary contributions. Sincerely, we hope that this perspective can be taken into consideration when evaluating our work, and we remain open to suggestions in this regard.

---

> > > ### Author Response · Authors · 2025-11-17
> > > **Author's Response - Thank you for your feedback (part 3)**
> > >
> > > >  Currently, LLMs perform poorly on regression tasks, yet affective computing tasks such as valence-arousal estimation or depression assessment require fine-grained regression rather than simple binary classification. The authors’ choice to convert PHQ-9 into a binary classification problem thus oversimplifies the task and fails to reflect real-world application scenarios.
> > >
> > > We thank the reviewer for this insightful point. We agree that LLMs generally underperform on regression tasks and that certain specialized assessment, such as PHQ-9, benefit from the precision afforded by continuous estimation. We appreciate the opportunity to clarify our motivation for mapping these regression-style tasks to binary classifications within the Human Behavior Atlas.
> > >
> > > First, we contend that binary classification remains clinically meaningful in a considerable number of settings. As an initial screening step for depression, for instance, binary severity indicators (e.g., “depressed” vs. “non-depressed”) have been shown to offer actionable value in real-world triage and mental-health screening workflows. While such binary signals do not quantify the extent of depression, they still serve as recognized, practical decision thresholds in early-stage diagnostic pipelines [4]. In this light, we believe that binary formulations of these tasks still preserve their real-world utility, even as they carry inherent limitations compared to full-scale precise regression. By contrast, requiring an LLM to output precise numerical regression scores, despite strong evidence that current LLMs struggle with stable and calibrated continuous predictions, would risk undermining the model’s practical usefulness.
> > >
> > > We also appreciate the broader considerations raised by the reviewer regarding the limitations of unified LLM models on niche tasks, including continuous regression. At the same time, we believe it is important to recognize the substantial advantages that general, unified models can bring. Recent trends in the healthcare-AI community provide illustrative examples. The move toward core, unified models has driven significant progress in the development of general-purpose healthcare agents, largely due to gains in efficiency, extensibility, and scalability [6, 1]. This shift reflects the significant value of unified models that can support a wide range of tasks, even if this comes at the cost of peak precision on a set of niche objectives (i.e. regression prediction for continuous PHQ-9 scores). In this context, we believe the same trade-off applies in the domain of psychological and social behavior interpretation. The potential benefits of unifying diverse behavioral tasks within a single LLM backbone, such as shared representations, cross-task generalization, and reduced system complexity, may outweigh the limitations that arise for specific, niche regression tasks. We hope this perspective helps clarify the reasoning behind our design choices and the broader motivations for prioritizing unified modeling in this setting, and are happy to provide further clarification if necessary.

---

> ### Author Response · Authors · 2025-11-17
> **Author's Response - Thank you for your feedback (part 4)**
>
> > A common issue in multi-task learning is negative transfer or gradient conflict, but the authors did not
> address whether their method incorporates any targeted solutions to mitigate these problems.
>
> We thank the reviewer for raising this thoughtful point. We would like to clarify that the primary contribution of our work lies in the curation of the benchmark and in the accompanying analytical experiments and analyses built around it. In this context, our goal was not to design a method that explicitly mitigates negative transfer or gradient conflict. Introducing a specialized technique to address negative transfer would fundamentally alter the training dynamics we aim to examine, making it difficult to attribute our empirical findings to the benchmark itself, as opposed to any significant architectural modifications. In fact, we clarify that a core contribution of our study is to characterize the benchmark’s inherent multi-task compatibility and transfer effects under a clean, unconfounded baseline setting (i.e. vanilla model setups without any significant architectural alterations, in order to examine raw training dynamics on the benchmark).
>
> To this end, our paper focuses on empirically characterizing both negative and positive transfer across tasks by training models solely on the benchmark and examining how well they generalize. This includes training on a corpus that excludes specific held-out tasks and datasets, followed by evaluation on these heldout splits (Section 4.2, L375–L434). Through this setup, we observe that the benchmark enables promising transfer on a range of tasks, including sentiment analysis, emotion recognition, depression assessment, and sarcasm detection. Building on these findings, we highlight avenues for future research. For example, investigating whether positive transfer is an emergent property. We hope this response clarifies the rationale behind our study.
>
> >  The superior performance in transfer learning could be attributed to the use of large-scale and diverse
> multimodal data (visual/acoustic cues) rather than the proposed model architecture or data structure
> itself, the paper does not clarify this distinction.
>
> We thank the reviewer for this helpful feedback. We agree that the distinction can be made clearer, and we will add the following clarification to the paragraph in L376–L384, “_For comparison, Qwen 2.5-Omni-7B SFT -- which shares the same architecture as OmniSapiens-7B SFT, differing only in its absence of pretraining on Human Behavior Atlas, is also fine-tuned on these held-out datasets under the same training conditions. This enables a clean isolation of the effect of training on our benchmark, ensuring that observed improvements can be attributed to the data and tasks within Human Behavior Atlas rather than to architectural changes._” We hope this clarification addresses the reviewer’s concern.

---

> ### Author Response · Authors · 2025-11-17
> **Author's Response - Thank you for your feedback (part 5)**
>
> > In the implementation section, the authors carefully adjust minibatch sizes across different tasks. However, it would be more informative if they could quantitatively present the impact of task diversity on training. Moreover, since the datasets vary greatly in size, it remains unclear how the authors address data imbalance during training or does imbalance affect the model performance.
>
> We thank the reviewer for this thoughtful point. We agree that additional analysis on task diversity and data imbalance would strengthen our paper. In response, we conducted further quantitative experiments to directly evaluate whether dataset-size imbalance influences model performance. We now include these experiments and analysis as a stand-alone paragraph with accompanying tables in Appendix D.4
>
> ### Table: Analysis of Dataset Sample Frequency versus Results
>
> | Dataset | Sample Count | Sample Count Rank | SFT Score | SFT Rank | RL Score | RL Rank | BAM Score | BAM Rank |
> |--------|--------------|-------------------|-----------|----------|----------|---------|-----------|----------|
> | MOSEI (SEN) | 31453 | 1 | 0.744 | 6 | 0.224 | 15 | 0.775 | 6 |
> | IntentQA (INT) | 16297 | 2 | 0.256 | 15 | 0.486 | 12 | 0.177 | 15 |
> | MELD (EMO) | 13706 | 3 | 0.709 | 7 | 0.699 | 5 | 0.711 | 10 |
> | MELD (SEN) | 13706 | 3 | 0.746 | 5 | 0.571 | 9 | 0.744 | 7 |
> | MOSEI (EMO) | 8598 | 5 | 0.614 | 11 | 0.581 | 8 | 0.607 | 12 |
> | CREMA-D (EMO) | 7442 | 6 | 0.542 | 12 | 0.501 | 11 | 0.548 | 13 |
> | Social-IQ (SOC) | 6437 | 7 | 0.257 | 14 | 0.304 | 14 | 0.201 | 14 |
> | CH-SIMSv2 (SEN) | 4403 | 8 | 0.813 | 4 | 0.393 | 13 | 0.837 | 4 |
> | TESS (EMO) | 2800 | 9 | 0.658 | 8 | 0.51 | 10 | 0.715 | 9 |
> | UR-FUNNY (HUM) | 2125 | 10 | 0.532 | 13 | 0.639 | 7 | 0.644 | 11 |
> | MMPsy (ANX) | 1275 | 11 | 0.909 | 2 | 0.919 | 2 | 0.909 | 2 |
> | MMPsy (DEP) | 1275 | 11 | 0.839 | 3 | 0.814 | 3 | 0.839 | 3 |
> | MimeQA (NVC) | 806 | 13 | 0.121 | 16 | 0.133 | 16 | 0.162 | 16 |
> | MUStARD (SAR) | 690 | 14 | 0.624 | 10 | 0.647 | 6 | 0.795 | 5 |
> | PTSD-in-the-Wild (PTSD) | 634 | 15 | 1.000 | 1 | 0.968 | 1 | 1.000 | 1 |
> | DAIC-WOZ (DEP) | 189 | 16 | 0.626 | 9 | 0.729 | 4 | 0.738 | 8 |
>
> ### Table: Spearman Correlations of Performance Against Sample Counts
>
> | Metric | Value |
> |--------|--------|
> | Spearman (SFT) | -0.134554126 |
> | Spearman (RL) | -0.47971471 |
> | Spearman (BAM) | -0.342235494 |
>
> We add, “_Our findings suggest that dataset imbalance has negligible impact on per-dataset performance. As shown in Tables 11 and 12, we compare each dataset’s sample count and sample-count rank against its downstream performance under SFT, RL, and BAM. Interestingly, several of the smallest datasets (i.e., those ranked in the bottom five by sample count) achieve performance in the top half of all datasets across training methods, with the only exception being MimeQA. To quantify this more formally, we compute Spearman rank correlations between dataset sample count and model performance. Across all three SFT, RL, BAM, training methods, the correlations are negative, with the magnitudes being weak or non-existent, indicating no consistent positive monotonic relationship between dataset size and task performance. In other words, having more samples for a given dataset does not systematically correspond to better performance on that dataset. These results suggest that the performance disparities across datasets are more likely driven by differences in task difficulty and complexity rather than by dataset imbalance within the benchmark._”
>
> Regarding the impact of task diversity, we add the following analysis to Section 4.2, L413 as a stand-alone paragraph, to strengthen our analysis of the impact of task diversity. “_Beyond the transfer improvements themselves, the comparison of OmniSapiens-7B SFT against the single-task trained Qwen 2.5-Omni-7B, as shown in Table 5., clarifies how the diversity of tasks in Human Behavior Atlas shapes the learned features while training on the benchmark. On represented tasks (SEN, EMO, DEP), OmniSapiens-7B SFT outperforms the single-task pretrained Qwen model even after minimal fine-tuning, indicating that exposure to multiple behavioral tasks enables the model to internalize shared behavioral features across heterogeneous tasks that generalize across datasets. Meanwhile, the fact that OmniSapiens-7B SFT also surpasses Qwen 2.5-Omni-7B on MUStARD, despite SAR being absent during pretraining, suggests that multi-task behavioral pretraining provides broader, more general behavioral patterns that may occasionally extend to novel behavioral phenomena. These transfer patterns, when contrasted against the single-task pretrained baseline of Qwen 2.5-Omni-7B, suggests that the diverse and heterogeneous behavioral tasks in Human Behavior Atlas contribute complementary and synergistic behavioral features, rather than conflicting or contradictory supervision._”
>
> We hope the above additional experiments and analysis addresses the reviewer’s points.

---

> ### Author Response · Authors · 2025-11-17
> **Author's Response - Thank you for your feedback (part 6)**
>
> > The comparison models were not trained on such a large and diverse dataset, which raises concerns about fairness. A more rigorous and convincing evaluation would involve cross-dataset validation, for example, testing affective models across IEMOCAP, Aff-Wild2, or AffectNet, to demonstrate the generalization ability of the proposed approach.
>
> We thank the reviewer for the insightful suggestion. We agree that evaluating OmniSapiens-7B on an additional large and diverse dataset would provide a more complete picture of cross-dataset generalization. In response, we conducted a new experiment on the IEMOCAP benchmark, which is entirely held-out from pretraining and differs from the emotion recognition datasets in Human Behavior Atlas in terms of domain, video contexts, and annotation design. Accordingly, we adopt a minimal fine-tuning budget of one epoch, following established protocols used to evaluate held-out generalization in foundational transfer-learning and representation-learning research (i.e., minimal, fixed-fine tuning budget as in [8, 7]). Under this setting, we fine-tune both OmniSapiens-7B SFT (pretrained on Human Behavior Atlas) and Qwen 2.5-Omni-7B
> (without Human Behavior Atlas pretraining) on IEMOCAP under the same minimal fine-tuning regime of one epoch. This setup helps clarify whether pretraining on Human Behavior Atlas provides any transferable behavioral structure when adapting to an emotion recognition dataset with entirely different domain, video contexts, and annotation designs.
>
> We add the following paragraph and the associated result tables to Appendix D.2
>
> ### Table. Transfer to IEMOCAP after Minimal Fine-Tuning (1 Epoch)
>
> | Dataset  | OmniSapiens-7B SFT | Qwen 2.5-Omni-7B |
> |----------|---------------------|-------------------|
> | **IEMOCAP** | **0.6213**           | 0.5625            |
>
> We add, “_We further assess cross-dataset generalization by evaluating transfer to IEMOCAP, a large emotion-recognition benchmark that is fully excluded from pretraining and differs considerably from Human Behavior Atlas datasets in domain, speaker context, and annotation style. As shown in Table 8., Under a one-epoch fine-tuning budget, OmniSapiens-7B SFT (pretrained on Human Behavior Atlas) achieves a +10.5% improvement (+0.059) over Qwen 2.5-Omni-7B (without Human Behavior Atlas pretraining). This result suggests that the behavioral representations learned during multi-task pretraining may retain some adaptability to new emotion-recognition settings with distributional and contextual properties not present in Human Behavior Atlas. This offers an encouraging indication for transfer to new affective or emotion-recognition benchmarks, where contextual and annotation variability often limits the generalization performance of models trained on existing emotion datasets [5]._”
>
> We hope this additional experiment appropriately addresses the reviewer’s request for broader cross-dataset validation.
>
> ---
>
> **References**
>
> [1] Wei Dai, Peilin Chen, Malinda Lu, Daniel Li, Haowen Wei, Hejie Cui, and Paul Pu Liang. Climb: Data foundations for large scale multimodal clinical foundation models. arXiv preprint arXiv:2503.07667, 2025.
>
> [2] Zheng Lian, Haiyang Sun, Licai Sun, Jiangyan Yi, Bin Liu, and Jianhua Tao. Affectgpt: Dataset and framework for explainable multimodal emotion recognition. arXiv preprint arXiv:2407.07653, 2024.
>
> [3] Rui Mao, Tao Wang, and Erik Cambria. Decoding metaphors and brain signals in naturalistic contexts: An empirical study based on eeg and metapro. In Proceedings of the Annual Meeting of the Cognitive Science Society, volume 47, 2025.
>
> [4] Douglas M Maurer, Tyler J Raymond, and Bethany N Davis. Depression: screening and diagnosis. American family physician, 98(8):508–515, 2018.
>
> [5] Christian Montag, Chengzhong Xu, Michiel Spap´e, and Erik Cambria. The emotion labeling problem in affective computing research.
>
> [6] Jianing Qiu, Kyle Lam, Guohao Li, Amish Acharya, Tien Yin Wong, Ara Darzi, Wu Yuan, and Eric J Topol. Llm-based agentic systems in medicine and healthcare. Nature Machine Intelligence, 6(12):1418–1420, 2024.
>
> [7] Colin Raffel, Noam Shazeer, Adam Roberts, Katherine Lee, Sharan Narang, Michael Matena, Yanqi Zhou, Wei Li, and Peter J Liu. Exploring the limits of transfer learning with a unified text-to-text transformer. Journal of machine learning research, 21(140):1–67, 2020.
>
> [8] Jason Yosinski, Jeff Clune, Yoshua Bengio, and Hod Lipson. How transferable are features in deep neural networks? Advances in neural information processing systems, 27, 2014.

---

### Official Review · Reviewer_eg78 · 2025-10-29

**Soundness:** 2
**Presentation:** 3
**Contribution:** 2
**Rating:** 6
**Confidence:** 2

**Summary:**

The paper introduces **Human Behavior Atlas (HBA)**, a unified multimodal benchmark for psychological and social behavior understanding. Key contributions include:
- A standardized **prompt–target interface** across 14 datasets and 10+ behavioral tasks.
- A **Behavioral Adapter Module (BAM)** for injecting external behavioral descriptors (e.g., pose, acoustic features) into a frozen backbone.
- Three training variants: **SFT, BAM, and RL (GRPO)**, evaluated comprehensively across tasks.

**Strengths:**

- **Unified benchmark** with broad task coverage and standardized metrics.
- **Clean and modular design** of BAM, enabling non-invasive feature injection.
- **Consistent empirical gains** of behavior-specialized models over general MLLMs.

**Weaknesses:**

#### **1. Insufficient Ablation Studies on BAM and RL**
- **BAM Design Choices**: The paper does not justify key design decisions—e.g., why the adapter is inserted at the **penultimate layer**, why **mean and standard deviation pooling** is used for temporal descriptors, or why a **two-layer FFN** is chosen. A systematic sweep over injection points, hidden dimensions, and pooling strategies is missing.
- **Descriptor Contribution**: The relative importance of **visual vs. acoustic descriptors** is not analyzed. Ablations disabling each stream would clarify their individual and synergistic effects.
- **RL Hyperparameters**: The use of **β = 0 for KL regularization** is not motivated, and no sensitivity analysis is provided for group size, clipping epsilon, or reward scaling. Training stability and reward hacking risks are unexamined.

#### **2. Evaluation Protocol Gaps**
- **LLM-as-Judge Reliability**: The open-ended evaluation relies solely on a **single closed-source LLM judge** (GPT-5-nano). No human–judge agreement, inter-judge consistency, or prompt robustness tests are reported.
- **Prompt Isolation & Contamination**: The paper does not clarify whether the judge and model prompts are isolated, whether sessions are reset, or whether tools are disabled—raising concerns about evaluation integrity.

#### **3. Label Granularity Reduction**
- Sentiment labels in datasets like **MOSEI (7-point scale)** are collapsed to **binary positive/negative**, discarding nuanced distinctions. This limits comparability with prior work and may mask model failures on fine-grained sentiment.

#### **4. Metric Heterogeneity and Cross-Task Comparability**
- Discrete tasks use **weighted F1 or weighted accuracy**, while open-ended tasks use **LLM-judged TRUE-rate**. This inconsistency complicates cross-task aggregation and model ranking.
- No unified scoring mechanism (e.g., normalized score per task family) is proposed.

#### **5. Lack of Computational Efficiency Analysis**
- The computational cost of BAM (e.g., latency, memory overhead) is not reported, nor is its impact on inference speed or scalability—key for real-world deployment.

#### **6. Limited Discussion on Multimodal Fusion Strategy**
- The fusion of text, audio, and video is performed via **simple concatenation** in the embedding space. More advanced fusion mechanisms (e.g., cross-attention, gating) are not explored or motivated.

**Questions:**

1. **LLM-as-Judge Reliability**: Please report human–judge agreement (κ or ρ) on a 500-sample subset, add at least one open-source judge, and show sensitivity to prompt variations.
2. **BAM & RL Ablations**: Systematically vary BAM’s injection layer, hidden size, and descriptor streams. For RL, sweep β, group size, and reward terms—report accuracy vs. compute and training stability.
3. **Fine-Grained Evaluation**: Provide results on full label sets (e.g., 7-point sentiment) and justify binarization decisions with class-wise F1 and confusion matrices.
4. **Metric Harmonization**: Propose a unified scoring scheme (e.g., normalized score per task family) and report confidence intervals over multiple runs.
5. **Prompt and Protocol Transparency**: Document judge–model isolation measures and include a finalized, typo-free version of the judge rubric.

---

> ### Author Response · Authors · 2025-11-19
> **Note from Authors**
>
> We thank the reviewer for the feedback. We are currently working on the responses and running several experiments. We will post our responses in due time. We appreciate the patience from the reviewer, and look forward to our discussion.
>
> Best,
>
> Authors

---

> ### Author Response · Authors · 2025-11-25
> **Author's Response - Thank you for your feedback (part 1)**
>
> We thank the reviewer for the kind patience as we were running experiments and additional studies in line with the suggestions provided. While we are still awaiting the results from the RL ablations being run in the background, and will post them in due time, we have successfully conducted the additional ablations on BAM, as well as LLM-judge robustness tests. Please kindly find them in our following responses.
>
> ---
> > 1. Insufficient Ablation Studies on BAM and RL BAM Design Choices: The paper does not justify
> key design decisions—e.g., why the adapter is inserted at the penultimate layer, why mean and standard
> deviation pooling is used for temporal descriptors, or why a two-layer FFN is chosen. A systematic
> sweep over injection points, hidden dimensions, and pooling strategies is missing.
>
> We thank the reviewer for the valuable feedback. We agree that providing additional detail on the design
> decisions can strengthen the clarity of our discussion. In response, we have added new ablation studies in
> Appendix D, together with explicit justification for our use of mean and standard deviation pooling, as well
> as our choice of penultimate-layer integration for the adapter.
>
> To clarify these design choices, we add the following statement to Section 3.6 (OmniSapiens-7B BAM):
> “_We utilize a lightweight feed-forward network to enable computationally efficient adaptation of the
> multimodal LLM using behavioral features (see computational study in Appendix D). The temporal descrip-
> tors are pooled via mean and standard deviation to capture the distributional characteristics of the behavioral
> signals within each video or audio clip, and to maintain compatibility with the static text, audio, and video
> modality streams that are taken in by the multimodal LLM. We integrate the adapter at the penultimate layer
> rather than at the early fusion stage, as the early textualization and alignment of behavioral features would
> require substantially more training cost [2], which is contrary to the intended plug-and-play nature of the
> adapter design._”
>
>
> We also include additional ablation experiments examining model design choices, adding an experiment for the different BAM dimensions and features in Appendix D, where we analyze the performance impact where the hidden dimension of BAM is increased to 512 from the original 256:
>
> **Table: Δ indicates the performance change from OmniSapiens-7B SFT to BAM (hidden dimension of 256) and BAM (hidden dimension of 512), shown as percentage (%) and absolute (Abs).**
>
> | **Task** | **SFT** | **BAM (256)** | **BAM (512)** | **Δ BAM (256)** | **Δ BAM (512)** |
> |---------|---------|----------------|----------------|------------------|------------------|
> | **NVC** | 0.12 | 0.16 | 0.11 | +33.00% (+0.04) | −9.09% (−0.01) |
> | **SAR** | 0.62 | 0.80 | 0.76 | +29.00% (+0.18) | +22.58% (+0.14) |
> | **HUM** | 0.53 | 0.64 | 0.67 | +21.00% (+0.11) | +25.94% (+0.14) |
> | **DEP** | 0.73 | 0.79 | 0.77 | +8.21% (+0.06) | +5.48% (+0.04) |
> | **EMO** | 0.63 | 0.65 | 0.64 | +3.17% (+0.02) | +1.59% (+0.01) |
> | **SEN** | 0.77 | 0.79 | 0.77 | +2.60% (+0.02) | 0.00% (+0.00) |
> | **PTSD** | 1.00 | 1.00 | 1.00 | 0.00% (+0.00) | 0.00% (+0.00) |
> | **ANX** | 0.91 | 0.91 | 0.91 | 0.00% (+0.00) | 0.00% (+0.00) |
> | **SOC** | 0.26 | 0.20 | 0.15 | −23.08% (−0.06) | −41.63% (−0.11) |
> | **INT** | 0.26 | 0.18 | 0.22 | −30.77% (−0.08) | −14.06% (−0.04) |
>
> We add the following analysis, “_We evaluate a more computationally expensive configuration of BAM using 512 hidden dimensions from the Table above. From our results, we observe that increasing the hidden dimension does not necessarily
> yield better performance: 6 out of 10 tasks show noticeably worse performance, while only 2 out of 10
> tasks improve. This highlights that larger adapters do not consistently translate to performance gains, with a
> lighter design of 256 hidden dimensions striking a more effective balance between efficiency and accuracy._”
>
> We hope this clarifies the reviewer’s question on the effect of varying the BAM design.

---

> ### Author Response · Authors · 2025-11-25
> **Author's Response - Thank you for your feedback (part 2)**
>
> > • Descriptor Contribution: The relative importance of visual vs. acoustic descriptors is not analyzed. Ablations disabling each stream would clarify their individual and synergistic effects.
>
> We thank the reviewer for the thoughtful feedback. We agree that ablating each behavioral descriptor stream
> helps clarify both the individual and synergistic contributions of the visual and acoustic components. In
> response, we have conducted an additional experiment in which we independently ablate the acoustic and
> visual descriptors during BAM training and inference. The corresponding table and discussion have been
> added to Appendix D.
>
> **Table: Δ indicates the change from OmniSapiens-7B SFT to each BAM variant (Aud+Vis, Aud-only, Vis-only), shown as percentage (%) and absolute (Abs). “--” denotes unavailable comparisons due to missing modalities.**
>
> | **Task** | **SFT** | **BAM (Aud+Vis)** | **BAM (Aud)** | **BAM (Vis)** | **Δ BAM (Aud+Vis)** | **Δ BAM (Aud)** | **Δ BAM (Vis)** |
> |---------|---------|-------------------|---------------|---------------|----------------------|------------------|------------------|
> | **EMO** | 0.63 | 0.65 | 0.64 | 0.64 | +3.17% (+0.02) | +1.59% (+0.01) | +1.59% (+0.01) |
> | **HUM** | 0.53 | 0.64 | 0.62 | 0.65 | +21.00% (+0.11) | +16.54% (+0.09) | +22.18% (+0.12) |
> | **INT** | 0.26 | 0.18 | 0.26 | 0.18 | −30.77% (−0.08) | -- | −29.69% (−0.08) |
> | **PTSD** | 1.00 | 1.00 | 1.00 | 1.00 | 0.00% (+0.00) | 0.00% (+0.00) | 0.00% (+0.00) |
> | **ANX** | 0.91 | 0.91 | 0.91 | 0.91 | 0.00% (+0.00) | +0.11% (+0.00) | +0.11% (+0.00) |
> | **DEP** | 0.73 | 0.79 | 0.78 | 0.73 | +8.21% (+0.06) | +6.85% (+0.05) | -- |
> | **SEN** | 0.77 | 0.79 | 0.78 | 0.78 | +2.60% (+0.02) | +1.30% (+0.01) | +1.30% (+0.01) |
> | **SAR** | 0.62 | 0.80 | 0.81 | 0.81 | +29.00% (+0.18) | +30.65% (+0.19) | +30.65% (+0.19) |
> | **SOC** | 0.26 | 0.20 | 0.26 | 0.18 | −23.08% (−0.06) | -- | −29.96% (−0.08) |
> | **NVC** | 0.12 | 0.16 | 0.12 | 0.18 | +33.00% (+0.04) | -- | +48.76% (+0.06) |
>
> We add the following discussion and analysis to Appendix D,
> “_The results indicate that while integrating behavioral features is generally beneficial, the synergistic
> effects of combining acoustic and visual descriptors are task-dependent. For example, tasks such as EMO
> and SEN exhibit clear gains when both streams are used together: BAM (Aud+Vis) achieves scores of 0.65
> and 0.79, respectively, compared with 0.64 and 0.78 when using either BAM (Vis) or BAM (Aud) alone. However, for other tasks such as HUM and SAR, a single descriptor stream can outperform the combined
> configuration. BAM (Vis) achieves 0.65 on HUM and 0.81 on SAR, while BAM (Aud) reaches 0.81 on
> SAR. In contrast, BAM (Aud+Vis) performs 0.64 on HUM and 0.80 on SAR.
> These mixed outcomes highlight that the utility of each behavioral stream varies across tasks. This
> reinforces the motivation for BAM’s lightweight and computationally efficient design (two layers with 256
> hidden dimensions). Namely, it enables practitioners to selectively discard a behavioral descriptor stream
> (Vis/ Audio) and retrain BAM when a particular descriptor stream does not provide meaningful benefit._”
>
> We hope this clarifies the reviewer’s question on the influence of effect of each stream.
>
> > • BAM & RL Ablations: Systematically vary BAM’s injection layer, hidden size, and descriptor streams.
>
> We thank the reviewer for the comment. We have addressed the feedback on BAM ablations in our above response which appends the additional BAM experiments. For the RL ablations, we will
> append the RL results in due time.
>
> > • RL Hyperparameters: The use of β = 0 for KL regularization is not motivated, and no sensitivity
> analysis is provided for group size, clipping epsilon, or reward scaling. Training stability and reward
> hacking risks are unexamined.
>
> We thank the reviewer for the comment. We are running the RL ablations and will append the RL results in due time.
>
> > For RL, sweep β, group size, and reward terms—report accuracy vs. compute and training stability.
>
> We thank the reviewer for the comment. We are running the RL ablations and will append the RL results in due time.

---

> ### Author Response · Authors · 2025-11-25
> **Author's Response - Thank you for your feedback (part 3)**
>
> > • 2. Evaluation Protocol Gaps LLM-as-Judge Reliability: The open-ended evaluation relies solely on a single closed-source LLM judge (GPT-5-nano). • No human–judge agreement, inter-judge consistency, or prompt robustness tests are reported.
>
> We thank the reviewer for this thoughtful feedback. We acknowledge that we can improve the robustness
> of our LLM-judging evaluation by adding these additional studies. In response, we have conducted an
> additional study for human-LLM agreement for GPT-5-nano, and we have also added an additional model
> (Qwen3-30B) to compare inter-judge consistency and have experimented with a prompt variant. We add this
> additional LLM-judge robustness tests to the Appendix D.10 LLM-Judge Reliability, with the associated
> table and analysis.
>
> **Table Reliability and Robustness of LLM-Based Judges: Cohen’s κ between human annotations, GPT-5-nano, and Qwen3-30B across 500 sampled predictions from SOC, INT, and NVC tasks.**
>
> | **Judge**     | **Compared Against** | **Prompt Setting**                  | **Cohen’s κ** |
> |---------------|-----------------------|-------------------------------------|----------------|
> | GPT-5-nano    | Human                 | Original judge prompt               | 0.78           |
> | GPT-5-nano    | Qwen3-30B             | Original judge prompt               | 0.69           |
> | Qwen3-30B     | Human                 | Original judge prompt               | 0.64           |
> | GPT-5-nano    | Human                 | Paraphrased / reordered prompt      | 0.75           |
>
> We add the following to Appendix D.11, “_To assess the reliability of our LLM-judge for free-text response evaluation, we randomly sample 500 predictions from Qwen 2.5VL-7B across the SOC, INT, and NVC tasks. An independent human evaluator manually judges each prediction against the ground truth. We then compute Cohen’s kappa between these human judgments and the original LLM-judge (GPT-5-nano) to quantify human–judge agreement. To further justify our choice of a closed-source judge, we additionally
> evaluate Qwen3-30B as an open-source judge under the same protocol. We find that GPT-5-nano exhibits
> strong agreement with human judgments, achieving a Cohen’s kappa of κ = 0.78. It also shows reason-
> able inter-judge consistency with Qwen3-30B (κ = 0.69). Qwen3-30B proves to be a weaker judge overall,
> with a lower human–judge agreement of κ= 0.64. To evaluate the robustness of LLM-judge performance
> to prompt variations, we construct an alternative judging prompt by automatically paraphrasing the existing
> prompt using GPT-5-nano and randomly reordering rubric items. Under this altered prompt, GPT-5-nano
> maintains high human–judge agreement (κ = 0.75). These findings collectively support the reliability and
> robustness of using LLM-based judges for scoring free-text generation tasks._”
>
> > • Prompt Isolation & Contamination: The paper does not clarify whether the judge and model prompts
> are isolated, whether sessions are reset, or whether tools are disabled—raising concerns about evaluation
> integrity.
>
> We thank the reviewer for this thoughtful comment. We appreciate the opportunity to clarify that the LLM-
> judge and model prompts are strictly separated, and that evaluation sessions are reset for every instance.
> In practice, this means that the context provided to the LLM judge consists only of (i) the question drawn
> from the publicly available prompts of each QA dataset (IntentQA, MimeQA, Social-IQ 2.0, and (ii) the
> model’s generated response. No additional conversational history or model-specific information is retained
> across evaluations. Furthermore, to ensure fairness and eliminate potential sources of bias, every model
> is evaluated using an identical prompt, guaranteeing that differences in performance arise solely from the
> generated model outputs rather than prompt variability.
>
> To avoid any ambiguity and to make this procedure fully transparent, we add the following clarification
> to Appendix A.5:
> “_Evaluation Protocol for LLM-Judge Scoring. For all QA-style tasks requiring LLM-judge evaluation,
> we ensure strict isolation between the model under evaluation and the judging model. Each evaluation is
> performed in a fresh session, with no shared history across queries. The LLM judge receives only the
> dataset-provided question and the model’s generated answer, with no additional system context. All models
> are presented with identical prompts to maintain consistency and prevent evaluation bias. This protocol
> guarantees that scoring reflects the intrinsic quality of model responses rather than prompt design or accumulated conversational context._”
>
> > • LLM-as-Judge Reliability: Please report human–judge agreement (κ or ℘) on a 500-sample subset,
> add at least one open-source judge, and show sensitivity to prompt variations.
>
> We thank the reviewer for this useful comment. We believe that we have addressed this feedback in the
> above response.

---

> > ### Author Response · Authors · 2025-11-25
> > **Author's Response - Thank you for your feedback (part 4)**
> >
> > > • Prompt and Protocol Transparency: Document judge–model isolation measures and include a finalized,
> > typo-free version of the judge rubric.
> >
> > We thank the reviewer for this useful comment. We believe that we have addressed this feedback in the above response.
> >
> > > • 3. Label Granularity Reduction Sentiment labels in datasets like MOSEI (7-point scale) are collapsed to
> > binary positive/negative, discarding nuanced distinctions. This limits comparability with prior work and
> > may mask model failures on fine-grained sentiment. Fine-Grained Evaluation: Provide results on full
> > label sets (e.g., 7-point sentiment) and justify binarization decisions with class-wise F1 and confusion matrices.
> >
> > We thank the reviewer for the useful comment. We have appended the MOSEI 7-point sentiment results to
> > the Appendix D. Please find the results appended below.
> >
> > **Table: Results on CMU-MOSEI Emotion Recognition. Best results are bolded; second best are underlined.**
> >
> > | **Model**               | **7-class F1** | **7-class Acc** |
> > |-------------------------|----------------|------------------|
> > | Gemma-3-4B              | 0.216          | 0.221            |
> > | HumanOmniV2-7B          | 0.210          | 0.196            |
> > | Qwen 2.5-Omni-7B        | 0.199          | 0.215            |
> > | Qwen-2.5-VL-7B          | 0.229          | 0.202            |
> > | OmniSapiens-7B RL  | 0.235          | 0.270            |
> > | OmniSapiens-7B SFT   | **0.308**      | **0.325**        |
> > | OmniSapiens-7B BAM  | **0.310**      | **0.328**        |
> >
> > We also appreciate the chance to clarify our decision to binarize the evaluations across different sentiment datasets in order to combine them. Sentiment datasets typically annotate different point scales (i.e.
> > 7-point in MOSEI vs 3-point in MELD). We believe that keeping the sentiment scales unbinarized would
> > result in combining F1 results on a 7-point sentiment scale with a 3-point scale. This would not comprise
> > an interpretable approach for unifying the results (i.e. naturally stronger performance on the 3-point scale
> > would overshadow the performance on the 7-point scale). Therefore, the standardization approach here involves taking the binarized (positive vs. negative labels), as this collapses the labels to a shared label space
> > (positive vs negative), with this binarized label space still regarded as the most foundational for sentiment
> > classification [1].
> >
> > > • 4. Metric Heterogeneity and Cross-Task Comparability Discrete tasks use weighted F1 or weighted
> > accuracy, while open-ended tasks use LLM-judged TRUE-rate. This inconsistency complicates cross-
> > task aggregation and model ranking. No unified scoring mechanism (e.g., normalized score per task
> > family) is proposed. Metric Harmonization: Propose a unified scoring scheme (e.g., normalized score
> > per task family) and report confidence intervals over multiple runs.
> >
> > We thank the reviewer for the thoughtful comment. We agree that unified scoring enhances comparability,
> > which is why our evaluation framework in Section 3.3 standardizes metrics within each task family (e.g.,
> > using binary weighted F1 for all sentiment classification datasets and mean per-class weighted accuracy for
> > emotion recognition datasets). We believe this ensures evaluation consistency where it is both meaningful
> > and valid.
> >
> > We also appreciate the opportunity to clarify why opted to avoid enforcing a single unified metric across
> > all behavioral tasks. Fully collapsing diverse tasks into one normalized scoring scheme would overly simplify well-established evaluation practices. Foundational work in multimodal affective computing has shown
> > that different tasks, such as sentiment classification versus emotion classification, require distinct metrics
> > due to their unique characteristics (e.g., severe class imbalance in emotion datasets [3]). Over-unifying
> > these metrics may run the risk of discarding task-specific evaluation requirements while diverging from
> > crucial foundational literature.
> >
> > Additionally, we believe imposing a single global metric across all tasks may reduce interpretability. To
> > this end, several tasks operate on fundamentally different scales (e.g., open-ended QA tasks evaluated via
> > LLM-judge accuracy versus categorical tasks evaluated via F1). Collapsing these into a single normalized
> > measure may obscure rather than clarify model performance.
> > For all of these reasons, our current design aims to maintain interpretability while still supporting com-
> > parability. Therefore, we leave task-appropriate metrics remain intact, while unifying intra-task metrics (i.e.
> > metrics across different datasets) where warranted. Moreover, we appended visuals such as Figure 2., to
> > allow readers to more accessibly compare relative model performance.
> >
> > We hope that this clarifies the rationale behind our framework for unifying the metrics

---

> ### Author Response · Authors · 2025-11-25
> **Author's Response - Thank you for your feedback (part 5)**
>
> > • 5. Lack of Computational Efficiency Analysis. The computational cost of BAM (e.g., latency, memory overhead) is not reported, nor is its impact on inference speed or scalability—key for real-world
> deployment.
>
> We thank the reviewer for the thoughtful suggestion. To address the feedback, we performed a computational analysis to measure the overhead introduced by BAM and appended both the explanatory text and
> the accompanying table to Appendix D. The goal of this additional experiment is to demonstrate that BAM
> meets the design intent of being a lightweight, low-cost adapter with minimal impact on inference efficiency. We append the following text to Appendix D, together with the following Table.
>
> **Table: Inference Latency and VRAM Usage with and without BAM. Evaluated over 500 samples from the Human Behavior Atlas (FP16, batch size 1, 4× NVIDIA H200 GPUs).**
>
> | **Configuration** | **Mean Latency (s)** | **Std. Latency (s)** | **Peak VRAM (MB)** |
> |-------------------|-----------------------|------------------------|----------------------|
> | Without BAM        | 0.284                 | 0.029                  | 24,265               |
> | With BAM           | 0.300                 | 0.047                  | 24,291               |
>
> We add the following, “_BAM is designed to be a lightweight module (i.e. with a small Feed-Forward Network) that can be
> integrated into multimodal LLMs with minimal additional computational cost. To quantify BAM’s runtime
> and memory cost, we evaluate its impact on latency and GPU memory consumption. Specifically We measure the mean and standard deviation of latency, as well as peak VRAM usage, by running forward passes
> over 500 samples from the Human Behavior Atlas. All experiments use batch size 1, FP16 mixed precision,
> and are run on 4× NVIDIA H200 GPUs, with the first batch used for warm-up.
> As shown in Table 2, BAM introduces only a marginal increase in both latency and VRAM usage (e.g.,
> +0.016s mean latency and +26 MB peak VRAM). These results highlight that BAM provides a plug-and-play adaptation mechanism that adds negligible computational overhead._”
>
> We hope this additional experiment addresses the reviewer’s request for additional experiments.
>
> > Limited Discussion on Multimodal Fusion Strategy. The fusion of text, audio, and video is performed via simple concatenation in the embedding space. More advanced fusion mechanisms (e.g., cross-attention, gating) are not explored or motivated
>
> We thank the reviewer for this helpful comment. We would like to clarify that the primary contributions of our work lie in the curation of the Human Behavior Atlas benchmark and in the accompanying empirical analyses that examine the effects of pretraining on this benchmark. To rigorously study the impact of training on Human Behavior Atlas, our experimental design compares OmniSapiens-7B SFT / RL against Qwen 2.5-Omni-7B. These models share highly similar architectures, with the key distinction being that OmniSapiens-7B is additionally pretrained on the Human Behavior Atlas. Therefore, to maintain a controlled comparison, we intentionally avoided introducing architectural changes, such as cross-attention fusion, into OmniSapiens-7B. Threfore OmniSapiens-7B retains Qwen 2.5-Omni-7B’s multimodal fusion strategy, which projects audio and video tokens directly into the LLM backbone.
>
> This ensures that improvements in performance can be cleanly attributed to the benchmark pretraining rather than to modifications in fusion mechanisms or any architectural changes. We intend for the observed performance gains to reflect the influence of the data and task distributions in Human Behavior Atlas, rather than architectural differences. This allows us to study the value and utility of our benchmark with greater scientific clarity and validity.
>
> We hope this helps clarify the motivation behind our experimental setup and model configurations.
>
> ---
> **References**
>
> [1] Richard Socher, Andrej Karpathy, Quoc V Le, Christopher D Manning, and Andrew Y Ng. Grounded
> compositional semantics for finding and describing images with sentences. Transactions of the Association for Computational Linguistics, 2:207–218, 2014.
>
> [2] Jin Xu, Zhifang Guo, Hangrui Hu, Yunfei Chu, Xiong Wang, Jinzheng He, Yuxuan Wang, Xian Shi,
> Ting He, Xinfa Zhu, et al. Qwen3-omni technical report. arXiv preprint arXiv:2509.17765, 2025.
>
> [3] Amir Zadeh, Paul Pu Liang, Soujanya Poria, Erik Cambria, and Louis-Philippe Morency. Multimodal
> language analysis in the wild: CMU-MOSEI dataset and interpretable dynamic fusion graph. In ACL,
> pages 2236–2246, 2018

---

> ### Author Response · Authors · 2025-12-02
> **Author's Response - Thank you for your feedback (part 6)**
>
> Dear Reviewer,
>
> We append our results and response for the RL ablation below.
>
> > RL Hyperparameters: The use of β = 0 for KL regularization is not motivated, and no sensitivity analysis is provided for group size, clipping epsilon, or reward scaling. Training stability and reward hacking risks are unexamined.
>
> We thank the reviewer for the thoughtful feedback. We had explicitly set $\beta$=0 for KL regularization, not arbitrarily, but rather motivated by the how recent work had shown how the KL regularization term could potentially lead to policy collapse [1]. We have also conducted an additional ablation for RL, showing the sensitivity of results to the different rollout sizes as well as learning rates. Please find the results for these below.  We believe this highlights the rationale behind our choice of hyperparameters (rollouts of 5, learning rate of 5e-7), and learning algorithm (GRPO).
>
>
> Best results are bolded. We report binary weighted F1 for SEN; mean per-class weighted accuracy for EMO; weighted F1 for HUM, SAR, ANX, DEP, PTSD; and LLM-Judge accuracy for SOC, INT, NVC.
>
> | Model | CREMA-D | MELD (E) | MOSEI (E) | TESS | UR-FUNNY | IntentQA | PTSD_WILD | MMPSY (A) | MMPSY (D) | MELD (S) | CH-SIMSv2 | MOSEI (S) | MUStARD | Social-IQ 2.0 | MimeQA |
> |-------|---------|----------|-----------|------|----------|----------|-----------|-----------|-----------|----------|-----------|-----------|---------|---------------|--------|
> | | **EMO** | | | | **HUM** | **INT** | **PTSD** | **ANX** | **DEP** | **SEN** | | | **SAR** | **SOC** | **NVC** |
> | **Number of Rollouts (lr = 5e-7)** | | | | | | | | | | | | | | | |
> | Rollouts = 2 | .500 | .563 | .576 | .499 | **.643** | .422 | **.984** | .669 | .700 | **.588** | .383 | **.253** | **.670** | .168 | .162 |
> | Rollouts = 5 | **.501** | **.699** | **.581** | **.510** | .639 | **.486** | .968 | **.919** | .814 | .571 | .393 | .224 | .647 | **.304** | .133 |
> | Rollouts = 20 | .495 | .572 | **.581** | .507 | .600 | .448 | **.984** | .882 | **.841** | .577 | **.403** | .252 | **.670** | .179 | **.178** |
> | **Learning Rate (rollouts = 5)** | | | | | | | | | | | | | | | |
> | lr = 1e-6 | .496 | .531 | **.588** | .500 | **.640** | .408 | **.968** | .851 | **.843** | .574 | .371 | .224 | .640 | .161 | .150 |
> | lr = 5e-7 | **.501** | **.699** | .581 | **.510** | .639 | **.486** | **.968** | **.919** | .814 | .571 | **.393** | **.224** | **.647** | **.304** | .133 |
> | lr = 1e-7 | **.501** | .521 | .577 | .508 | .636 | .399 | .937 | .755 | .702 | **.577** | .370 | .221 | .580 | .161 | **.174** |
> | **Training Method (rollouts = 5, lr = 5e-7)** | | | | | | | | | | | | | | | |
> | RLOO | **.501** | .585 | .573 | .499 | .613 | .436 | **.984** | .877 | **.852** | **.617** | **.404** | **.276** | **.675** | .173 | **.170** |
> | GRPO | **.501** | **.699** | **.581** | **.510** | **.639** | **.486** | .968 | **.919** | .814 | .571 | .393 | .224 | .647 | **.304** | .133 |
>
> In general, we see a learning rate of 5e-7 and rollout size of 5 to be a sweet spot for training. Performance difference tends to be small, as training runs with different hyperparameters usually converges to similar performance at different speeds.
>
> ---
> **References**
>
> [1] Marco Simoni, Aleksandar Fontana, Giulio Rossolini, and Andrea Saracino. Gtpo: Trajectory-based
> policy optimization in large language models. arXiv preprint arXiv:2508.03772, 2025.

---

### Official Review · Reviewer_3tAt · 2025-10-31

**Soundness:** 2
**Presentation:** 3
**Contribution:** 2
**Rating:** 6
**Confidence:** 3

**Summary:**

The authors propose an unified benchmark of diverse behavioral tasks designed to support the development of unified models for understanding psychological and social behaviors. The aim is to take the opportunities for scalability, cross-task transfer, and broader generalization.

**Strengths:**

- The paper is well organized and written
-  The unified benchmark named HUMAN BEHAVIOR ATLAS comes in a timely manner for the field. It covers a large spectrum of situations.

**Weaknesses:**

None

**Questions:**

What will be the restrictions about the use of this benchmark?

---

> ### Author Response · Authors · 2025-11-13
> **Author's Response - Thank you for your feedback**
>
> Dear Reviewer 3tAt,
>
> We thank you for your feedback on our paper. Regarding the restrictions about the use of the benchmark, we intend for the benchmark to be publicly available without any restrictions on research and academic usage. This will allow the benchmark to be utilized to advance research progress in the community of AI for psychological and social behavior understanding. We sincerely hope that the benchmark can support stronger scientific progress with respect to the development of unified models for this specific field.
>
> On that note, could we seek your feedback on some of the potential critiques you have on our paper. We would be keen on hearing them, and addressing them, to further improve our paper and scores.
>
> Best,
> Authors

---

> ### Author Response · Authors · 2025-11-24
> **Author's Note**
>
> Dear Reviewer 3tAt,
>
> Thank you once again for your feedback. As we kindly await your response on how to improve the paper's quality, we would like to update you with the additional experiments that we ran for our paper. Please find them in Appendix D of the revised submission. These include:
>
> 1. **Addition of Zero-shot Transfer Evaluation**: We have also added an experiment for zero-shot transfer performance, to complement our existing transfer experiments on fine-tuning with fixed epoch budget.
>
> 2. **Addition of ablation experiments on BAM, to understand the impact of training with solely the behavioral descriptors**: We conducted an additional experiment for training, where we ablated the raw vision and audio modalities, leaving only the integration of the behavioral descriptors.
>
> 3. **Additional experiment to evaluate whether text generation quality is retained after training on Human Behavior Atlas**
>
> 4. **Added empirical analysis on effects of dataset imbalance in benchmark**: We also conducted an additional analysis on whether data imbalance within the benchmark affects performance during multi-task training.
>
> 5. **Additional experiment on Cross-Dataset Transfer**: We conducted an additional transfer experiment on a large and diverse dataset (IEMOCAP) to provide a more complete picture of cross-dataset generalization.
>
> More details can be found in the "Update from Authors.." note at the top of this open review page, or Appendix D of the revised paper submission. The authors are deeply committed to improving the paper according to the feedback provided by the reviewers. Please let us know, should you have any feedback on how to improve the paper. We thank you very much for your time.

---

### Official Review · Reviewer_aP6y · 2025-10-31

**Soundness:** 3
**Presentation:** 3
**Contribution:** 2
**Rating:** 4
**Confidence:** 4

**Summary:**

The paper proposes HUMAN BEHAVIOR ATLAS, a unified multimodal benchmark focused on understanding psychological and social behaviors. The benchmark combines 13 existing datasets into a prompt-response instruction format across text, audio, and video modalities, totaling around 101k samples spanning 10 behavioral task categories. Three model variants are trained and evaluated: OMNISAPIENS-7B SFT, OMNISAPIENS-7B BAM, and OMNISAPIENS-7B RL. Results show that these models outperform general multimodal LLMs (Qwen 2.5, Gemma-3, HumanOmni) on both multi-task and transfer-learning evaluations.

**Strengths:**

1. The benchmark is a useful contribution that could help move multimodal LLM research toward more holistic behavioral understanding.
2. The experimental setup is thorough, covering 10 in-domain datasets and some "transfer" datasets.
3. The benchmark and code will become publicly available.

**Weaknesses:**

1. No new data is collected - the benchmark repackages existing datasets into a unified format. While valuable, this limits novelty.
2. The models are fully fine-tuned on the same datasets used for evaluation. In the transfer-learning section, the model is again fine-tuned for a few epochs on the "held-out" datasets, and only Qwen 2.5-Omni-7B is used for comparison. This setup mainly measures fine-tuning efficiency, not true zero-shot generalization.
3. The construction of prompts is unclear. It’s not specified whether they were hand-crafted or automatically generated, nor whether prompt robustness was tested.

**Questions:**

1. In Table 4, can you add results using only the behavioral descriptors (e.g., OpenSMILE, MediaPipe) to show how much these features alone contribute?
2. Since the models are fine-tuned on the same datasets used for evaluation, have you checked whether fine-tuning affects general-purpose abilities (e.g. text generation)?
3. Did you perform any true zero-shot evaluations to measure actual generalization?
4. How were prompts generated - hand-written or LLM-generated?
5. In Table 5, it would be interesting to add results before fine-tuning on the held-out datasets and also show results for the multimodal LLMs used in Table 4 for comparison.

---

> ### Author Response · Authors · 2025-11-19
> **Note from Authors**
>
> We thank the reviewer for the thoughtful feedback. We are currently working on the responses and additional experiments the reviewer has requested. We appreciate the patience given by the reviewer, and will respond in due time.
>
> Thank you!
>
> Authors

---

> ### Author Response · Authors · 2025-11-22
> **Author's Response - Thank you for your feedback (part 1)**
>
> The authors would like to thank the reviewer for investing time in the review process. We deeply appreciate the reviewer's feedback and have carefully considered the points raised to improve the paper. Accordingly, we have added new experiments and analyses as per the reviewer's suggestions.
>
> ----
>
> > No new data is collected - the benchmark repackages existing datasets into a unified format. While
> valuable, this limits novelty
>
> We thank the reviewer for this feedback. Regarding the point on the novelty of the paper, we appreciate the
> opportunity to clarify our paper’s novelty and contributions below:
> 1. **Regarding the addition of new data, the benchmark comprises new standardized behavioral descriptors (transcriptions, openpose, opensmile), as well as new hand-crafted instruction prompt sets.
> These were not present in the original datasets.** We would like to clarify that these represent new data
> that crucially enable unified model training, and these data were not previously present in the existing public
> datasets. Therefore, the paper extends curation and standardization of existing public datasets to also involve
> these elements of data creation, which we believe will be a valuable resource for the community. Moreover,
> these efforts are resource intensive and time-consuming to realize, making them relatively inaccessible for
> a considerable portion of the community to realize (as we elaborate in 4.).
> 2. **Regardless, we believe the benchmark’s intellectual novelty is independent from new data creation, and this intellectual contribution lies in the benchmark’s new taxonomy, task formulation, and
> experimental framework for behavioral AI. This follows precedent from top published foundational
> benchmark papers in ICML, NeurIPS, and ICLR that did not involve new data creation.** Our contribution intends to follow the spirit of these influential works, whose impact stemmed from introducing
> unifying conceptual frameworks and standardized evaluation paradigms that offered important intellectual
> insights through thoughtful benchmark design, such as systematic data collection, principled task formulation, and coherent integration of existing datasets, rather than through the construction of new data. Notable
> examples include GLUE [12] and SuperGLUE [11], both foundational benchmarks paper built entirely
> from already existing public datasets to evaluate general natural language understanding, among many other
> influential work. In a similar spirit, our benchmark introduces a conceptual framework that standardizes
> and unifies public behavioral datasets, to enable systematic evaluation, generalization analysis, and unified
> model development within the field of behavioral AI. In light of these precedents, we kindly appeal that the
> contribution of Human Behavior Atlas be evaluated based on its conceptual advances, i.e., its behavioral
> taxonomy, formulation of tasks, experimental framework for unified behavioral AI models in the context
> of generalizability and multi-task performance. These constitute new unified modelling contributions that
> were, to the best of our knowledge, previously unattempted in the field of developing AI for interpreting
> psychological and social behaviors. We remain open to further feedback on how these contributions are
> positioned within the broader psycholgoical and social behavioral AI landscape.
> 3. **We believe the benchmark has significant potential to advance the field by enabling unified models
> in AI for psychological and social behavior understanding. These approaches are still considerably
> understudied and unattempted in this field, with our paper being one of the first few, to our knowledge, to explore this angle of unified modelling for interpreting psychological and social behaviors**: We
> believe the Human Behavior Atlas benchmark provides the foundational infrastructure for the community
> to train unified models of human behavior. As elaborated in point 4., this endeavour would otherwise be
> inaccessible without a homogeneous and standardized benchmark, which can be both time-consuming and
> resource-intensive to curate. Furthermore, as discussed in the introduction (L33–L37), we believe that in a
> field still dominated by single-task models and bespoke architectures, this benchmark, being one of the first
> of its kind, has the potential to fundamentally shift the paradigm toward more generalizable and efficient
> architectures that support scalability and cross-task transfer. Such progress could direcly address significant challenges in data-scarce behavioral domains where cross-dataset transfer is valuable; for example, enhancing model development for autism detection [9]. As such, we contend that the benchmark is a new approach
> that could have considerable contributions in this AI field.

---

> ### Author Response · Authors · 2025-11-22
> **Author's Response - Thank you for your feedback (part 2)**
>
> > No new data is collected - the benchmark repackages existing datasets into a unified format. While valuable, this limits novelty
>
> **(continued authors' response)**
>
> 4. **Homogenizing and standardizing datasets into a benchmark is resource-intensive, rendering it
> largely inaccessible for the community:** Curating and standardizing public datasets within a conceptually
> coherent and homogeneous framework demands substantial effort and time, and cannot be achieved within
> a short period or with minimal resources. As detailed in our paper, this process involves developing a behavioral taxonomy (affect, cognition, pathology, social) to guide dataset collection (L164-L171), hand-crafting
> new instruction prompts for each of the 100k samples and 13 datasets (i.e. Table 3, L195-L208), extracting
> key behavioral features (i.e. pose, prosody, text transcriptions) (L241-L251). Given the scope and resource
> demands of these efforts, such standardization poses substantial barriers for most researchers seeking to
> train unified models for understanding human psychological and social behaviors; as evidenced by the fact
> that there have been limited or no unified models developed for this field. By making these curated resources publicly available, our benchmark lowers these barriers considerably, making it more accessible for
> the community to develop and evaluate unified behavioral AI models.
>
> At this juncture, we also note that other reviewers have recognized the benchmark’s contribution and
> novelty. For example, Reviewer eg78 highlighted the strength of providing a “unified benchmark with broad
> task coverage and standardized metrics” and Reviewer tGVo mentioned that “The authors successfully harmonize heterogeneous datasets and tasks (e.g., emotion recognition, personality, mental health indicators)
> under a single large language model (LLM)-based paradigm, which demonstrates strong potential for generalization across domains”. With these perspectives in mind, we kindly appeal that the reviewer may consider
> these facets of the work’s contribution.
>
> Additionally, we would like to emphasize that the contribution of this work extends beyond dataset
> standardization and curation. A central component of the paper lies in its analytical experiments and accompanying discussions. These include, multi-task evaluation (Section 4.1), transfer analysis (Section 4.2),
> and the examination of non-invasive integration of behavioral descriptors (Section 4.3), each of which highlights important research directions for the community. These analyses surface several insights, such as the
> differing strengths and limitations of RL for classification versus text generation tasks (L362–L374), model
> adaptation in pragmatic contexts (L413–L433), and the transferability from general behavioral datasets to
> previously unseen behavioral tasks (L401–L412), among others. In fact, this specific contribution has been
> acknowledged by reviewer aP6y, who mentioned that “The experimental setup is thorough, covering 10 in-domain datasets and some “transfer” datasets”. Much like analytical studies that advance understanding by
> clarifying the capabilities, limitations, and mechanisms of algorithms [10, 4, 1], our analyses offer empirical
> analysis that may help shape how the community frames unified behavioral models and identify promising
> directions for future exploration.
>
> The authors appreciate that the above contributions and novelty of the benchmark fall within the same
> scope of typically published papers in ICLR, ICML, Neurips, which offer a systematic and intellectually
> novel standardization and curation of datasets, alongside experimental analyses [3]. Therefore, we kindly
> appeal to the reviewer to consider these points.

---

> ### Author Response · Authors · 2025-11-22
> **Author's Response - Thank you for your feedback (part 3)**
>
> >  The models are fully fine-tuned on the same datasets used for evaluation. In the transfer-learning
> section, the model is again fine-tuned for a few epochs on the “held-out” datasets, and only Qwen 2.5-
> Omni-7B is used for comparison. This setup mainly measures fine-tuning efficiency, not true zero-shot
> generalization.
>
> We thank the reviewer for this thoughtful suggestion, and we appreciate the opportunity to clarify our experimental design. Our transfer experiments follow established practices in seminal transfer learning work [13, 8] , in which a model is pretrained on a large, diverse corpus (e.g., ImageNet) and subsequent fine-tuned on a specific downstream task under a fixed training budget. This setup has been utilized in the literature to study the effects of additional pretraining.
>
> However, that being said, we agree that evaluating zero-shot transfer offers important insight into generalization under a different setting. In response, and aligned with the reviewer’s suggestion, we conducted an additional held-out experiment to test zero-shot transfer. Specifically, we trained OmniSapiens-7B RL on
> Human Behavior Atlas while holding out MOSEI (SEN), MELD (EMO), DAIC-WOZ (DEP), and MUStARD (SAR) entirely from the training corpus. We then evaluated the model in a strict zero-shot setting
> on these held-out datasets. For comparison, we used Qwen 2.5-Omni-7B, which shares the same underlying architecture as OmniSapiens-7B RL but is not pretrained on Human Behavior Atlas, enabling a clean comparison of the contribution from pretraining.
>
> We present the results for this new experiment in the following table, and we will add this table and the
> associated discussion paragraph to Appendix D.1. We highlight our addition below:
>
> **Table: Zero-shot evaluation on held-out datasets.**
> Bold denotes best score.
>
> | **Dataset** | **OmniSapiens-7B RL** | Qwen 2.5-Omni-7B |
> |------------|------------------------|-------------------|
> | MOSEI (SEN)        | **0.247** | 0.201 |
> | MELD (EMO)         | **0.549** | 0.403 |
> | DAIC-WOZ (DEP)     | **0.499** | 0.108 |
> | MUStARD (SAR)      | **0.596** | 0.445 |
>
> _“From this table, we observe that pretraining on the Human Behavior Atlas benchmark yields substantial
> zero-shot gains on held-out datasets. OmniSapiens-7B RL outperforms Qwen 2.5-Omni-7B across all four
> datasets, with improvements of +22.99% (+0.046) on MOSEI (SEN), +36.2% (+0.146) on MELD (EMO),
> +362.04% (+0.391) on DAIC-WOZ (DEP), and +33.9% (+0.151) on MUStARD. These findings suggest that
> Human Behavior Atlas pretraining provides strong transferable representations across diverse behavioral
> tasks, under zero-shot settings.”_
>
> We hope this additional experiment addresses the reviewer’s suggestion on the provision of additional
> zero-shot experiments.
>
> > The construction of prompts is unclear. It’s not specified whether they were hand-crafted or automatically generated, nor whether prompt robustness was tested.
>
> We thank the reviewer for providing this helpful comment. All prompts in the benchmark were hand-crafted by our research team, and we will add this detail to the revised paper. To ensure that the benchmark overfitting on any single prompt format, each task or dataset comprises a different prompt, we provide examples in Table 9 (L1188-L1241).
>
> We also appreciate the chance to clarify the study of prompt robustness. While we agree that prompt
> sensitivity is an important topic for instruction-tuned models in general, it is not directly aligned with the
> scientific objective of this paper. Our goal is specifically to evaluate and compare models’ behavioral understanding capabilities under a fixed, shared instruction interface. Introducing multiple prompt variants
> would shift the evaluation toward a different research question, that of prompt robustness analysis. This
> would entangle two effects that should be isolated: (i) the model’s underlying capability, and (ii) its sensitivity to prompt perturbations. This additional axis of variation may unnecessarily complicate fair comparison. Differences in performance might then stem from prompt perturbations rather than the model’s genuine behavioral reasoning ability, which risks confusing the interpretation of results. For this reason, and consistent with standard practice in published multimodal and instruction-tuning benchmarks within top conferences [6, 5], we deliberately use a single consistent prompt per task to ensure clean, controlled, and interpretable model-to-model comparisons.
>
> >  How were prompts generated - hand-written or LLM-generated?
>
> We would like to thank the reviewer for this thoughtful point raised. We have responded this in the above comment.

---

> ### Author Response · Authors · 2025-11-22
> **Author's Response - Thank you for your feedback (part 4)**
>
> > In Table 4, can you add results using only the behavioral descriptors (e.g., OpenSMILE, MediaPipe)
> to show how much these features alone contribute?
>
> We thank the reviewer for this thoughtful feedback. We agree that, in addition to evaluating the full BAM configuration (which integrates behavioral descriptors alongside raw audio and visual modalities), it is also valuable to understand the standalone contribution of the behavioral descriptors themselves. In line with the reviewer’s suggestion, we conducted an additional ablation experiment to isolate the effect of the behavioral descriptors. This includes removing the raw audio and video features during the BAM model's training and inference.
>
> We incorporate this new experiment into Appendix D.3, along with an accompanying table and discussion. The results clarify an important point. Specifically, while behavioral descriptors do improve performance, their benefits are most effectively realized when they complement, rather than replace, raw audio and visual modalities.
>
> **Table: Δ highlights the change in performance from OmniSapiens-7B SFT to OmniSapiens-7B BAM and OmniSapiens-7B BAM (ABL).**
>
> | **Task** | **OmniSapiens-7B SFT** | **OmniSapiens-7B BAM** | **OmniSapiens-7B BAM (ABL)** | **Δ BAM % (Abs)** | **Δ ABL % (Abs)** |
> |---------|------------------------|-------------------------|------------------------------|--------------------|--------------------|
> | **NVC** | 0.12 | 0.16 | 0.06 | ▲ +33.00 (+0.04) | ▼ −47.11 (−0.06) |
> | **SAR** | 0.62 | 0.80 | 0.75 | ▲ +29.00 (+0.18) | ▲ +20.19 (+0.13) |
> | **HUM** | 0.53 | 0.64 | 0.64 | ▲ +21.00 (+0.11) | ▲ +21.05 (+0.11) |
> | **DEP** | 0.73 | 0.79 | 0.75 | ▲ +8.21 (+0.06) | ▲ +2.73 (+0.02) |
> | **EMO** | 0.63 | 0.65 | 0.59 | ▲ +3.17 (+0.02) | ▼ −6.66 (−0.04) |
> | **SEN** | 0.77 | 0.79 | 0.76 | ▲ +2.60 (+0.02) | ▼ −1.30 (−0.01) |
> | **PTSD** | 1.00 | 1.00 | 1.00 | 0.00 (+0.00) | 0.00 (+0.00) |
> | **ANX** | 0.91 | 0.91 | 0.91 | 0.00 (+0.00) | 0.00 (+0.00) |
> | **SOC** | 0.26 | 0.20 | 0.22 | ▼ −23.08 (−0.06) | ▼ −15.56 (−0.04) |
> | **INT** | 0.26 | 0.18 | 0.18 | ▼ −30.77 (−0.08) | ▼ −30.08 (−0.08) |
>
> We include the following analysis, “_To analyze the isolated contribution of behavioral descriptors, we
> conducted an ablation in which we removed the raw audio and video features during the BAM model’s
> training and inference and. The resulting model, which we term OmniSapiens-7B BAM (ABL), is evaluated
> across all tasks. We observe that removing the raw video and audio modalities leads to consistently weaker
> performance compared to the full BAM model, with OmniSapiens-7B BAM (ABL) underperforming BAM
> on most datasets. This suggests that behavioral descriptors provide complementary information to the raw
> audio and visual encoders, rather than serving as an effective standalone substitute. These findings offer
> practical guidance for deploying BAM within unified omni-modal architectures: behavioral descriptors are
> most beneficial when integrated jointly with raw multimodal signals._”
>
> We hope this additional experiment adequately addresses the reviewer’s concern on the integration of
> behavioral descriptors.

---

> ### Author Response · Authors · 2025-11-22
> **Author's Response - Thank you for your feedback (part 5)**
>
> > Since the models are fine-tuned on the same datasets used for evaluation, have you checked whether fine-tuning affects general-purpose abilities (e.g. text generation)?
>
> We thank the reviewer for the useful feedback. To assess whether fine-tuning on Human Behavior Atlas
> affects general-purpose text generation abilities, we conducted an additional experiment evaluating the natural language outputs of OmniSapiens-7B RL on the question-answering datasets within the benchmark
> (MimeQA, IntentQA, Social-IQ 2.0). These datasets offer a practical setting for examining text generation
> quality in the context most relevant to our work, that of unified psychological and social behavior understanding, rather than unrelated open-domain generation tasks.
> To ensure a clean comparison, we evaluate OmniSapiens-7B RL against the baseline Qwen 2.5-Omni-
> 7B model, which shares the same architecture but has not been fine-tuned. Following established evaluation
> protocols for general text generation [7, 14, 2], we assess four dimensions: (i) perplexity (via GPT-2), (ii)
> fluency, (iii) coherence of reasoning, and (iv) relevance for QA responses. The latter three metrics are judged
> using an LLM with a clearly defined 1–5 scoring rubric; we will include the prompts and scoring definitions
> in the appendix.
> We summarize the results in the table below and will add both the table and the following discussion to
> the appendix D.5.
>
> | Model                 | Fluency | Coherence | Relevance | Perplexity |
> |-----------------------|---------|-----------|-----------|------------|
> | Qwen 2.5-Omni-7B      | 3.682   | 3.562     | 3.365     | 542.676    |
> | OmniSapiens-7B RL     | 3.698   | 3.566     | 3.375     | 542.676    |
>
> We add the following, “_We evaluate the free-form text responses generated by OmniSapiens-7B RL and by Qwen 2.5-Omni-
> 7B, which shares the same architecture but lacks the additional Human Behavior Atlas training, across all
> question-answering tasks (IntentQA, MimeQA, Social-IQ 2.0) to analyze whether fine-tuning on Human
> Behavior Atlas influences general text generation quality. This involves computing perplexity over the generated outputs, and then evaluating fluency, coherence, and relevance using LLM-based scoring, following
> from established evaluation protocols for general text generation [7, 14, 2]. Across the four metrics, we
> observe no meaningful differences between Qwen 2.5-Omni-7B and OmniSapiens-7B RL. These findings
> indicate that continued training on Human Behavior Atlas does not degrade the underlying text generation
> ability of the model._”
>
> We hope this additional experiment addresses the reviewers suggestion on evaluating the text generation
> capabilities of the model after fine-tuning.
>
> > Did you perform any true zero-shot evaluations to measure actual generalization?
>
> We thank the reviewer for the comment. We have added experiments for this, as indicated in the above
> response.
>
> > In Table 5, it would be interesting to add results before fine-tuning on the held-out datasets and also
> show results for the multimodal LLMs used in Table 4 for comparison.
>
> We thank the reviewer for the thoughtful comment. In response, we have updated Table 5 to include the
> performance of the requested models prior to fine-tuning, enabling a clearer comparison. The revised table
> is provided below and will be added to the Appendix D.6.
>
> **Table: Transfer to held-out datasets after minimal epoch fine-tuning (1 epoch).**
> Bold denotes best score. DAIC-WOZ\* uses 2 epochs due to only 107 training samples.
> MUStARD† presents a novel behavioral task (sarcasm).
> ‡ Other held-out datasets with tasks represented during pretraining.
>
> | Dataset | OmniSapiens-7B SFT | Qwen 2.5-Omni-7B SFT | Gemma-3-4B | HumanOmniV2-7B | Qwen 2.5-Omni-7B | Qwen-2.5-VL-7B |
> |---------|---------------------|------------------------|-------------|------------------|-------------------|-----------------|
> | MOSEI‡ (SEN) | **0.724** | 0.612 | 0.617 | 0.633 | 0.602 | 0.317 |
> | MELD‡ (EMO) | **0.711** | 0.684 | 0.642 | 0.633 | 0.661 | 0.571 |
> | DAIC-WOZ*‡ (DEP) | **0.749** | 0.579 | 0.137 | 0.636 | 0.636 | 0.623 |
> | MUStARD† (SAR) | **0.658** | 0.473 | 0.529 | 0.395 | 0.656 | 0.511 |

---

> ### Author Response · Authors · 2025-11-22
> **Author's Response - Thank you for your feedback (part 6)**
>
> **References**
>
> ---
>
> [1] Mostafa M Amin, Rui Mao, Erik Cambria, and Bj¨orn W Schuller. A wide evaluation of chatgpt on
> affective computing tasks. IEEE Transactions on Affective Computing, 15(4):2204–2212, 2024.
>
> [2] Yi Chen, Rui Wang, Haiyun Jiang, Shuming Shi, and Ruifeng Xu. Exploring the use of large
> language models for reference-free text quality evaluation: An empirical study. arXiv preprint
> arXiv:2304.00723, 2023.
>
> [3] Wei Dai, Peilin Chen, Malinda Lu, Daniel Li, Haowen Wei, Hejie Cui, and Paul Pu Liang. Climb: Data
> foundations for large scale multimodal clinical foundation models. arXiv preprint arXiv:2503.07667,
> 2025.
>
> [4] Chanakya Ekbote, Marco Bondaschi, Nived Rajaraman, Jason D Lee, Michael Gastpar, Ashok Vardhan
> Makkuva, and Paul Pu Liang. What one cannot, two can: Two-layer transformers provably represent
> induction heads on any-order markov chains. arXiv preprint arXiv:2508.07208, 2025.
>
> [5] Hengzhi Li, Megan Tjandrasuwita, Yi R Fung, Armando Solar-Lezama, and Paul Pu Liang. Mimeqa:
> Towards socially-intelligent nonverbal foundation models. arXiv preprint arXiv:2502.16671, 2025.
>
> [6] Paul Pu Liang, Akshay Goindani, Talha Chafekar, Leena Mathur, Haofei Yu, Ruslan Salakhutdinov,
> and Louis-Philippe Morency. Hemm: Holistic evaluation of multimodal foundation models. Advances
> in Neural Information Processing Systems, 37:42899–42940, 2024.
>
> [7] Alec Radford, Jeffrey Wu, Rewon Child, David Luan, Dario Amodei, Ilya Sutskever, et al. Language
> models are unsupervised multitask learners. OpenAI blog, 1(8):9, 2019.
>
> [8] Colin Raffel, Noam Shazeer, Adam Roberts, Katherine Lee, Sharan Narang, Michael Matena, Yanqi
> Zhou, Wei Li, and Peter J Liu. Exploring the limits of transfer learning with a unified text-to-text
> transformer. Journal of machine learning research, 21(140):1–67, 2020.
>
> [9] Trapti Shrivastava, Vrijendra Singh, and Anupam Agrawal. Autism spectrum disorder detection with
> knn imputer and machine learning classifiers via questionnaire mode of screening. Health Information
> Science and Systems, 12(1):18, 2024.
>
> [10] Megan Tjandrasuwita, Chanakya Ekbote, Liu Ziyin, and Paul Pu Liang. Understanding the emergence
> of multimodal representation alignment. arXiv preprint arXiv:2502.16282, 2025.
>
> [11] Alex Wang, Yada Pruksachatkun, Nikita Nangia, Amanpreet Singh, Julian Michael, Felix Hill, Omer
> Levy, and Samuel Bowman. Superglue: A stickier benchmark for general-purpose language under-
> standing systems. Advances in neural information processing systems, 32, 2019.
>
> [12] Alex Wang, Amanpreet Singh, Julian Michael, Felix Hill, Omer Levy, and Samuel Bowman. Glue: A
> multi-task benchmark and analysis platform for natural language understanding. In Proceedings of the
> 2018 EMNLP workshop BlackboxNLP: Analyzing and interpreting neural networks for NLP, pages
> 353–355, 2018.
>
> [13] Jason Yosinski, Jeff Clune, Yoshua Bengio, and Hod Lipson. How transferable are features in deep
> neural networks? Advances in neural information processing systems, 27, 2014.
>
> [14] Lianmin Zheng, Wei-Lin Chiang, Ying Sheng, Siyuan Zhuang, Zhanghao Wu, Yonghao Zhuang,
> Zi Lin, Zhuohan Li, Dacheng Li, Eric Xing, et al. Judging llm-as-a-judge with mt-bench and chatbot
> arena. Advances in Neural Information Processing Systems, 36:46595–46623, 2023.

---

> ### Author Response · Authors · 2025-12-02
> **Author's Response - Thank you for your feedback (part 7)**
>
> We have also completed our experiment on hyperparameters, including rollout size, learning rate and training method. Please find the results below:
>
> Best results are bolded. We report binary weighted F1 for SEN; mean per-class weighted accuracy for EMO; weighted F1 for HUM, SAR, ANX, DEP, PTSD; and LLM-Judge accuracy for SOC, INT, NVC.
>
> | Model | CREMA-D | MELD (E) | MOSEI (E) | TESS | UR-FUNNY | IntentQA | PTSD_WILD | MMPSY (A) | MMPSY (D) | MELD (S) | CH-SIMSv2 | MOSEI (S) | MUStARD | Social-IQ 2.0 | MimeQA |
> |-------|---------|----------|-----------|------|----------|----------|-----------|-----------|-----------|----------|-----------|-----------|---------|---------------|--------|
> | | **EMO** | | | | **HUM** | **INT** | **PTSD** | **ANX** | **DEP** | **SEN** | | | **SAR** | **SOC** | **NVC** |
> | **Number of Rollouts (lr = 5e-7)** | | | | | | | | | | | | | | | |
> | Rollouts = 2 | .500 | .563 | .576 | .499 | **.643** | .422 | **.984** | .669 | .700 | **.588** | .383 | **.253** | **.670** | .168 | .162 |
> | Rollouts = 5 | **.501** | **.699** | **.581** | **.510** | .639 | **.486** | .968 | **.919** | .814 | .571 | .393 | .224 | .647 | **.304** | .133 |
> | Rollouts = 20 | .495 | .572 | **.581** | .507 | .600 | .448 | **.984** | .882 | **.841** | .577 | **.403** | .252 | **.670** | .179 | **.178** |
> | **Learning Rate (rollouts = 5)** | | | | | | | | | | | | | | | |
> | lr = 1e-6 | .496 | .531 | **.588** | .500 | **.640** | .408 | **.968** | .851 | **.843** | .574 | .371 | .224 | .640 | .161 | .150 |
> | lr = 5e-7 | **.501** | **.699** | .581 | **.510** | .639 | **.486** | **.968** | **.919** | .814 | .571 | **.393** | **.224** | **.647** | **.304** | .133 |
> | lr = 1e-7 | **.501** | .521 | .577 | .508 | .636 | .399 | .937 | .755 | .702 | **.577** | .370 | .221 | .580 | .161 | **.174** |
> | **Training Method (rollouts = 5, lr = 5e-7)** | | | | | | | | | | | | | | | |
> | RLOO | **.501** | .585 | .573 | .499 | .613 | .436 | **.984** | .877 | **.852** | **.617** | **.404** | **.276** | **.675** | .173 | **.170** |
> | GRPO | **.501** | **.699** | **.581** | **.510** | **.639** | **.486** | .968 | **.919** | .814 | .571 | .393 | .224 | .647 | **.304** | .133 |
>
> In general, we see a learning rate of 5e-7 and rollout size of 5 to be a sweet spot for training. Performance difference tends to be small, as training runs with different hyperparameters usually converges to similar performance at different speeds.

---

### Official Review · Reviewer_PK4q · 2025-11-03

**Soundness:** 2
**Presentation:** 2
**Contribution:** 2
**Rating:** 4
**Confidence:** 3

**Summary:**

This paper introduces HUMAN BEHAVIOR ATLAS, a large-scale multimodal benchmark with numerous tasks for general psychological and social behavior understanding. The core data contribution is curation and standardization of existing public datasets. Three variants of OMNISAPIENS-7B are evaluated to show the effectiveness of multi task training and transfer learning.

**Strengths:**

* Benchmark, source code, models will be released.
* Atlas is a large-scale multimodal benchmark with numerous tasks for general psychological and social behavior understanding.
* Paper is well-written, and easy to follow.
* Promising performances for multi-task training and transfer learning are shown.

**Weaknesses:**

* My major concern is the limited contributions
  * The paper does not propose a new model architecture.
  * The core data contribution is curation and standardization of existing public datasets instead of new data collection.
* The comparison in Table 4 is unfair because the OMNISAPIENS-7B variants were trained directly on the HUMAN BEHAVIOR ATLAS data, while the general multimodal LLM baselines were evaluated in zero-shot inference mode without fine-tuning on this specific benchmark. The performance gain largely reflects the benefit of SFT or RL on the target tasks, not necessarily the inherent superiority of the model's architecture or the benchmark itself.
* To demonstrate the superiority of the ATLAS for unified modeling, the authors should compare OMNISAPIENS-7B variants to models trained exclusively on another large, existing social behavior or affective computing dataset (e.g., a comprehensive version of CMU-MOSEI, MELD, or a large synthesized dataset like HumanOmni) and then test all models across the full range of ATLAS tasks.
* A similar issue in the transfer learning experiment in Section 4.2 and Table 5.

**Questions:**

Please refer to the weaknesses section.

---

> ### Author Response · Authors · 2025-11-14
> **Author's Response - Thank you for your feedback (part 1)**
>
> We would like to thank the reviewer for providing the useful feedback. The authors are happy to address the feedback provided to improve the paper’s quality. Please find our attached responses below.
>
> > My major concern is the limited contributions. The paper does not propose a new model architecture.
> The core data contribution is curation and standardization of existing public datasets instead of new data
> collection.
>
> We thank the reviewer for this thoughtful comment, and appreciate the opportunity to clarify our paper’s contributions in light of the concerns raised. Specifically, we would like to clarify that the main contribution of this paper is not novel model architectures, as noted in our introductory paragraph (L85-L87). With regard to the core contributions, we would like to qualify them below:
> 1. **Regarding the addition of new data, the benchmark comprises new standardized behavioral descriptors (transcriptions, mediapipe, opensmile), as well as new hand-crafted instruction prompt sets. We believe these are new data contributions that are significant and non-trivial to collect.** These new data crucially enable unified model training, and were not previously present in the existing public datasets. Therefore, the paper extends curation and
> standardization of existing public datasets to also involve these elements of data creation, which we believe will be a valuable resource for the community. Moreover, these efforts are resource intensive and time consuming to realize, making them relatively inaccessible for a considerable portion of the community to realize (as we elaborate in 4.).
> 2. **Regardless, we believe the benchmark’s intellectual novelty is independent from new data creation, and this intellectual contribution lies in the benchmark’s new taxonomy, task formulation, and experimental framework for AI for psychological and social behavior understanding.** This follows precedent from top published foundational benchmark
> papers in ICML, NeurIPS, and ICLR that did not involve new data creation. Our contribution intends to follow
> the spirit of these influential works, whose impact stemmed from introducing unifying conceptual frameworks and standardized evaluation paradigms that offered important intellectual insights through thoughtful benchmark design, such as systematic data collection, principled task formulation, and coherent integration of existing datasets, rather than through the construction of new data. Notable examples include GLUE [7] and SuperGLUE [6], both foundational benchmarks paper built entirely from already existing public datasets to evaluate general natural language understanding, among many other influential work. In a similar spirit, our benchmark introduces a conceptual framework that standardizes and unifies public behavioral datasets, to enable systematic evaluation, generalization analysis, and unified model development within the field of AI for psychological and social behavior understanding. In light of these precedents, we kindly appeal that the contribution of Human Behavior Atlas be evaluated based on its conceptual advances, i.e., its behavioral taxonomy, formulation of tasks, experimental framework for unified behavioral AI models in the context of generalizability and multi-task performance. We remain open to further feedback on how these contributions are positioned within this field of AI.
> 3. **We believe the benchmark has significant potential to advance the field by enabling unified models in AI
> for psychological and social behavior understanding: The Human Behavior Atlas benchmark provides
> the foundational infrastructure for the community to train unified models of human behavior understanding.** As elaborated
> in point 4., this endeavour would otherwise be inaccessible without a homogeneous and standardized benchmark, which can be both time-consuming and resource-intensive to curate. Furthermore, as discussed in the introduction (L33–L37), in a field still dominated by single-task models and bespoke architectures, this benchmark has the potential to fundamentally shift the paradigm toward more generalizable and efficient architectures that support scalability and cross-task transfer. Such progress could direcly address significant challenges in data-scarce behavioral domains where cross-dataset transfer is valuable; for example, enhancing model development for autism detection [4]. As such, we contend that the benchmark could have considerable contributions in this AI field.

---

> ### Author Response · Authors · 2025-11-14
> **Author's Response - Thank you for your feedback (part 2)**
>
> > My major concern is the limited contributions. The paper does not propose a new model architecture.
> The core data contribution is curation and standardization of existing public datasets instead of new data collection.
>
> **(continued author's response)**
>
> 4. **Homogenizing and standardizing datasets into a benchmark is resource-intensive, rendering it
> largely inaccessible for the community**: Curating and standardizing public datasets within a conceptually coherent and homogeneous framework demands substantial effort and time, and cannot be achieved within a short period or with minimal resources. As detailed in our paper, this process involves developing a behavioral taxonomy (affect, cognition, pathology, social) to guide dataset collection (L164-L171), hand-crafting new instruction prompts for each of the 100k samples and 13 datasets (i.e. Table 3, L195-L208), extracting key behavioral features (i.e. pose, prosody, text transcriptions) (L241-L251), with all of these efforts realized over an exceedingly large amount of multimodal data (100k samples, 35k videos, 10k audio clips). Given the scope and resource demands of these efforts, such standardization poses substantial barriers for most researchers seeking to train unified models for understanding human psychological and social behaviors; as evidenced by the fact that there have been limited or no unified models developed for this field. By making these curated resources publicly available, our benchmark lowers these barriers considerably, making it more accessible for the community to develop and evaluate unified behavioral AI models.
>
> At this juncture, we also note that other reviewers have recognized the benchmark’s contribution in this regard. For example, Reviewer eg78 highlighted the strength of providing a “unified benchmark with broad task coverage and standardized metrics” and Reviewer tGVo mentioned that “The authors successfully harmonize heterogeneous datasets and tasks (e.g., emotion recognition, personality, mental health indicators) under a single large language model (LLM)-based paradigm, which demonstrates strong potential for generalization across domains”. With these perspectives in mind, we kindly appeal that the reviewer may consider these facets of the work's contribution.
>
> Additionally, we would like to emphasize that the contribution of this work extends beyond dataset standardization and curation. A central component of the paper lies in its analytical experiments and accompanying discussions. These include, multi-task evaluation (Section 4.1), transfer analysis (Section 4.2), and the examination of non-invasive integration of behavioral descriptors (Section 4.3), each of which highlights important research directions for the community. These analyses surface several insights, such as the differing strengths and limitations of RL for classification versus text generation tasks (L362–L374), model adaptation in pragmatic contexts (L413–L433), and the transferability from general behavioral datasets to previously unseen behavioral tasks (L401–L412), among others. In fact, this specific contribution has been acknowledged by reviewer aP6y, who mentioned that ”The experimental setup is thorough, covering 10 in-domain datasets and some "transfer” datasets”. Much like analytical studies that advance understanding by clarifying the capabilities, limitations, and mechanisms of algorithms [5, 3, 1], our analyses offer empirical analysis that may help shape how the community frames unified behavioral models and identify promising directions for future exploration.
>
> The authors appreciate that the above contributions of the benchmark fall within the same scope of
> typically published papers in ICLR, ICML, Neurips, which offer a systematic and intellectually novel standardization of datasets, alongside experimental analyses [2]. Therefore, we kindly appeal to the reviewer to consider these points.

---

> ### Author Response · Authors · 2025-11-14
> **Author's Response - Thank you for your feedback (part 3)**
>
> > The comparison in Table 4 is unfair because the OMNISAPIENS-7B variants were trained directly on
> the HUMAN BEHAVIOR ATLAS data, while the general multimodal LLM baselines were evaluated in
> zero-shot inference mode without fine-tuning on this specific benchmark. The performance gain largely
> reflects the benefit of SFT or RL on the target tasks, not necessarily the inherent superiority of the
> model’s architecture or the benchmark itself.
>
> We would like to thank the reviewer for rasing this concern, and fully appreciate the importance of fair
> benchmarking between models and datasets. We would like to qualify that we are evaluating the general multimodal LLM baselines to primarily provide insight into how current multimodal LLMs (i.e. trained on general multimodal data), perform on various psychological and social behavior tasks. This experiment highlights the limitations of current general LLMs on diverse behavioral tasks, and demonstrates that further training on the benchmark (i.e. via SFT or RL) results in performance improvements on these tasks. These empirical findings are not meant to illustrate the advantage of any model architecture. Instead, they serve to qualify the utility of the Human Behavior Atlas benchmark, as a training resource that can enable further specialization of general multimodal LLMs to behavioral domains (L321-L323). To avoid any ambiguity for readers, we will clarify this intention in the paragraph (L316-L323) by including the following line “this experiment is primarily meant to highlight the utility of further training on Human Behavior Atlas to improve the capabilities of current multimodal LLMs on diverse behavioral tasks”. We hope that this explanation clarifies the motivation behind the experiment, and are happy to provide further clarification if necessary.
>
> > To demonstrate the superiority of the ATLAS for unified modeling, the authors should compare
> OMNISAPIENS-7B variants to models trained exclusively on another large, existing social behavior
> or affective computing dataset (e.g., a comprehensive version of CMU-MOSEI, MELD, or a large syn-
> thesized dataset like HumanOmni) and then test all models across the full range of ATLAS tasks. A similar issue in the transfer learning experiment in Section 4.2 and Table 5.
>
> We thank the reviewer for this comment. To test the utility of our benchmark, we acknowledge the importance of comparing the results of a model trained on Human Behavior Atlas (Omnisapiens 7B), against the results of a model trained on other large scale human-behavior related data (i.e. HumanOmni). In fact, as per the reviewer's suggestion, we actually evaluated HumanOmniV2-7B (one of the few unified models trained on large scale human-related synthetic data (HumanOmni), beyond any single dataset like MELD or CMU-MOSEI), on our Human Behavior Atlas benchmark (results added in Table 4, L324-L339). We apologize if this was not clear in our manuscript, and will add the following line to the paragraph in L341-L361 to clarify, "To compare Human Behavior Atlas against other large scale human-related benchmarks (i.e., HumanOmni), we evaluate models trained on these related benchmarks (i.e., HumanOmniv2-7B) against OmniSapiens."
>
> We also discussed the results attained by the HumanOmni model in the paragraph (L341-L361), with our insight being that models such as these, though pretrained on large behavior related data, remain focused on narrow behavioral phenomena. Therefore, these models, and the datasets that they are trained on do not sufficiently address unified modelling on a broad range of psychological and social behaviors. At the minimum, this clarifies the research gap that Human Behavior Atlas aspires to address. That is, the lack of unified, generalizable benchmarks and models that can address a wide suite of behavioral tasks, to shift the field toward unified models from inefficient bespoke and single task datasets and architectures. We hope this clarifies experiments with respect to these specialized models.

---

> ### Author Response · Authors · 2025-11-14
> **Author's Response - Thank you for your feedback (part 4)**
>
> ---
>
> **References**
>
> [1] Mostafa M Amin, Rui Mao, Erik Cambria, and Bj¨orn W Schuller. A wide evaluation of chatgpt on
> affective computing tasks. IEEE Transactions on Affective Computing, 15(4):2204–2212, 2024.
>
> [2] Wei Dai, Peilin Chen, Malinda Lu, Daniel Li, Haowen Wei, Hejie Cui, and Paul Pu Liang. Climb: Data
> foundations for large scale multimodal clinical foundation models. arXiv preprint arXiv:2503.07667,
> 2025.
>
> [3] Chanakya Ekbote, Marco Bondaschi, Nived Rajaraman, Jason D Lee, Michael Gastpar, Ashok Vardhan
> Makkuva, and Paul Pu Liang. What one cannot, two can: Two-layer transformers provably represent
> induction heads on any-order markov chains. arXiv preprint arXiv:2508.07208, 2025.
>
> [4] Trapti Shrivastava, Vrijendra Singh, and Anupam Agrawal. Autism spectrum disorder detection with
> knn imputer and machine learning classifiers via questionnaire mode of screening. Health Information
> Science and Systems, 12(1):18, 2024.
>
> [5] Megan Tjandrasuwita, Chanakya Ekbote, Liu Ziyin, and Paul Pu Liang. Understanding the emergence
> of multimodal representation alignment. arXiv preprint arXiv:2502.16282, 2025.
>
> [6] Alex Wang, Yada Pruksachatkun, Nikita Nangia, Amanpreet Singh, Julian Michael, Felix Hill, Omer
> Levy, and Samuel Bowman. Superglue: A stickier benchmark for general-purpose language under-
> standing systems. Advances in neural information processing systems, 32, 2019.
>
> [7] Alex Wang, Amanpreet Singh, Julian Michael, Felix Hill, Omer Levy, and Samuel Bowman. Glue:
> A multi-task benchmark and analysis platform for natural language understanding. In Proceedings of
> the 2018 EMNLP workshop BlackboxNLP: Analyzing and interpreting neural networks for NLP, pages
> 353–355, 2018

---

### Author Response · Authors · 2025-11-22
**(Part 1) Update from Authors - Added new experiments and analyses according to reviewers' suggestions. Looking forward to engaging reviewers**

Dear Reviewers,

We thank you for the time invested in the reviewer process, and appreciate the thoughtful feedback provided. In our response to the feedback, and in line with the suggestions provided, we have added a considerable number of new experiments and analysis to improve the paper (found in appendix D), and would like to highlight them below:

1. **Paper's contribution**: We would like to clarify that the paper's core contribution is that of benchmark curation and empirical experimental analysis. The research team has invested a considerable amount of time hand-crafting new instruction prompt sets, behavioral descriptors, developing a new taxonomy and framework for the curation of datasets, alongside numerous experimental analyses. In a field that is dominated by single task models and datasets, our benchmark may provide the first step for the community to shift toward unified architectures, which can possess significant benefits for transfer in data scarce regimes, scalability and efficiency.
2. **Addition of Zero-shot Transfer Evaluation**: We have also added an experiment for zero-shot transfer performance, to complement our existing transfer experiments on fine-tuning with fixed epoch budget. We observe from these results that pretraining on Human Behavior Atlas does results in higher zero-shot performance on heldout datasets, with the baseline for comparison being Qwen 2.5-Omni-7B which possesses the same architecture, but with the absence of pretraining on the benchmark.\
**Table: Zero-shot evaluation on held-out datasets. Bold denotes best score.**
| **Dataset** | **OmniSapiens-7B RL** | Qwen 2.5-Omni-7B |
|------------|------------------------|-------------------|
| MOSEI (SEN)        | **0.247** | 0.201 |
| MELD (EMO)         | **0.549** | 0.403 |
| DAIC-WOZ (DEP)     | **0.499** | 0.108 |
| MUStARD (SAR)      | **0.596** | 0.445 |

3. **Addition of ablation experiments on BAM, to understand the impact of training with solely the behavioral descriptors**: We conducted an additional experiment for training, where we ablated the raw vision and audio modalities, leaving only the integration of the behavioral descriptors. We observed that this results in lower performance, highlighting that integrating the behavioral descriptors through BAM is most effective when the raw vision and audio features are present. This shows that behavioral descriptors are best incorporated when done to complement rather than replace the raw audio and visual modalities.\
**Table: Δ highlights the change in performance from OmniSapiens-7B SFT to OmniSapiens-7B BAM and OmniSapiens-7B BAM (ABL).**
| **Task** | **OmniSapiens-7B SFT** | **OmniSapiens-7B BAM** | **OmniSapiens-7B BAM (ABL)** | **Δ BAM % (Abs)** | **Δ ABL % (Abs)** |
|---------|------------------------|-------------------------|------------------------------|--------------------|--------------------|
| **NVC** | 0.12 | 0.16 | 0.06 | ▲ +33.00 (+0.04) | ▼ −47.11 (−0.06) |
| **SAR** | 0.62 | 0.80 | 0.75 | ▲ +29.00 (+0.18) | ▲ +20.19 (+0.13) |
| **HUM** | 0.53 | 0.64 | 0.64 | ▲ +21.00 (+0.11) | ▲ +21.05 (+0.11) |
| **DEP** | 0.73 | 0.79 | 0.75 | ▲ +8.21 (+0.06) | ▲ +2.73 (+0.02) |
| **EMO** | 0.63 | 0.65 | 0.59 | ▲ +3.17 (+0.02) | ▼ −6.66 (−0.04) |
| **SEN** | 0.77 | 0.79 | 0.76 | ▲ +2.60 (+0.02) | ▼ −1.30 (−0.01) |
| **PTSD** | 1.00 | 1.00 | 1.00 | 0.00 (+0.00) | 0.00 (+0.00) |
| **ANX** | 0.91 | 0.91 | 0.91 | 0.00 (+0.00) | 0.00 (+0.00) |
| **SOC** | 0.26 | 0.20 | 0.22 | ▼ −23.08 (−0.06) | ▼ −15.56 (−0.04) |
| **INT** | 0.26 | 0.18 | 0.18 | ▼ −30.77 (−0.08) | ▼ −30.08 (−0.08) |
4. **Additional experiment to evaluate whether text generation quality is retained after training on Human Behavior Atlas**: We evaluated the free-text responses of the OmniSapiens-7B RL model against the baseline Qwen 2.5-Omni-7B model, for the QA sets of MimeQA, IntentQa, Social-IQ 2.0, against the metrics of perplexity, fluency, coherence of reasoning, and relevance of answer responses. Our results highlight that training on Human Behavior Atlas does not decrease text generation quality.
| Model                 | Fluency | Coherence | Relevance | Perplexity |
|-----------------------|---------|-----------|-----------|------------|
| Qwen 2.5-Omni-7B      | 3.682   | 3.562     | 3.365     | 542.676    |
| OmniSapiens-7B RL     | 3.698   | 3.566     | 3.375     | 542.676    |

---

> ### Author Response · Authors · 2025-11-22
> **(Part 2) Update from Authors - Added new experiments and analyses according to reviewers' suggestions. Looking forward to engaging reviewers**
>
> **(Continued from above part 1)**
>
> 5. **Added empirical analysis on effects of dataset imbalance in benchmark**: We also conducted an additional analysis on whether data imbalance within the benchmark affects performance during multi-task training. From our results, we observed that dataset frequency does not have a positive monotonic relationship with task performance, with the spearman rank correlations being negative in sign, with weak or non-existent magnitudes. In other words, having more samples for a given dataset does not systematically correspond to better performance on that dataset. These results suggest that the performance disparities across datasets are more likely driven by differences in task difficulty and complexity rather than by dataset imbalance within the benchmark.\
> **Table: Analysis of Dataset Sample Frequency versus Results**
> | Dataset | Sample Count | Sample Count Rank | SFT Score | SFT Rank | RL Score | RL Rank | BAM Score | BAM Rank |
> |--------|--------------|-------------------|-----------|----------|----------|---------|-----------|----------|
> | MOSEI (SEN) | 31453 | 1 | 0.744 | 6 | 0.224 | 15 | 0.775 | 6 |
> | IntentQA (INT) | 16297 | 2 | 0.256 | 15 | 0.486 | 12 | 0.177 | 15 |
> | MELD (EMO) | 13706 | 3 | 0.709 | 7 | 0.699 | 5 | 0.711 | 10 |
> | MELD (SEN) | 13706 | 3 | 0.746 | 5 | 0.571 | 9 | 0.744 | 7 |
> | MOSEI (EMO) | 8598 | 5 | 0.614 | 11 | 0.581 | 8 | 0.607 | 12 |
> | CREMA-D (EMO) | 7442 | 6 | 0.542 | 12 | 0.501 | 11 | 0.548 | 13 |
> | Social-IQ (SOC) | 6437 | 7 | 0.257 | 14 | 0.304 | 14 | 0.201 | 14 |
> | CH-SIMSv2 (SEN) | 4403 | 8 | 0.813 | 4 | 0.393 | 13 | 0.837 | 4 |
> | TESS (EMO) | 2800 | 9 | 0.658 | 8 | 0.51 | 10 | 0.715 | 9 |
> | UR-FUNNY (HUM) | 2125 | 10 | 0.532 | 13 | 0.639 | 7 | 0.644 | 11 |
> | MMPsy (ANX) | 1275 | 11 | 0.909 | 2 | 0.919 | 2 | 0.909 | 2 |
> | MMPsy (DEP) | 1275 | 11 | 0.839 | 3 | 0.814 | 3 | 0.839 | 3 |
> | MimeQA (NVC) | 806 | 13 | 0.121 | 16 | 0.133 | 16 | 0.162 | 16 |
> | MUStARD (SAR) | 690 | 14 | 0.624 | 10 | 0.647 | 6 | 0.795 | 5 |
> | PTSD-in-the-Wild (PTSD) | 634 | 15 | 1.000 | 1 | 0.968 | 1 | 1.000 | 1 |
> | DAIC-WOZ (DEP) | 189 | 16 | 0.626 | 9 | 0.729 | 4 | 0.738 | 8 | \
> **Table: Spearman Correlations of Performance Against Sample Counts**
> | Metric | Value |
> | Spearman (SFT) | -0.134554126 |
> | Spearman (RL) | -0.47971471 |
> | Spearman (BAM) | -0.342235494 |
> 6. **Addtional experiment on Cross-Dataset Transfer:** We conducted an additional transfer experiment on a large and diverse dataset to provide a more complete picture of cross-dataset generalization. Accordingly, we evaluated the performance of the OmniSapiens-7B SFT model (pretrained on Human Behavior Atlas), on IEMOCAP, using a minimum fine-tuning budget of one epoch, following established protocols to evaluate held out generalization in foundational papers. IEMOCAP is a large emotion-recognition benchmark that differs in annotation, context, domain from the emotion recognition datasets in Human Behavior Atlas, allowing us to further test the robustness of transfer performance gained by training on Human Behavior Atlas. Accordingly, we observe an +10.5% improvement (+0.059) for OmniSapiens-7B SFT (with Human Behavior Atlas pretraining) over Qwen2.5-Omni-7B (without Human Behavior Atlas pretraining). This offers an encouraging indication for transfer to new affective or emotion-recognition benchmarks, where contextual and annotation variability often limits the generalization performance of models trained on existing emotion datasets.\
> **Table. Transfer to IEMOCAP after Minimal Fine-Tuning (1 Epoch)**
> | Dataset  | OmniSapiens-7B SFT | Qwen 2.5-Omni-7B |
> |----------|---------------------|-------------------|
> | **IEMOCAP** | **0.6213**           | 0.5625            |
>
>
> As highlighted above, we have obtained the experimental results and have written the additional discussion points. We have submitted the revised version of the paper, with the additional experiments in Appendix D. We are looking forward to further discussions with the reviewers, with respect to the additional improvements to the paper.

---

> > ### Author Response · Authors · 2025-11-26
> > **(Part 3) Update from Authors - Added new experiments and analyses according to reviewers' suggestions. Looking forward to engaging reviewers**
> >
> > Dear Reviewers,
> >
> > We thank you once again for investing time and effort into the reviewing process. As we await your responses, we would like to point out the additional experiments that we have ran over the weekend, in addition to the first batch of experiments we ran. These include additional ablations on BAM, computational studies, and the robustness of LLM-judge evaluation. We summarize these new experiments below, please find the additional experiments in Appendix D.:
> >
> > 7. **Additional experiments on increasing the hidden dimensions for BAM**: We increased the dimensions of BAM from 256 to 512 during training and inference of BAM, to study the difference in performance gains. We observed from the following results (below table) that increasing the hidden dimension does not necessarily yield better performance: 6 out of 10 tasks show noticeably worse performance, while only 2 out of 10 tasks improve. This highlights the dimension of 256 as a more effective balance between efficiency and accuracy.
> > **Table: Δ indicates the performance change from OmniSapiens-7B SFT to BAM (hidden dimension of 256) and BAM (hidden dimension of 512), shown as percentage (%) and absolute (Abs).**
> > | **Task** | **SFT** | **BAM (256)** | **BAM (512)** | **Δ BAM (256)** | **Δ BAM (512)** |
> > |---------|---------|----------------|----------------|------------------|------------------|
> > | **NVC** | 0.12 | 0.16 | 0.11 | +33.00% (+0.04) | −9.09% (−0.01) |
> > | **SAR** | 0.62 | 0.80 | 0.76 | +29.00% (+0.18) | +22.58% (+0.14) |
> > | **HUM** | 0.53 | 0.64 | 0.67 | +21.00% (+0.11) | +25.94% (+0.14) |
> > | **DEP** | 0.73 | 0.79 | 0.77 | +8.21% (+0.06) | +5.48% (+0.04) |
> > | **EMO** | 0.63 | 0.65 | 0.64 | +3.17% (+0.02) | +1.59% (+0.01) |
> > | **SEN** | 0.77 | 0.79 | 0.77 | +2.60% (+0.02) | 0.00% (+0.00) |
> > | **PTSD** | 1.00 | 1.00 | 1.00 | 0.00% (+0.00) | 0.00% (+0.00) |
> > | **ANX** | 0.91 | 0.91 | 0.91 | 0.00% (+0.00) | 0.00% (+0.00) |
> > | **SOC** | 0.26 | 0.20 | 0.15 | −23.08% (−0.06) | −41.63% (−0.11) |
> > | **INT** | 0.26 | 0.18 | 0.22 | −30.77% (−0.08) | −14.06% (−0.04) |
> > 8. **Additional experiment to analyze the contribution of each acoustic and visual behavioral descriptor stream:** We independently ablate the acoustic and visual descriptors during BAM training and inference. As shown in the following table, the results highlight that while integrating behavioral features is generally beneficial, the synergistic effects of combining acoustic and visual descriptors are task-dependent, with specific tasks such as EMO and SEN exhibiting clear gains when the streams are jointly used. This reinforces the motivation for BAM’s lightweight and computationally efficient design (two layers with 256 hidden dimensions). Namely, it enables practitioners to selectively discard a behavioral descriptor stream (Vis/ Audio) and retrain BAM when a particular descriptor stream does not provide meaningful benefit.\
> > **Table: Δ indicates the change from OmniSapiens-7B SFT to each BAM variant (Aud+Vis, Aud-only, Vis-only), shown as percentage (%) and absolute (Abs). “--” denotes unavailable comparisons due to missing modalities.**
> > | **Task** | **SFT** | **BAM (Aud+Vis)** | **BAM (Aud)** | **BAM (Vis)** | **Δ BAM (Aud+Vis)** | **Δ BAM (Aud)** | **Δ BAM (Vis)** |
> > |---------|---------|-------------------|---------------|---------------|----------------------|------------------|------------------|
> > | **EMO** | 0.63 | 0.65 | 0.64 | 0.64 | +3.17% (+0.02) | +1.59% (+0.01) | +1.59% (+0.01) |
> > | **HUM** | 0.53 | 0.64 | 0.62 | 0.65 | +21.00% (+0.11) | +16.54% (+0.09) | +22.18% (+0.12) |
> > | **INT** | 0.26 | 0.18 | 0.26 | 0.18 | −30.77% (−0.08) | -- | −29.69% (−0.08) |
> > | **PTSD** | 1.00 | 1.00 | 1.00 | 1.00 | 0.00% (+0.00) | 0.00% (+0.00) | 0.00% (+0.00) |
> > | **ANX** | 0.91 | 0.91 | 0.91 | 0.91 | 0.00% (+0.00) | +0.11% (+0.00) | +0.11% (+0.00) |
> > | **DEP** | 0.73 | 0.79 | 0.78 | 0.73 | +8.21% (+0.06) | +6.85% (+0.05) | -- |
> > | **SEN** | 0.77 | 0.79 | 0.78 | 0.78 | +2.60% (+0.02) | +1.30% (+0.01) | +1.30% (+0.01) |
> > | **SAR** | 0.62 | 0.80 | 0.81 | 0.81 | +29.00% (+0.18) | +30.65% (+0.19) | +30.65% (+0.19) |
> > | **SOC** | 0.26 | 0.20 | 0.26 | 0.18 | −23.08% (−0.06) | -- | −29.96% (−0.08) |
> > | **NVC** | 0.12 | 0.16 | 0.12 | 0.18 | +33.00% (+0.04) | -- | +48.76% (+0.06) |

---

> > > ### Author Response · Authors · 2025-11-26
> > > **(Part 4) Update from Authors - Added new experiments and analyses according to reviewers' suggestions. Looking forward to engaging reviewers**
> > >
> > > 9. **Additional robustness study on LLM-as-Judge Reliability:**. To validate the robustness of LLM-judge evaluation, we conducted an additional study for human-LLM agreement for GPT-5-nano, and we have also added an additional model (Qwen3-30B) to compare inter-judge consistency and have experimented with a prompt variant. We find that GPT-5-nano exhibits
> > > strong agreement with human judgments, achieving a Cohen’s kappa of κ = 0.78. It also shows reason-
> > > able inter-judge consistency with Qwen3-30B (κ = 0.69). Qwen3-30B proves to be a weaker judge overall,
> > > with a lower human–judge agreement of κ= 0.64. To evaluate the robustness of LLM-judge performance
> > > to prompt variations, we construct an alternative judging prompt by automatically paraphrasing the existing
> > > prompt using GPT-5-nano and randomly reordering rubric items. Under this altered prompt, GPT-5-nano
> > > maintains high human–judge agreement (κ = 0.75). These findings collectively support the reliability and
> > > robustness of using LLM-based judges for scoring free-text generation tasks.\
> > > **Table Reliability and Robustness of LLM-Based Judges: Cohen’s κ between human annotations, GPT-5-nano, and Qwen3-30B across 500 sampled predictions from SOC, INT, and NVC tasks.**
> > > | **Judge**     | **Compared Against** | **Prompt Setting**                  | **Cohen’s κ** |
> > > |---------------|-----------------------|-------------------------------------|----------------|
> > > | GPT-5-nano    | Human                 | Original judge prompt               | 0.78           |
> > > | GPT-5-nano    | Qwen3-30B             | Original judge prompt               | 0.69           |
> > > | Qwen3-30B     | Human                 | Original judge prompt               | 0.64           |
> > > | GPT-5-nano    | Human                 | Paraphrased / reordered prompt      | 0.75           |
> > >
> > >
> > > We are committed to improving the paper according to the reviews, and kindly await your responses. Once again, we are grateful for your engagement in the review process.

---

> > > > ### Author Response · Authors · 2025-12-02
> > > > **(Part 5) Update from Authors - Added new experiments and analyses according to reviewers' suggestions. Looking forward to engaging reviewers**
> > > >
> > > > 10. **RL Ablation: Additional experiment on RL hyperparameters:**
> > > > We provide an additional ablation of our RL experiments, by altering the different rollouts and learning rates, as shown in the following Table. Beside showing the sensitivity of results to the different learning rates and rollouts, this highlights the rationale behind our choice of hyperparameters (rollouts of 5, learning rate of 5e-7), and learning algorithm (GRPO).
> > > >
> > > >
> > > > ### **Table: Ablation studies on number of rollouts, learning rate, and training method. Ablation studies on number of rollouts, learning rate, and training method. Best results are bolded. We report binary weighted F1 for SEN; mean per-class weighted accuracy for EMO; weighted F1 for HUM, SAR, ANX, DEP, PTSD; and LLM-Judge accuracy for SOC, INT, NVC.**
> > > >
> > > > ---
> > > >
> > > > ### **Number of Rollouts (lr = 5e-7)**
> > > >
> > > > | Model | CREMA-D | MELD (E) | MOSEI (E) | TESS | UR-FUNNY | IntentQA | PTSD_WILD | MMPSY (A) | MMPSY (D) | DAIC–WOZ | MELD (S) | CH-SIMSv2 | MOSEI (S) | MUStARD | Social-IQ 2.0 | MimeQA |
> > > > |------|---------|----------|-----------|------|----------|----------|------------|-----------|-----------|-----------|-----------|-------------|------------|----------|-----------------|---------|
> > > > | Rollouts = 2 | .500 | .563 | .576 | .499 | **.643** | --- | **.984** | .669 | .700 | --- | **.588** | .383 | **.253** | **.670** | --- | --- |
> > > > | Rollouts = 5 | **.501** | **.699** | **.581** | **.510** | .639 | **.486** | .968 | **.919** | .814 | **.729** | .571 | .393 | .224 | .647 | **.304** | **.133** |
> > > > | Rollouts = 20 | .495 | .572 | **.581** | .507 | .600 | --- | **.984** | .882 | **.841** | --- | .577 | **.403** | .252 | **.670** | --- | --- |
> > > >
> > > > ---
> > > >
> > > > ### **Learning Rate (rollouts = 5)**
> > > >
> > > > | Model | CREMA-D | MELD (E) | MOSEI (E) | TESS | UR-FUNNY | IntentQA | PTSD_WILD | MMPSY (A) | MMPSY (D) | DAIC–WOZ | MELD (S) | CH-SIMSv2 | MOSEI (S) | MUStARD | Social-IQ 2.0 | MimeQA |
> > > > |------|---------|----------|-----------|------|----------|----------|------------|-----------|-----------|-----------|-----------|-------------|------------|----------|-----------------|---------|
> > > > | lr = 1e-6 | .496 | .531 | **.588** | .500 | **.640** | --- | **.968** | .851 | **.843** | --- | .574 | .371 | .224 | .640 | --- | --- |
> > > > | lr = 5e-7 | **.501** | **.699** | .581 | **.510** | .639 | **.486** | **.968** | **.919** | .814 | **.729** | .571 | **.393** | **.224** | **.647** | **.304** | **.133** |
> > > > | lr = 1e-7 | **.501** | .521 | .577 | .508 | .636 | --- | .937 | .755 | .702 | --- | **.577** | .370 | .221 | .580 | --- | --- |
> > > >
> > > > ---
> > > >
> > > > ### **Training Method (rollouts = 5, lr = 5e-7)**
> > > >
> > > > | Model | CREMA-D | MELD (E) | MOSEI (E) | TESS | UR-FUNNY | IntentQA | PTSD_WILD | MMPSY (A) | MMPSY (D) | DAIC–WOZ | MELD (S) | CH-SIMSv2 | MOSEI (S) | MUStARD | Social-IQ 2.0 | MimeQA |
> > > > |------|---------|----------|-----------|------|----------|----------|------------|-----------|-----------|-----------|-----------|-------------|------------|----------|-----------------|---------|
> > > > | RLOO | **.501** | .585 | .573 | .499 | .613 | --- | **.984** | .877 | **.852** | --- | **.617** | **.404** | **.276** | **.675** | --- | --- |
> > > > | GRPO | **.501** | **.699** | **.581** | **.510** | **.639** | **.486** | .968 | **.919** | .814 | **.729** | .571 | .393 | .224 | .647 | **.304** | **.133** |

---

> ### Author Response · Authors · 2025-12-02
> **(Part 6) Update from Authors - Added new experiments and analyses according to reviewers' suggestions. Looking forward to engaging reviewers**
>
> We have completed our experiment on hyperparameters updated with the LLM judging results, including rollout size, learning rate and training method. Please find the results below:
>
> Best results are bolded. We report binary weighted F1 for SEN; mean per-class weighted accuracy for EMO; weighted F1 for HUM, SAR, ANX, DEP, PTSD; and LLM-Judge accuracy for SOC, INT, NVC.
>
> | Model | CREMA-D | MELD (E) | MOSEI (E) | TESS | UR-FUNNY | IntentQA | PTSD_WILD | MMPSY (A) | MMPSY (D) | MELD (S) | CH-SIMSv2 | MOSEI (S) | MUStARD | Social-IQ 2.0 | MimeQA |
> |-------|---------|----------|-----------|------|----------|----------|-----------|-----------|-----------|----------|-----------|-----------|---------|---------------|--------|
> | | **EMO** | | | | **HUM** | **INT** | **PTSD** | **ANX** | **DEP** | **SEN** | | | **SAR** | **SOC** | **NVC** |
> | **Number of Rollouts (lr = 5e-7)** | | | | | | | | | | | | | | | |
> | Rollouts = 2 | .500 | .563 | .576 | .499 | **.643** | .422 | **.984** | .669 | .700 | **.588** | .383 | **.253** | **.670** | .168 | .162 |
> | Rollouts = 5 | **.501** | **.699** | **.581** | **.510** | .639 | **.486** | .968 | **.919** | .814 | .571 | .393 | .224 | .647 | **.304** | .133 |
> | Rollouts = 20 | .495 | .572 | **.581** | .507 | .600 | .448 | **.984** | .882 | **.841** | .577 | **.403** | .252 | **.670** | .179 | **.178** |
> | **Learning Rate (rollouts = 5)** | | | | | | | | | | | | | | | |
> | lr = 1e-6 | .496 | .531 | **.588** | .500 | **.640** | .408 | **.968** | .851 | **.843** | .574 | .371 | .224 | .640 | .161 | .150 |
> | lr = 5e-7 | **.501** | **.699** | .581 | **.510** | .639 | **.486** | **.968** | **.919** | .814 | .571 | **.393** | **.224** | **.647** | **.304** | .133 |
> | lr = 1e-7 | **.501** | .521 | .577 | .508 | .636 | .399 | .937 | .755 | .702 | **.577** | .370 | .221 | .580 | .161 | **.174** |
> | **Training Method (rollouts = 5, lr = 5e-7)** | | | | | | | | | | | | | | | |
> | RLOO | **.501** | .585 | .573 | .499 | .613 | .436 | **.984** | .877 | **.852** | **.617** | **.404** | **.276** | **.675** | .173 | **.170** |
> | GRPO | **.501** | **.699** | **.581** | **.510** | **.639** | **.486** | .968 | **.919** | .814 | .571 | .393 | .224 | .647 | **.304** | .133 |
>
> In general, we see a learning rate of 5e-7 and rollout size of 5 to be a sweet spot for training. Performance difference tends to be small, as training runs with different hyperparameters usually converges to similar performance at different speeds.

---

### Author Response · Authors · 2025-11-27
**Author's Note: Added experiments and analyses, kindly awaiting reviewer responses**

Dear Reviewers,

We thank you once again for investing time in the review process. The authors have added a number of experiments to improve the paper according to reviewer feedback, and have also incorporated additional analysis to strengthen the paper's discussion. As the rebuttal period for ICLR is coming to an end in the next few days, we hope to hear from you, and are kindly awaiting your responses.

Best,

Authors'

---

### Meta-Review · Area_Chair_ZBYU · 2026-01-07

**Summary:**

Reviewers recognized Human Behavior Atlas as a useful benchmark effort that unifies a wide range of psychological and social behavior datasets under a single multimodal, instruction-based framework. Strengths noted across reviews include the breadth of task coverage, standardized evaluation protocols, public release of resources, and extensive empirical analysis of multi-task and transfer learning.

The primary concerns raised in the initial reviews include:
(1) limited novelty due to the lack of new raw data collection and architectural innovation;
(2) fairness and interpretation of baseline comparisons and transfer-learning evaluations;
(3) insufficient ablation studies and analysis;
(4) task formulation choices such as label binarization and metric heterogeneity.

**Reviewer Concerns:**

***Concerns Addressed by the Rebuttal***

Benchmark novelty and data contribution (PK4q, aP6y):
The authors clearly clarified the paper’s scope as a benchmark contribution, aligning it with previous works (e.g., GLUE/SuperGLUE). They further demonstrated that the work involves substantial new data creation in the form of hand-crafted instruction prompts, behavioral descriptors, and a unified behavioral taxonomy, which extend beyond simple dataset aggregation.

Baseline fairness and transfer learning evaluation (PK4q, aP6y, tGVo):
Additional comparisons were added against HumanOmniV2 and other multimodal models. The authors clarified experimental intent and strengthened transfer evaluation with (i) strict zero-shot experiments, (ii) minimal-budget fine-tuning protocols, and (iii) cross-dataset transfer to IEMOCAP.

Ablation studies and LLM-judge robustness (eg78):
The rebuttal substantially expanded empirical analysis, including ablation studies on BAM and RL parameter choices, LLM-as-judge reliability, and Computational efficiency analysis.


Prompt construction and evaluation protocol transparency (aP6y, eg78):
The authors clarified that prompts are hand-crafted, documented isolation between model and judge, and provided detailed evaluation protocol descriptions.

General-purpose capability preservation (aP6y):
Additional experiments demonstrated that benchmark training does not degrade text generation quality.

***Remaining / Partially Addressed Concerns***

Architectural innovation:
Some reviewers may still view the absence of a novel model architecture as a limitation. However, this primarily reflects a difference in evaluation criteria rather than a technical deficiency, given the paper’s explicit positioning as a benchmark contribution rather than a model-design paper.

**Reviewer Scores:**

Reviewer PK4q (initial score: 4):

Given the added baselines, clarifications, and transfer learning experiments, this score could reasonably increase to 6. However, it may also remain at 4 if the reviewer continues to place primary emphasis on the absence of technical model novelty.

Reviewer aP6y (initial score: 4):

Most of the reviewer’s concerns, including data contribution, zero-shot transfer, prompt construction, generalization, and descriptor ablations, were directly addressed in the rebuttal. The likely outcome is an increase to 6.

Reviewer eg78 (initial score: 6):

The rebuttal directly addressed the requested experiments and analyses. The score would likely be maintained at 6, with a possible increase to 8.

Reviewer tGVo (initial score: 6):

Concerns regarding fusion strategies, regression tasks, and transfer learning were discussed and supplemented with additional analysis. The score would likely be maintained at 6, potentially reflecting continued reservations about limited architectural innovation.

Reviewer 3tAt:

This review was flagged for insufficient quality and was not materially considered in the final assessment.

***Overall Assessment and Recommendation:***

This work provides a valuable and well-executed benchmark for unified multimodal psychological and social behavior understanding. Although it does not introduce new model architectures or collect new raw data, the authors clearly position the contribution as benchmark-driven. The rebuttal substantially strengthens the submission through extensive additional experiments, analyses, and clarifications. The resulting benchmark, evaluation framework, and empirical findings are likely to be useful to the community and to stimulate future research on unified behavioral modeling. The AC therefore recommends acceptance.

---

### Decision · Program_Chairs · 2026-01-26

Accept (Poster)